# Supply-Side Equilibria in Recommender Systems

**Meena Jagadeesan**
UC Berkeley
mjagadeesan@berkeley.edu

**Nikhil Garg**
Cornell Tech
ngarg@cornell.edu

**Jacob Steinhardt**
UC Berkeley
jsteinhardt@berkeley.edu

## Abstract

Algorithmic recommender systems such as Spotify and Netflix affect not only consumer behavior but also *producer incentives*. Producers seek to create content that will be shown by the recommendation algorithm, which can impact both the diversity and quality of their content. In this work, we investigate the resulting supply-side equilibria in personalized content recommender systems. We model the decisions of producers as choosing *multi-dimensional* content vectors and users as having *heterogenous* preferences, which contrasts with classical low-dimensional models. Multi-dimensionality and heterogeneity creates the potential for *specialization*, where different producers create different types of content at equilibrium. Using a duality argument, we derive necessary and sufficient conditions for whether specialization occurs. Then, we characterize the distribution of content at equilibrium in concrete settings with two populations of users. Lastly, we show that specialization can enable producers to achieve *positive profit at equilibrium*, which means that specialization can reduce the competitiveness of the marketplace. At a conceptual level, our analysis of supply-side competition takes a step towards elucidating how personalized recommendations shape the marketplace of digital goods.

## 1 Introduction

Algorithmic recommender systems have disrupted the production of digital goods such as movies, music, and news. In the music industry, artists have changed the length and structure of songs in response to Spotify's algorithm and payment structure [Hodgson, 2021]. In the movie industry, personalization has led to low-budget films catering to specific audiences [McDonald, 2019], in some cases constructing data-driven "taste communities" [Adalian, 2018]. Across industries, recommender systems shape how producers decide what content to create, influencing the supply side of the digital goods market. This raises the questions: *What factors drive and influence the supply-side marketplace? What content will be produced at equilibrium?*

Intuitively, supply-side effects are induced by the multi-sided interaction between producers, the recommendation algorithm, and users. Users tend to follow recommendations when deciding what content to consume [Ursu, 2018]—thus, recommendations influence how many users consume each digital good and impact the profit (or utility) generated by each content producer. As a result, content producers shape their content to maximize appearance in recommendations; this creates competition between the producers, which can be modeled as a game. However, understanding such producer-side effects has been difficult, both empirically and theoretically. This is a pressing problem, as these gaps in understanding have hindered the regulation of digital marketplaces [Stigler Committee, 2019].

37th Conference on Neural Information Processing Systems (NeurIPS 2023).

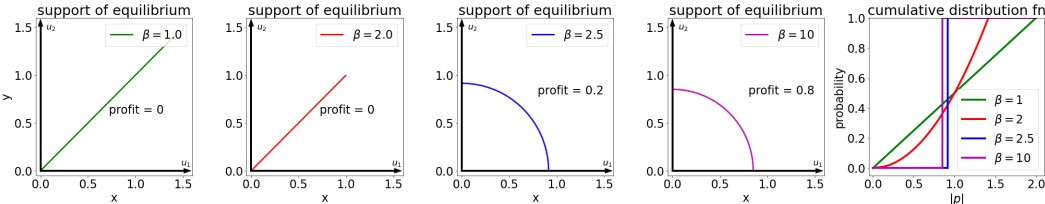

Figure 1: A symmetric equilibrium for different settings of $\beta$, for 2 users located at the standard basis vectors $e_1$ and $e_2$, $P = 2$ producers, and producer cost function $c(p) = \|p\|_2^\beta$. The first 4 plots show the support of the equilibrium $\mu$. As $\beta$ increases, there is a phase transition from a single-genre equilibrium to an equilibrium with infinitely many genres (Theorem 2). The profit also transitions from zero to positive, demonstrating how specialization reduces the competitiveness of the marketplace (Propositions 2-3). The last plot shows the cumulative distribution function of $\|p\|$.

At a high level, there are two primary challenges that complicate theoretical analyses of these supply-side effects. (1) Digital goods such as movies have many attributes and thus must be embedded in a *multi-dimensional* continuous space, leading to a large producer action space. This multi-dimensionality is a departure from traditional economic models of price and spatial competition. (2) A core aspect of such marketplaces is the potential for specialization: that is, different producers may produce different items at equilibrium. Incentives to specialize depend on the level of *heterogeneity of user preferences* and the cost structure for producing goods (whether it is more expensive to produce items that are good in multiple dimensions). As a result, supply-side equilibria have the potential to exhibit rich economic phenomena, but pose a challenge to both modeling and analysis.

We introduce a simple game-theoretic model for supply-side competition in personalized recommender systems, that captures the multi-dimensional space of producer decisions, rich structures of production costs, and general configurations of users. Users and digital goods are represented as $D$-dimensional vectors, and the value that a user with vector $u$ derives from a digital good $p$ is equal to the inner product $\langle u, p \rangle$. The platform offers personalized recommendations to each user, showing them the digital good with maximum value to them. Each producer *chooses* a single digital good $p \in \mathbb{R}_{\geq 0}^D$ to create. The goal of a producer to maximize their profit, which is equal to the number of users who are recommended their content minus the (one-time) cost of producing the content. We study the Nash equilibria of this game between producers. We defer model specifics to Section 2.

To ground our investigation of the complex producer choices, we focus on one particular economic phenomena—*the potential for specialization*. Specialization, which occurs when different producers create different *genres* (i.e., directions) of content at equilibrium, has several economic consequences. For example, whether specialization occurs, as well as the form that specialization takes, determines the diversity of content available on the platform. Moreover, specialization influences market competitiveness by reducing the amount of competition in each genre. This raises the questions:

> *Under what conditions does specialization occur at equilibrium? What is its impact on market competitiveness? What form does specialization take?*

We investigate specialization and its consequences on the supply-side market. We analyze how the specialization exhibited at equilibrium varies with user geometry and producer cost function parameters. Figure 1 and 3 depict the equilibria in our model across several concrete instantiations. Our main results provide insight into each of the questions from above: we characterize when specialization occurs (Section 3) and investigate the impact of specialization on market competitiveness (Section 4). We provide an empirical analysis of supply-side equilibria using the MovieLens-100K dataset [Harper and Konstan, 2015] (Figure 4), and we analyze the particular form that specialization takes in the case of 2 equally sized populations (deferred to Appendix D).

Our simple model yields a nuanced picture of supply-side equilibria in recommender systems. Our results provide insight into specialization and its implications, and en route to proving these results, we develop a technical toolkit to analyzing the multi-dimensional behavior of producers (see Appendix C.1 for a summary). More broadly, our model and results open the door to investigating how recommender systems shape the diversity and quality of content created by producers.

## 1.1 Related work

Our work is related to research on *societal effects in recommender systems*, *models of competition in economics*, and *strategic effects induced by algorithmic decisions*.

**Supply-side effects of recommender systems.** A line of work in the machine learning literature has studied supply-side effects from a theoretical perspective, but existing models do not capture the infinite, multi-dimensional decision space of producers. Ben-Porat and Tennenholtz [2018] study supply-side effects with a focus on mitigating strategic effects by content producers; Ben-Porat et al. [2020], building on Basat et al. [2017], also studied supply-side equilibria with a focus on convergence of learning dynamics for producers. The main difference from our work is that producers in these models choose a topic from a *finite* set of options; in contrast, our model captures the infinite, multi-dimensional producer decision space that drives the emergence of genres. Moreover, we focus on the structure of equilibria rather than the convergence of learning.

In concurrent and independent work, Hron et al. [2022] study a related model for supply-side competition in recommender systems where producers choose content embeddings in $\mathbb{R}^D$. One main difference is that, rather than having a cost on producer content, they constrain producer vectors to the $\ell_2$ unit ball (this corresponds to our model when $\beta \to \infty$ and the norm is the $\ell_2$-norm, although the limit behaves differently than finite $\beta$). Additionally, Hron et al. incorporate a softmax decision rule to capture exploration and user non-determinism, whereas we focus entirely on hardmax recommendations. Thus, our model focuses on the role of producer costs while Hron et al.'s focuses on the role of the recommender environment. Technically, Hron et al. study the existence of different types of equilibria and the use of behaviour models for auditing, whereas we analyze the economic phenomena exhibited by symmetric mixed strategy Nash equilibria, with a focus on specialization.

Other work has studied the emergence of filter bubbles [Flaxman et al., 2016], the ability of users to reach different content [Dean et al., 2020], the shaping of user preferences [Adomavicius et al., 2013], and stereotyping [Guo et al., 2021].

**Models of competition in microeconomics.** Our model and research questions relate to classical models of competition in economics; however, particular aspects of recommender systems—high-dimensionality of digital goods, rich structure of producer costs, and user geometry—are not captured by these classical models. For example, in *price competition*, producers set a *price*, but do not decide what good to produce (e.g. Bertrand competition, see [Baye and Kovenock, 2008] for a textbook treatment). Price is one-dimensional, but producer decisions in our model are multi-dimensional.

Another line of work on *product selection* has investigated how producers choose goods (i.e., content) at equilibrium (see Anderson et al. [1992] for a textbook treatment). For example, in *spatial location models*, producers choose a direction in a low-dimensional space (e.g., $\mathbb{R}^1$ in [Hotelling, 1929, d'Aspremont et al., 1979] and $\mathbb{S}^1$ in [Salop, 1979]), and users typically receive utility based on the negative of the Euclidean distance. In contrast, producers in our model jointly select direction and magnitude, and users receive utility based on inner product. Since some spatial location models allow producers to set prices, it may be tempting to draw an analogy between $\|p\|$ in our model and the price in these models. However, this analogy breaks down because costs can be nonlinear in $\|p\|$.

Other related work has investigated *supply function equilibria* (e.g. [Grossman, 1981]), where the producer chooses a function from quantity to prices, rather than what content to produce, and the *pure characteristics model* (e.g. [Berry, 1994]), where attributes of users and producers are also embedded in $\mathbb{R}^D$ like in our model, but which focuses on demand estimation for a fixed set of content, rather than analyzing the content that arises at equilibrium in the marketplace. Recent work in economics has extended Berry [1994] to allow for endogenous product choice (e.g. [Wollmann, 2018]) and also studied specialization (e.g. [Vogel, 2008, Perego and Yuksel, 2022]), though with different modeling assumptions than our work.

**Strategic classification.** A line of work of *strategic classification* [Brückner et al., 2012, Hardt et al., 2016] has studied how classification can induce participants to strategically change their features to improve their outcomes. Models for participant behavior (e.g. Kleinberg and Raghavan [2019], Jagadeesan et al. [2021], Ghalme et al. [2021]) generally do not capture competition between participants. One exception is Liu et al. [2022], where participants compete to appear higher in a ranking; in contrast, the participants in our model compete for users with *heterogeneous* preferences.

## 2 Model and Preliminaries

We introduce a game-theoretic model for supply-side competition in recommender systems. Consider a platform with $N \geq 1$ heterogeneous users who are offered personalized recommendations and $P \geq 2$ producers who strategically decide what digital good to create.

**Embeddings of users and digital goods.** Each user $i$ is associated with a $D$-dimensional embedding $u_i$ that captures their preferences. We assume that $u_i \in \mathbb{R}_{\geq 0}^D \setminus \{\vec{0}\}$—i.e., the coordinates of each embedding are nonnegative and each embedding is nonzero. While user vectors are fixed, producers *choose* what content to create. Each producer $j$ creates a single digital good, which is associated with a content vector $p_j \in \mathbb{R}_{\geq 0}^D$. The value of good $p$ to user $u$ is $\langle u, p \rangle$.

**Personalized recommendations.** After the producers decide what content to create, the platform offers personalized recommendations to each user. We consider a stylized model where the platform has complete knowledge of the user and content vectors. The platform recommends to each user the content (and thus producer) of maximal value to them. Mathematically, the platform assigns a user $u$ to the producer $j^*$, where $j^*(u; \ p_{1:P}) = \arg\max_{1 \leq j \leq P} \langle u, p_j \rangle$. If there are ties, the platform sets $j^*(u; \ p_{1:P})$ to be a producer chosen uniformly at random from the argmax.

**Producer cost function.** Each producer faces a *fixed (one-time) cost* for producing content $p$, which depends on the magnitude of $p$. Since the good is digital and thus cheap to replicate, the production cost does not scale with the number of users. We assume that the cost function $c(p)$ takes the form $\|p\|^{\beta}$, where $\|\cdot\|$ is any norm and the exponent $\beta$ is at least 1. The magnitude $\|p\|$ captures the *quality* of the content: in particular, if a producer chooses content $\lambda p$, they win at least as many users as if they choose $\lambda' p$ for $\lambda' < \lambda$. (This relies on the fact that all vectors are in the positive orthant.) The norm and $\beta$ together encode the cost of producing a content vector $v$, and reflect cost tradeoffs for excelling in different dimensions (for example, producing a movie that is both a drama and a comedy). Large $\beta$, for instance, means that this cost grows superlinearly. In Section 3, we will see that these tradeoffs capture the extent to which producers are incentivized to specialize.

**Producer profit.** A producer receives profit equal to the number of users who are recommended their content minus the cost of producing the content. The profit of producer $j$ is equal to:

$$\mathcal{P}(p_j; p_{-j}) = \mathbb{E}\Big[\Big(\sum_{i=1}^{n} \mathbb{1}[j^*(u_i; \ p_{1:P}) = j]\Big) - \|p_j\|^{\beta}\Big], \tag{1}$$

where $p_{-j} = [p_1, \ldots, p_{j-1}, p_{j+1}, \ldots, p_P]$ denotes the content produced by all of the other producers; the expectation is over the randomness over platform recommendations in the case of ties.

We defer further discussion of our model to Appendix B.1.

### 2.1 Equilibrium concept and existence of equilibrium

We study the Nash equilibria of the game between producers. Each producer $j$ chooses a (random) strategy over content, given by a probability measure $\mu_j$ over the content embedding space $\mathbb{R}_{\geq 0}^D$. Recall that strategies $(\mu_1, \ldots, \mu_P)$ form a *Nash equilibrium* if no producer—given the strategies of other producers—can chose a different strategy where they achieve higher expected profit.[1]

A salient feature of our model is that there are discontinuities in the producer utility function in equation (1), since the function $\arg\max_{1 \leq i \leq P} \langle u_i, p_j \rangle$ changes discontinuously with the producer vectors $p_j$. Due to these discontinuities, pure strategy equilibria do not exist (Proposition 4 in Appendix B.2). Nonetheless, we show that a symmetric mixed strategy equilibrium (i.e. an equilibrium where $\mu_1 = \ldots = \mu_P$) is nonetheless guaranteed to exist.

**Proposition 1.** *For any set of users and any $\beta \geq 1$, a symmetric mixed equilibrium exists.*

These equilibria are interestingly atomless (Proposition 5 in Appendix B.2), so there is randomization over *quality* $\|p\|$ as well as randomness over *genres* $p/\|p\|$. We defer an example of the equilibria

---

[1]That is, $p_j \in \arg\max_{p \in \mathbb{R}_{\geq 0}^D} \mathbb{E}_{p_{-j} \sim \mu_{-j}}[\mathcal{P}(p_j; p_{-j})]$ for every $j \in [P]$ and every $p_j \in \text{supp}(\mu_j)$.

in our model for homogenous users to Appendix B.3. We take the symmetric mixed equilibria of this game as the main object of our study, since they are both tractable to analyze and rich enough to capture asymmetric solution concepts such as specialization.

## 2.2 Specialization and the formation of genres

When users are heterogeneous, there are inherent tensions between catering to one user and catering to other users. As a result, the producer make nontrivial choices not only about the *quality* (vector norm $\|p\|$) of their content, but also the *genre* (direction $p/\|p\|$) of their content. This can lead to *specialization*, which is when different producers create goods tailored to different users; alternatively, all producers might still produce the same genre of content at equilibrium and thus only exhibit differentiation on the axis of quality.

We define *genres* as the set of *directions* that arise at a symmetric mixed Nash equilibrium $\mu$:

$$\text{Genre}(\mu) := \Big\{ \frac{p}{\|p\|} \mid p \in \text{supp}(\mu) \Big\}, \tag{2}$$

where we normalize by $\|p\|$ to separate out the quality (norm) from the genre (direction). The set of genres $\text{Genre}(\mu)$ captures the set of content that may arise on the platform in some realization of randomness of the producers' strategies. When an equilibrium has a single genre, all producers cater to an average user, and only a single type of content appears. On the other hand, when an equilibrium has multiple genres, many types of digital content are likely to appear on the platform.

We thus say that *specialization* occurs at an equilibrium $\mu$ if and only if the support of $\mu$ has more than one *genre* (direction).[2] The particular form of specialization exhibited by $\mu$ is further captured by the number and set of genres in the support of $\mu$. See Figure 1 for a depiction of markets with a single- and multi-genre equilibria, respectively.

# 3 When does specialization occur?

In order to investigate whether specialization occurs in a given marketplace, we distinguish between two regimes of marketplaces based on whether or not a single-genre equilibrium exists:

1. A marketplace is in the *single-genre regime* if there exists an equilibrium $\mu$ such that $|\text{Genre}(\mu)| = 1$. All producers thus create content of the same genre.
2. A marketplace is in the *multi-genre regime* if all equilibria $\mu$ satisfy $|\text{Genre}(\mu)| > 1$. Producers thus necessarily differentiate in the genre of content that they produce.

To understand these regimes, we ask: *what conditions on the user vectors $u_1, \ldots, u_N$ and the cost function parameters $\| \cdot \|$ and $\beta$ determine which regime the marketplace is in?*

In Section 3.1, we give necessary and sufficient conditions for all equilibria to have multiple genres (Theorem 1). In Section 3.2, we show several corollaries of Theorem 1. In Section 3.3, we show that the location of the single-genre equilibrium (in cases where it exists) maximizes the Nash social welfare. In Section 3.4, we provide an empirical analysis using the MovieLens-100K dataset [Harper and Konstan, 2015] that validate our theoretical results. We defer proofs of these results to Appendix F.

## 3.1 Characterization of single-genre and multi-genre regimes

We first provide a tight geometric characterization of when a marketplace is in the single-genre regime versus in the multi-genre regime. More formally, let $\mathbf{U} = [u_1; \cdots; u_N]$ be the $N \times D$ matrix of user vectors, and let $\mathcal{S}$ denote the image of the unit ball under $\mathbf{U}$:

$$\mathcal{S} := \big\{ \mathbf{U}p \mid \|p\| \leq 1, p \in \mathbb{R}^D_{\geq 0} \big\} \tag{3}$$

Each element of $\mathcal{S}$ is an $N$-dimensional vector, which represents the user values for some unit-norm producer $p$. Additionally, let $\mathcal{S}^\beta$ be the image of $\mathcal{S}$ under coordinate-wise powers, i.e. if

---

[2]In this definition, we implicitly interpret the randomness in the producer strategies as differentiation between producers. This formalization of specialization obviates the need to reason about asymmetric equilibria, thus making the model much more tractable to analyze.

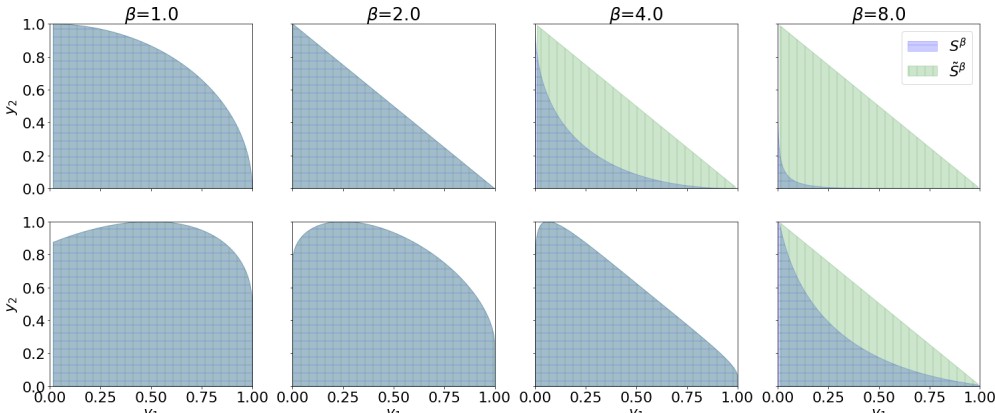

Figure 2: The sets $\mathcal{S}^\beta$ and $\bar{\mathcal{S}}^\beta$ for two configurations of user vectors (rows) and settings of $\beta$ (columns). The user vectors are $[1, 0], [0, 1]$ (top, same as Figure 1) and $[1, 0], [0.5, 0.87]$ (bottom). $\mathcal{S}^\beta$ transitions from convex to non-convex as $\beta$ increases, though the transition point depends on the user vectors. When $\mathcal{S}^\beta$ is convex, a single vector $p$ can more easily satisfy both users at low cost.

$(z_1, \ldots, z_N) \in \mathcal{S}$ then $(z_1^\beta, \ldots, z_N^\beta) \in \mathcal{S}^\beta$. We show that genres emerge when $\mathcal{S}^\beta$ is sufficiently different from its convex hull $\bar{\mathcal{S}}^\beta$:

**Theorem 1.** *Let* $\mathbf{U} := [u_1; \cdots; u_N]$, *let* $\mathcal{S}$ *be* $\{\mathbf{U}p \mid \|p\| \leq 1, p \in \mathbb{R}_{\geq 0}^D\}$, *and let* $\mathcal{S}^\beta$ *be the image of* $\mathcal{S}$ *under coordinate-wise powers. Then, there is a symmetric equilibrium* $\mu$ *with* $|\mathrm{Genre}(\mu)| = 1$ *if and only if*

$$\max_{y \in \mathcal{S}^\beta} \prod_{i=1}^N y_i = \max_{y \in \bar{\mathcal{S}}^\beta} \prod_{i=1}^N y_i. \tag{4}$$

*Otherwise, all symmetric equilibria have multiple genres. Moreover, if* (4) *holds for some* $\beta$, *it also holds for every* $\beta' \leq \beta$.

Theorem 1 relates the existence of a single-genre equilibrium to the convexity of the set $\mathcal{S}^\beta$. As a special case, the condition in Theorem 1 always holds if $\mathcal{S}^\beta$ is convex, but is strictly speaking weaker than convexity. Interestingly, the condition depends on the geometry of the user embeddings $u_1, \ldots, u_n$ and the cost function $c(p) = \|p\|^\beta$ but *not* on the number of producers $P$. Intuitively, convexity of $\mathcal{S}^\beta$ relates to the ease with which a vector $p$ can satisfy all users simultaneously, at low cost—each dimension of $\mathcal{S}$ corresponds to a user's utility. In Figure 2, we display the sets $\mathcal{S}^\beta$ and $\bar{\mathcal{S}}^\beta$ for different configurations of user vectors and different settings of $\beta$.

Theorem 1 further shows that the boundary between the single-genre and multi-genre regimes can be represented by a *threshold* defined as follows

$$\beta^* := \sup \left\{ \beta \geq 1 \mid \max_{y \in \mathcal{S}^\beta} \prod_{i=1}^N y_i = \max_{y \in \bar{\mathcal{S}}^\beta} \prod_{i=1}^N y_i \right\}.$$

where single-genre equilibria exist exactly when $\beta \leq \beta^*$. Conceptually, larger $\beta$ make producer costs more superlinear, which eventually discentivizes producers from attempting to perform well on all dimensions at once.

The proof of Theorem 1 draws a connection to minimax theory in optimization. In particular, we show that the existence of a single-genre equilibrium is equivalent to strong duality holding for a certain optimization program that we define (Lemma 2). This allows us to leverage techniques from optimization theory to analyze when specialization occurs. We defer a full proof to Appendix F.1.

### 3.2 Corollaries of Theorem 1

To further understand the condition in equation (4), we consider a series of special cases that provide intuition for when single-genre equilibria exist (proofs deferred to Section F.2). First, let us consider

$\beta = 1$, in which case the cost function is a norm. Then $\mathcal{S}^1 = \mathcal{S}$ is convex, so a single-genre equilibrium always exists.

**Corollary 1.** *The threshold $\beta^*$ is always at least $1$. That is, if $\beta = 1$, there exists a single-genre equilibrium.*

The economic intuition is that norms incentivize averaging rather than specialization.

We next take a closer look at how the choice of norm affects the emergence of genres. For cost functions $c(p) = \|p\|_q^\beta$, we show that $\beta^* \geq q$ for any set of user vectors, with equality achieved at the standard basis vectors.

**Corollary 2.** *Let the cost function be $c(p) = \|p\|_q^\beta$. For any set of user vectors, it holds that $\beta^* \geq q$. If the user vectors are equal to the standard basis vectors $\{e_1, \ldots, e_D\}$, then $\beta^*$ is equal to $q$.*

Corollary 2 illustrates that the threshold $\beta^*$ relates closely to the convexity of the cost function and whether the cost function is superlinear. In particular, the cost function must be sufficiently nonconvex for all equilibria to be multi-genre. For example, for the $\ell_\infty$-norm, where producers only pay for the highest magnitude coordinate, it is never possible to incentivize specialization: there exists a single-genre equilibrium regardless of $\beta$. On the other hand, for norms where costs aggregate nontrivially across dimensions, specialization is possible.

In addition to the choice of norm, the geometry of the user vectors also influences whether multiple genres emerge. To illustrate this, we first show that in a concrete market instance with 2 equally sized populations of users, the threshold depends on the cosine similarity between the two user vectors:

**Corollary 3.** *Suppose that there are $N$ users split equally between two linearly independently vectors $u_1, u_2 \in \mathbb{R}_{\geq 0}^D$, and let $\theta^* := \cos^{-1}\left(\frac{\langle u_1, u_2 \rangle}{\|u_1\|_2 \|u_2\|}\right)$. If the cost function is $c(p) = \|p\|_2^\beta$, then,*

$$\beta^* = \frac{2}{1 - \cos(\theta^*)}.$$

Corollary 3 demonstrates the threshold $\beta^*$ increases as the angle $\theta^*$ between the users decreases (i.e. as the users become closer), because it is easier to simultaneously cater to all users. In particular, $\beta^*$ interpolates from 2 when the users are orthogonal to $\infty$ when the users point in the same direction.

Finally, we consider general configurations of users and cost functions, and we upper bound $\beta^*$:

**Corollary 4.** *Let $\|\cdot\|_*$ denote the dual norm of $\|\cdot\|$, defined to be $\|p\|_* = \max_{\|p\|=1, p \in \mathbb{R}_{\geq 0}^D} \langle q, p \rangle$. Let $Z := \|\sum_{n=1}^N \frac{u_n}{\|u_n\|_*}\|_*$. Then,*

$$\beta^* \leq \frac{\log(N)}{\log(N) - \log(Z)}. \tag{5}$$

In equation (5), the upper bound on the threshold $\beta^*$ increases as $Z$ increases. As an example, consider the cost function $c(p) = \|p\|_2^\beta$. We see that if the user vectors point in the same direction, then $Z = N$ and the right-hand side of (5) is $\infty$. On the other hand, if $u_1, \ldots, u_n$ are orthogonal, then $Z = \sqrt{N}$ and the right-side of (5) is 2, which exactly matches the bound in Corollary 2. In fact, for *random* vectors $u_1, \ldots, u_N$ drawn from a truncated gaussian distribution, we see that $Z = \tilde{O}(\sqrt{N})$ in expectation, in which case the right-hand side of (5) is close to 2 as long as $N$ is large. Thus, for many (but not all) choices of user vectors, even small values of $\beta$ are enough to induce multiple genres. In Section 3.4, we compute compute the right-hand side of (5) on user embeddings generated from the MovieLens dataset for different cost functions.

### 3.3 Location of single-genre equilibrium

We next study where the single-genre equilibrium is located, in cases where it exists. As a consequence of the proof of Theorem 1, we can show that the location of the single-genre equilibrium maximizes the *Nash social welfare* Nash [1950] of the users.

**Corollary 5.** *If there exists $\mu$ with $|\mathrm{Genre}(\mu)| = 1$, then the corresponding producer direction maximizes Nash social welfare of the users:*

$$\mathrm{Genre}(\mu) = \underset{\|p\|=1 | p \in \mathbb{R}_{\geq 0}^D}{\arg\max} \sum_{i=1}^N \log(\langle p, u_i \rangle). \tag{6}$$

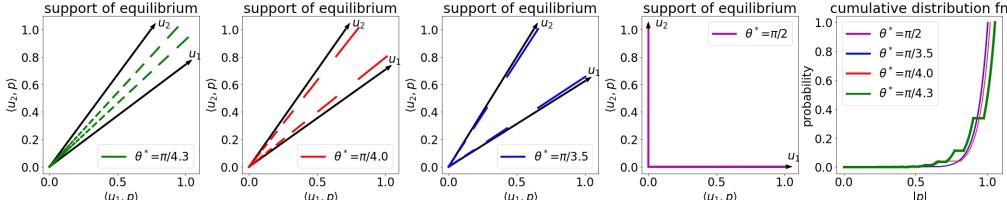

Figure 3: A symmetric equilibrium for different settings of $\theta^*$, for 2 users located at $u_1$ and $u_2$ such that $\theta^* = \cos^{-1}\left(\frac{\langle u_1, u_2 \rangle}{\|u_1\|\|u_2\|}\right)$, for producer cost function $c(p) = \|p\|_2^\beta$ with $\beta = 7$, and for $P = \infty$ producers (see Theorem 3). The first 4 plots show the support of the equilibrium in a reparameterized space: note that the the x-axis is $\langle u_1, p \rangle$ and the y-axis is $\langle u_2, p \rangle$, i.e., the values that the users derive from good $p$.

Corollary 5 demonstrates that the single-genre equilibrium directions maximizes the Nash social welfare Nash [1950] for users. Interestingly, this measure of welfare for *users* is implicitly maximized by *producers* competing with each other in the marketplace. Properties of the Nash social welfare are thus inherited by single-genre equilibria.

### 3.4 Empirical analysis on the MovieLens dataset

We provide an empirical analysis of supply-side equilibria using the MovieLens-100K dataset and recommendations based on nonnegative matrix factorization (NMF). In particular, we compute the single-genre equilibrium direction for different cost functions as well as an upper bound on $\beta^*$ (i.e., the threshold where specialization starts to occur) for different values of the dimension $D$. These experiments provide qualitative insights that offer additional intuition for our theoretical results.

We focus on the rich family of cost functions $c_{q,\alpha,\beta}(p) = \| [p_1 \cdot \alpha_1, \ldots, p_D \cdot \alpha_D] \|_q^\beta$ parameterized by weights $\alpha \in \mathbb{R}_{\geq 0}^D$, parameter $q \geq 1$, and cost function exponent $\beta \geq 1$. The weights $\alpha \in \mathbb{R}_{\geq 0}^D$ capture asymmetries in the costs of different dimensions (a higher value of $\alpha_i$ means that dimension $i$ is more costly). The parameters $q$ and $\beta$ together capture the tradeoffs between improving along a single dimension versus simultaneously improving among many dimensions. To isolate the impact of each parameter, we either fix $q = 2$ and vary $\alpha$ (and $\beta$) or we fix $\alpha = [1, 1, \ldots, 1]$ and vary $q$ (and $\beta$).

**Setup.** The MovieLens 100K dataset consists of 943 users, 1682 movies, and 100,000 ratings [Harper and Konstan, 2015]. For $D \in \{2, 3, 5, 10, 50\}$, we obtain $D$-dimensional user embeddings by running NMF (with $D$ factors) using the scikit-surprise library. We calculate the single-genre equilibrium genre $p^* = \arg\max_{\|p\|=1 | p \in \mathbb{R}_{\geq 0}^D} \sum_{i=1}^N \log(\langle p, u_i \rangle)$ (Corollary 5) by solving the optimization program, using the cvxpy library for $q = 2$ and projected gradient descent for $q \neq 2$. We calculate the upper bound $\beta_u := \frac{\log(N)}{\log(N) - \log\left(\| \sum_{n=1}^N \frac{u_n}{\|u_n\|_*} \|_*\right)} \geq \beta^*$ from Corollary 4.[3]

**Single-genre equilibrium direction $p^*$.** Figures 4a-4d show the direction of the single-genre equilibrium $p^*$ across different cost functions. These plots uncover several properties of the genre $p^*$. First, the genre generally does not coincide with the arithmetic mean of the users. Moreover, the genre varies significantly with the weights $\alpha$. In particular, the magnitude of the dimension $p_i$ is higher if $\alpha_i$ is *lower*, which aligns with the intuition that producers invest more in cheaper dimensions. In contrast, the genre turns out to not change significantly with the norm parameter. Altogether, these insights illustrate that how the genre can be influenced by specific aspects of producer costs.

**Upper bound on the threshold $\beta^*$ where specialization starts to occur.** Figure 4e shows the value of $\beta_u$ across different values of $D$ and $q$. As the dimension $D$ increases, the upper bound $\beta_u$ decreases, indicating that specialization is more likely to occur. The intuition is that $D$ amplifies the heterogeneity of user embeddings, subsequently increasing the likelihood of specialization. This insight has an interesting consequence for platform design: *the platform can influence the level of specialization by tuning the number of factors $D$ used in matrix factorization.* Producer costs also impact whether specialization occurs: as the norm $q$ increases, the value of $\beta_u$ increases and specialization is less likely to occur.

---

[3]See Appendix A and https://github.com/mjagadeesan/supply-side-equilibria for details.

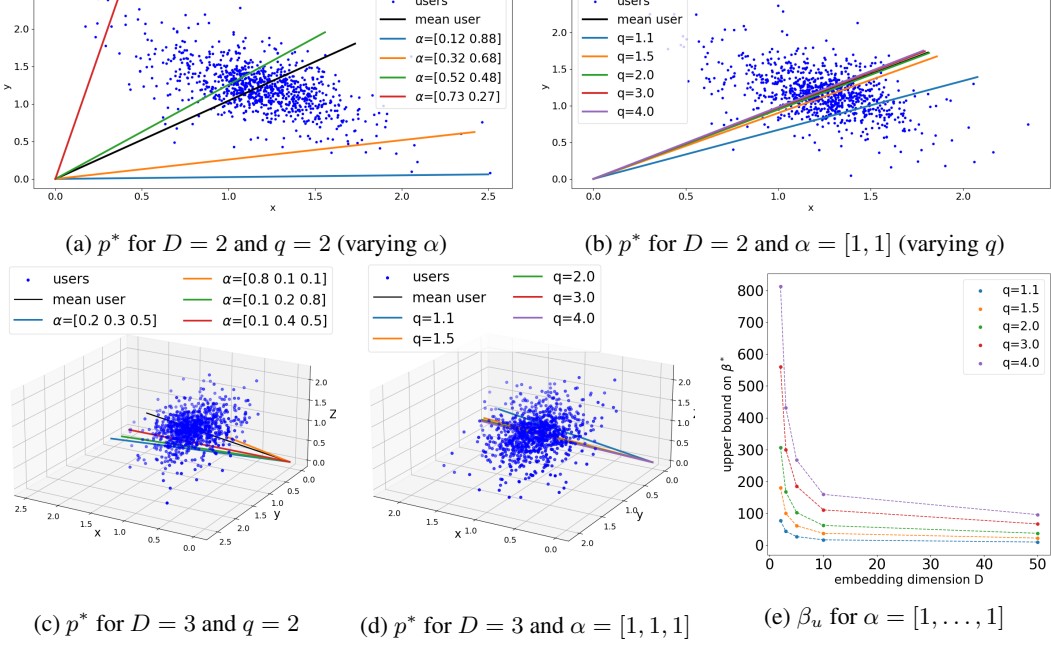

(a) $p^*$ for $D = 2$ and $q = 2$ (varying $\alpha$)      (b) $p^*$ for $D = 2$ and $\alpha = [1, 1]$ (varying $q$)

(c) $p^*$ for $D = 3$ and $q = 2$     (d) $p^*$ for $D = 3$ and $\alpha = [1, 1, 1]$     (e) $\beta_u$ for $\alpha = [1, \ldots, 1]$

Figure 4: Empirical analysis of supply-side equilibria on the MovieLens-100K dataset. *Plots (a)-(d):* Single-genre equilibrium direction $p^*$ (computed using Corollary 5) for different cost function weights $\alpha \in \mathbb{R}_{\geq 0}^D$ and parameters $q \geq 1$ as well as for different dimensions $D \geq 1$. Interestingly, the single-genre equilibrium direction is generally not aligned with the arithmetic mean and places a higher weight on cheaper dimensions. *Plot (e):* Upper bound $\beta_u$ on the threshold $\beta^*$ where specialization starts to occur (computed using Corollary 4) for different values of $q$. Observe that higher values of $D$ make specialization more likely to occur.

## 4 Impact of specialization on market competitiveness

Having studied the phenomenon of specialization, we next study its economic consequences on the resulting marketplace of digital goods. We show that producers can achieve *positive profit at equilibrium*, even though producers are competing with each other.

More formally, we can quantify the producer profit at equilibrium as follows. At a symmetric equilibria $\mu$, all producers receive the same expected profit given by:

$$\mathcal{P}^{\text{eq}}(\mu) := \mathbb{E}_{p_1,\ldots,p_P \sim \mu}[\mathcal{P}(p_1; p_{-1})] = \mathbb{E}\Big[\Big(\sum_{i=1}^N \mathbb{1}[j^*(u_i; \; p_{1:P}) = 1]\Big) - \|p_1\|^\beta\Big], \qquad (7)$$

where expectation in the last term is taken over $p_1, \ldots, p_P \sim \mu$ as well as randomness in recommendations. Intuitively, the equilibrium profit of a marketplace provides insight about market competitiveness. Zero profit suggests that competition has driven producers to expend their full cost budget on improving product quality. Positive profit, on the other hand, suggests that the market is not yet fully saturated and new producers have incentive to enter the marketplace.

We show a sufficient condition for positive profit in terms of the user geometry and the cost function, and we show this result relies on the equilibrium exhibiting specialization (Proposition 2).

To gain intuition, let us revisit two users located at the standard basis vectors and cost function $c(p) = \|p\|_2^\beta$. We can obtain the following characterization of profit.

**Corollary 6.** *Let there be 2 users located at the standard basis vectors $e_1, e_2 \in \mathbb{R}^2$, let the cost function be $c(p) = \|p\|_2^\beta$. For $P = 2$ and $\beta \geq \beta^* = 2$, there is an equilibrium $\mu$ with $\mathcal{P}^{eq}(\mu) = 1 - \frac{2}{\beta}$.*

Corollary 6 shows that there exist equilibria that exhibit strictly positive profit for any $\beta \geq 2$. The intuition is that (after sampling the randomness in $\mu$), different producers often produce different genres of content. This reduces the amount of competition along any single genre. Producers are thus no longer forced to maximize quality, enabling them to generate a strictly positive profit.

We generalize this finding to sets of many users and producers and to arbitrary norms. In particular, we provide the following sufficient condition under which the profit at equilibrium is strictly positive.

**Proposition 2.** *Suppose that*

$$\max_{\|p\|\leq 1} \min_{1\leq i\leq N} \left\langle p, \frac{u_i}{\|u_i\|} \right\rangle < N^{-P/\beta}. \tag{8}$$

*Then for any symmetric equilibrium $\mu$, the profit $\mathcal{P}^{eq}(\mu)$ is strictly positive.*

Proposition 2 provides insight into how the geometry of the users and structure of producer costs impact whether producers can achieve positive profit. To interpret Proposition 2, let us examine the quantity $Q := \max_{\|p\|\leq 1} \min_{i=1}^N \langle p, \frac{u_i}{\|u_i\|} \rangle$ that appears on the left-hand side of (8). Intuitively, $Q$ captures how easy it is to produce content that appeals simultaneously to all users. It is larger when the users are close together and smaller when they are spread out. For any set of vectors we see that $Q \leq 1$, with strict inequality if the set of vectors is non-degenerate. The right-hand side of (8), on the other hand, goes to 1 as $\beta \to \infty$. Thus, for any non-degenerate set of users, if $\beta$ is sufficiently large, the condition in Proposition 2 is met and producer profit is strictly positive.

Although Proposition 2 does not explicitly consider specialization, we show that specialization is nonetheless central to achieving positive profit at equilibrium. To illustrate this, we show that at a single-genre equilibrium, the profit is zero whenever there are at least $P \geq 2$ producers.

**Proposition 3.** *If $\mu$ is a single-genre equilibrium, then the profit $\mathcal{P}^{eq}(\mu)$ is equal to $0$.*

This draws a distinction between profit in the single-genre regime (where there is no specialization) and the multi-genre regime (where there is specialization). Interestingly, this distinction parallels between the distinction markets with homogeneous goods and markets with differentiated goods (as we describe in more detail in Appendix C.2).

Our results provide insight about the number of producers needed for a market to be saturated and fully competitive. Theorem 2 reveals that the marketplace of digital goods may need far more than 2 producers in order to be saturated. Nonetheless, the equilibrium profit does approach 0 as the number of producers in the marketplace goes to $\infty$: this is because the cumulative profit of all producers is at most $N$ and producers achieve the same profit, so $\mathcal{P}^{eq}(\mu) \leq N/P$. Perfect competition is therefore recovered in the infinite-producer limit.

## 5 Discussion

We presented a model for supply-side competition in recommender systems. The rich structure of production costs and the heterogeneity of users enable us to capture marketplaces that exhibit a wide range of forms of specialization. Our main results characterize when specialization occurs, analyze the form of specialization, and show that specialization can reduce market competitiveness. More broadly, we hope that our work serves as a starting point to investigate how recommendations shape the supply-side market of digital goods.

One direction for future work is to further examine the economic consequences of specialization. Several of our results take a step towards this goal: Corollary 5 illustrates that single-genre equilibria occur at the direction that maximizes the Nash user welfare, and Proposition 2 shows that specialization can lead to positive producer profit. These results leave open the question of how the welfare of users and producers relate to one another. Characterizing the welfare at equilibrium would elucidate whether specialization helps producers at the expense of users or helps all market participants.

Another interesting direction for future work is to extend our model to incorporate additional aspects of content recommender systems. For example, although we focus on perfect recommendations that match each user to their favorite content, we envision that this assumption could be relaxed in several ways: e.g., the platform may have imperfect information about users, users may not always follow platform recommendations, and producers may learn their best-responses over repeated interactions with the platform. Moreover, although we assume that producers earn fixed per-user revenue, this assumption could be relaxed to let producers set prices.

Addressing these questions would further elucidate the market effects induced by supply-side competition, and inform our understanding of the societal effects of recommender systems.

# 6 Acknowledgments

We would like to thank Jean-Stanislas Denain, Frances Ding, Erik Jones, Quitzé Valenzuela-Stookey, and Ruiqi Zhong for helpful comments on the paper. MJ acknowledges support from the Paul and Daisy Soros Fellowship and Open Phil AI Fellowship.

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
