# A Details of the empirical setup in Section 3.4

The code can be found at https://github.com/mjagadeesan/supply-side-equilibria.

**Dataset information.** We use the MovieLens-100K dataset which consists of $N = 943$ users, 1682 movies, and 100,000 ratings [Harper and Konstan, 2015]. We imported the dataset using the `scikit-surprise` library.

**Calculation of user embeddings.** For $D \in \{2, 3, 5, 10, 50\}$, we obtain $D$-dimensional user embeddings by running NMF (with $D$ factors). In particular, we ran NMF using the `scikit-surprise` library on the full MovieLens-100K dataset with the default hyperparameters.

**Calculation of single-genre equilibrium $p^*$.** We calculate the single-genre equilibrium genre $p^* = \arg\max_{\|p\|=1 \mid p \in \mathbb{R}_{\geq 0}^D} \sum_{i=1}^N \log(\langle p, u_i \rangle)$. We write $p^*$ as

$$p^* = \arg\max_{\|p\| \leq 1 \mid p \in \mathbb{R}_{\geq 0}^D} \sum_{i=1}^N \log(\langle p, u_i \rangle)$$

and solve the resulting optimization program. For $q = 2$, we directly use the `cvxpy` library with the default hyperparameters. For $q = 2$, we run projected gradient descent for $q \neq 2$ with learning rate 1.0 for 100 iterations where $p$ is initialized as a standard normal clamped so all the coordinates are at least 1. The projection step onto $\|p\| \leq 1 \mid p \in \mathbb{R}_{\geq 0}^D$ uses the `cvxpy` library with the default hyperparameters.

**Calculation of $\beta_u$.** We directly calculate $\beta_u$ according to the following formula:

$$\frac{\log(N)}{\log(N) - \log\left(\| \sum_{n=1}^N \frac{u_n}{\|u_n\|_*} \|_*\right)}.$$

# B Further details on the model

## B.1 Model discussion

Our model is one of the simplest possible that studies specialization in the supply-side marketplace. In particular, although many classical models[4] (e.g. spatial location models with specific user distributions and costs based on the Euclidean distance) permit closed-form equilibria, they elide important aspects of supply-side markets—such as the multi-dimensionality of producer decisions, the joint selection of genre and quality, and the structure of producer costs—which significantly influence the form that specialization takes. Our model incorporates these aspects at the cost of not having general closed-form equilibria; we nonetheless develop technical tools to study specialization without relying on closed-form solutions (while also obtaining closed forms in special cases). On the other side of the spectrum, we do not aim to provide a fully general model of product selection, production, and pricing. Instead, our model adds assumptions specific to recommender systems that provide sufficient structure to derive precise properties of specialization.

Our formalization of user preferences and the producer decision space is motivated by distinguishing aspects of content recommender systems. First, the infinite, high-dimensional content embedding space captures that digital goods can't be cleanly clustered into categories, but rather, are often mixtures of different dimensions (e.g. a movie can be both a drama and a comedy). Furthermore, the bilinear form (dot product) user values is motivated by standard recommendation algorithms: for example, matrix completion assumes that the user values are inner products between preference vectors and content attributes vectors [Koren et al., 2009].

Our assumptions on the structure of producer costs allow us to study specialization, while retaining mathematical tractability. The family of producer cost functions is stylized, but flexible, in that it accommodates arbitrary powers of arbitrary norms and it can capture both specialization and

---

[4]See Anderson et al. [1992] for a textbook treatment.

homogenization (Theorem 1). The assumption that all producers share the same cost function is also simplifying, but, potentially surprisingly, still allows us to study specialization. In particular, specialization occurs in a rich class of marketplaces (Corollary 4), *despite* the fact that producers have symmetric utility functions; we anticipate that the tendency towards specialization would only be amplified if producers could have different cost functions.

We hope that the simplicity of our model, and its ability to capture specialization, make it a useful starting point to further study the impact of recommender systems on production; we highlight some potential directions in Section 5.

## B.2 Equilibrium existence results

We first show that due to discontinuities in the utility function, pure strategy equilibria do not exist (Proposition 4 in Appendix B.2 Recall that $\mu_{1:P}$ is a *pure strategy equilibrium* if each $\mu_j$ contains only one vector in its support; it is a *mixed strategy equilibrium*.

**Proposition 4.** *For any set of users and any $\beta \geq 1$, a pure strategy equilibrium does not exist.*

The intuition is that if two producers are tied, then a producer can increase their utility by infinitesimally increasing the magnitude of their content.

Since pure strategy equilibria do not exist, we must turn to mixed strategy equilibria. Using the technology of equilibria in discontinuous games [Reny, 1999], we show that a mixed strategy equilibrium exists. In fact, because of the symmetries in the producer utility functions, we can actually show that a *symmetric* mixed strategy equilibrium (i.e. an equilibrium where $\mu_1 = \ldots = \mu_P$) exists.

**Proposition 1.** *For any set of users and any $\beta \geq 1$, a symmetric mixed equilibrium exists.*

Interestingly, symmetric mixed equilibria must exhibit significant randomness across different content embeddings. (Note that every symmetric equilibrium must exhibits some randomization, since pure strategy equilibria do not exist.) In particular, we show that a symmetric mixed equilibrium cannot contain point masses.

**Proposition 5.** *For any set of users and any $\beta \geq 1$, every symmetric mixed equilibrium is atomless.*

Proposition 5 implies that a symmetric mixed equilibrium has *infinite support*. The randomness can come from randomness over *quality* $\|p\|$ as well as randomness over *genres* $p/\|p\|$.

## B.3 Warmup: Homogeneous Users

To gain intuition for the structure of $\mu$, let's focus on a simple one-dimensional setting with one user. We show that the equilibria take the following form (see Figure 5):

**Example 1** (1-dimensional setup). *Let $D = 1$, and suppose that there is a single user $u_1 = 1$. Suppose the cost function is $c(p) = |p|^\beta$. The unique symmetric mixed equilibrium $\mu$ is supported on the full interval $[0, 1]$ and has cumulative distribution function $F(p) = (p/N)^{\beta/(P-1)}$. We defer the derivation to Appendix E.4.*

Since $D = 1$ in Example 1, content is specified by a single value $p \in \mathbb{R}^{\geq 0}$. Since the user will be assigned to the content with the highest value of $p$, we can interpret $p$ as the *quality* of the content. For a producer, setting $p$ to be larger increases the likelihood of being assigned to users, at the expense of a greater cost of production.

The equilibrium changes substantially with the parameters $\beta$ and $P$. First, for any fixed $P$, the equilibrium distribution for higher values of $\beta$ stochastically dominates the equilibrium distribution for lower values of $\beta$ (see Figure 5). The intuition is that increasing $\beta$ lowers production costs for content with a given quality, so producers must produce higher quality content at equilibrium. Similarly, for any fixed value of $\beta$, the equilibrium distribution for lower values of $P$ stochastically dominates the equilibrium distribution for higher values of $P$. This is because when more producers enter the market, any given producer is less likely to win users (i.e. a producer only wins a user with probability $1/P$ if all producers choose the same vector), so they cannot expend as high of a production cost.

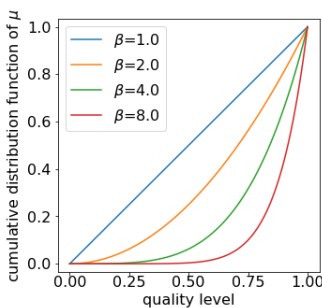

Figure 5: Cumulative distribution function (cdf) of the symmetric equilibrium $\mu$ for 1-dimensional setup (Example 1) with $P = 2$ producers. The equilibrium $\mu$ interpolates from a uniform distribution to a point mass as the exponent $\beta$ increases.

We next translate these insights about the equilibria for one-dimensional marketplaces to higher-dimensional marketplaces with a population of *homogeneous* users. If all users are embedded at the same vector $u \in \mathbb{R}_{\geq 0}^{D}$, then the producer's decision about what direction of content to choose is trivial: they would choose a direction in $\arg\max_{\|p\|=1}\langle p, u\rangle$. As a result, the producer's decision again boils down to a one-dimensional decision: choosing the quality $\|p\|$ of the content.

**Corollary 7.** *Suppose that there is a single population of $N$ users, all of whose embeddings are at the same vector $u$. Then, there is a symmetric mixed Nash equilibrium $\mu$ supported on $\left\{qp^* \mid q \in [0, N^{\frac{1}{\beta}}]\right\}$ where $p^* \in \arg\max_{\|p\|=1}\langle p, u\rangle$. The cumulative distribution function of $q = \|p\| \sim \mu$ is $F(q) = (q/N)^{\beta/(P-1)}$.*

Corollary 7 relies on the fact that when users are homogeneous, there is no tension between catering to one user and catering to other users.

## C Further discussion of our results

### C.1 Technical tools

En route to proving our results, we develop technical tools to analyze the complex, multi-dimensional behavior of producers. We highlight two tools here which may be of broader interest.

- To analyze when specialization occurs, we draw a connection to minimax theory in optimization. In particular, we show that the existence of a single-genre equilibrium is equivalent to strong duality holding for a certain optimization program that we define (Lemma 2). This allows us to leverage techniques from optimization theory to provide a necessary and sufficient condition for genre formation (Theorem 1).

- To analyze the properties of equilibria in concrete instances, we provide a decoupling lemma in terms of the equilibrium's support and its one-dimensional marginals (Lemma 1). This produces one-dimensional functional equations that make solving for the underlying equilibrium more tractable. We apply this decoupling lemma to analyze the form of specialization in the concrete setting of two equally sized populations of users with cost function $c(p) = \|p\|_2^{\beta}$.

Other technical ideas underlying our results include formalizing the formation of genres—which intuitively captures heterogeneity across producers—in terms of the support of a symmetric equilibrium distribution and applying the technology of discontinuous games Reny [1999] to establish the existence of symmetric mixed equilibria.

### C.2 Connection to markets with homogeneous and heterogeneous goods

The distinction between equilibrium profit in the single- and multi-genre equilibria parallels the classical distinctions in economics between markets with homogeneous goods and markets with differentiated goods (see [Baye and Kovenock, 2008] for a textbook treatment).

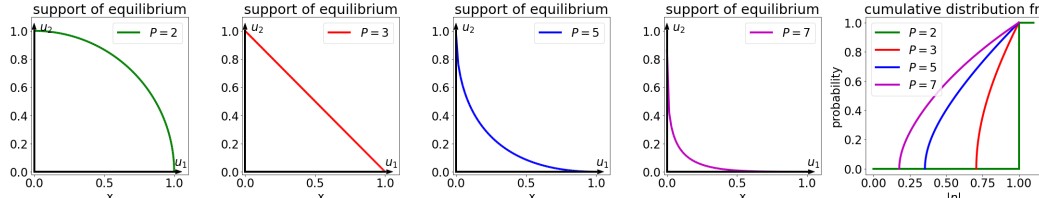

Figure 6: A symmetric equilibrium for different number of producers $P$, for 2 users located at the standard basis vectors $e_1$ and $e_2$, for producer cost function $c(p) = \|p\|_2^\beta$ with $\beta = 2$ (see Proposition 8). The first 4 plots show the support of an equilibrium $\mu$. As $P$ increases, the support goes from concave, to a line segment, to convex. The last plot shows the cumulative distribution function of $\|p\|$ for $p \sim \mu$. The distribution for lower $P$ stochastically dominates that of higher values of $P$.

Single-genre equilibria resemble markets with homogeneous goods where firms compete on price. If a firm sets their price above the zero profit level, they can be undercut by other firms and lose their users. The possibility of undercutting drives the profit to zero at equilibrium. Similarly, in the market that we study, when there is no specialization, producers all compete along the same direction, which drives profit to zero. The analogy is not exact: in our model, producers play a distribution of quality and thus might be out-competed in a given realization.

Multi-genre equilibria resemble markets with differentiated goods. In these markets, product differentiation reduces competition between firms, since firms compete for different users. This leads to local monopolies where firms can set prices above the zero profit level. Similarly, in the market that we study, specialization by producers leads to product differentiation and thus induces monopolistic behavior where the profit is positive. More specifically, specialization limits competition within each genre and can enable producers to set the quality of their goods below the zero profit level.

Our results formalize how the supply-side market of a recommender system can resemble a market with homogeneous goods or a market with differentiated goods, depending on whether specialization occurs. An empirical analysis could quantify where on this spectrum a given recommender system is located, and regulatory policy could seek to shift a recommender system towards one of the regimes.

## D   Equilibrium structure for two equally sized populations of users

We next investigate the form of specialization exhibited by multi-genre equilibria, focusing on the case of two equally sized populations and producer cost functions given by powers of the $\ell_2$ norm. More formally, there are $N$ users split equally between two linearly independently vectors $u_1, u_2 \in \mathbb{R}_{\geq 0}^D$, and the the cost function is $c(p) = \|p\|_2^\beta$. We establish *structural properties* of the equilibria (see Section D.1). We concretely compute the equilibria $\mu$ in several special instances that permit *closed-form solutions* (see Section D.2-D.3). We then provide an overview of proof techniques, which involves developing machinery to characterize these equilibria (see Section D.4).

### D.1   Structural properties of equilibria

We first establish properties about the *support* of the equilibrium distributions $\mu$. First, we show that the support of cannot contain an $\epsilon$-ball for any $\epsilon$ and is thus 1-dimensional.

**Proposition 6.** *Suppose that there are $N$ users split equally between two linearly independently vectors $u_1, u_2 \in \mathbb{R}_{\geq 0}^2$, and let $\theta^* := \cos^{-1}\left(\frac{\langle u_1, u_2 \rangle}{\|u_1\|_2 \|u_2\|}\right)$ be the angle between the user vectors. Let the cost function be $c(p) = \|p\|_2^\beta$, and let $P \geq 2$. Let $\mu$ be a symmetric Nash equilibrium such that the distributions $\langle u_1, p \rangle$ and $\langle u_2, p \rangle$ over $\mathbb{R}_{\geq 0}$ are absolutely continuous. As long as $\beta \neq 2$ or $\theta^* \neq \pi/2$, the support of $\mu$ does not contain an $\ell_2$-ball of radius $\epsilon$ for any $\epsilon > 0$.*[5]

Proposition 6 demonstrates that the support of $\mu$ must be a union of 1-dimensional curves. In the single-genre regime, the support is always a line segment through the origin. In the multi-genre regime, however, the support can be curves with different shapes (see Figure 6 for specific examples).

---

[5]The case of $\beta = 2$ and $\theta^* = \pi/2$ is degenerate and permits a range of possible equilibria.

We will later characterize where these curves are increasing or decreasing in terms of the location of the curve, the angle $\theta^* = \cos^{-1}\left(\frac{\langle u_1, u_2 \rangle}{\|u_1\|\|u_2\|}\right)$, and the cost function parameter $\beta$ (Lemma 12).

We next show that all equilibria must have either one or infinitely many genres, dictated by whether $\beta$ is above or below the critical value $\beta^*$ (see Figure 1):

**Theorem 2.** *Suppose that there are $N$ users split equally between two linearly independently vectors $u_1, u_2 \in \mathbb{R}_{\geq 0}^D$, and let $\theta^* := \cos^{-1}\left(\frac{\langle u_1, u_2 \rangle}{\|u_1\|_2\|u_2\|}\right)$ be the angle between the user vectors. Let the cost function be $c(p) = \|p\|_2^\beta$. Let $\mu$ be a a distribution on $\mathbb{R}^d$ such that the distributions $\langle u_1, p \rangle$ and $\langle u_2, p \rangle$ over $\mathbb{R}_{\geq 0}$ over $\mathbb{R}_{\geq 0}$ for $p \sim \mu$ are absolutely continuous and twice continuously differentiable within their supports. There are two regimes based on $\beta$ and $\theta^*$:*

1. *If $\beta < \beta^* = \frac{2}{1-\cos(\theta^*)}$ and if $\mu$ is a symmetric mixed equilibrium, then $\mu$ satisfies $|\mathrm{Genre}(\mu)| = 1$.*

2. *If $\beta > \beta^* = \frac{2}{1-\cos(\theta^*)}$, if $|\mathrm{Genre}(\mu)| < \infty$, and if the conditional distribution of $\|p\|$ along each genre is continuously differentiable, then $\mu$ is not an equilibrium.*

Theorem 2 provides a tight characterization of when specialization occurs in a marketplace: specialization occurs *if and only if* $\beta$ is above $\beta^*$ (subject to some mild continuity conditions). The threshold $\beta^*$ can thus be interpreted as a *phase transition* at which the equilibrium transitions from single-genre to infinitely many genres (see Figure 1). More specifically, the first part of Theorem 2 strengthens Theorem 1 to show that *all* equilibria are single-genre when $\beta < \beta^*$, which means that producers are *never* incentivized to specialize in this regime. The equality condition $\beta = \beta^*$ captures the transition point where both single-genre and multi-genre equilibria can exist.

In the multi-genre regime where $\beta \geq \beta^*$, Theorem 2 shows that producers do not fully personalize content to either of the two users $u_1$ and $u_2$, or even choose between finitely many types of content. Rather, producers choose infinitely many types of content that balance the preferences of the two populations in different ways. The lack of coordination between producers—as captured by a symmetric mixed Nash equilibrium—is what drives this result. Producers do not know exactly what content other producers will create in a given realization of the randomness, which results in a diversity of content on the platform.

### D.2 Closed-form equilibria for the standard basis vectors

We next compute the equilibria in the special case of user vectors located at the standard basis vectors, and we analyze the form of specialization that the equilibria exhibit. For ease of notation, for the remainder of the section, we assume these populations each consist of a *single* user (these results can be easily adapted to the case of $N/2$ users in each population).

Interestingly, all of these multi-genre equilibria exhibit the following *relaxation of pure horizontal differentiation*: producers can differentiate along genre, but the genre of content fully specifies the content's quality. More specifically, for any genre $p^* \in \mathrm{Genre}(\mu)$, the set $\mathrm{Genre}(\mu) \cap \{q \cdot p^* \mid q \in \mathbb{R}^{\geq 0}\}$ contains exactly one single element.[6] This stands in contrast to single-genre equilibria, which by definition exhibit pure vertical differentiation.[7]

We first explicitly compute the equilibria in the case of $P = 2$ producers (see Figure 1).

**Proposition 7.** *Suppose that there are 2 users located at the standard basis vectors $e_1, e_2 \in \mathbb{R}^2$, and the cost function is $c(p) = \|p\|_2^\beta$. For $P = 2$ and $\beta \geq \beta^* = 2$, there is an equilibrium $\mu$ supported on the quarter-circle of radius $(2\beta^{-1})^{1/\beta}$, where the angle $\theta \in [0, \pi/2]$ has density $f(\theta) = 2\cos(\theta)\sin(\theta)$.*

Proposition 7 demonstrates the support of the equilibrium distribution is a quarter circle with radius $(2\beta^{-1})^{1/\beta}$. This equilibrium exhibits pure horizontal differentation (as well as the relaxation of pure horizontal differentiation that we described above). Since all $(x, y)$ in the support have the same

---

[6]Pure horizontal differentiation is not satisfied, since content in different genres may not have the same quality (see Figure 6).

[7]Pure vertical differentiation is when producers only differentiate along quality, not along direction.

radius, producers always expend the same cost regardless of the realization of randomness in their strategy. Since $c(p) = \|p\|_2^\beta$, producers pay a cost of $2\beta^{-1}$. The cost of production therefore goes to $0$ as $\beta \to \infty$. This enables producers achieving *positive profit* at equilibrium (see Corollary 6) as we describe in more detail in Section 4.

We next vary the number of producers $P$ while fixing $\beta = 2$ (see Figure 6).

**Proposition 8.** *Suppose that there are 2 users located at the standard basis vectors $e_1, e_2 \in \mathbb{R}^2$, with cost function $c(p) = \|p\|_2^\beta$. For $\beta = 2$, there is a multi-genre equilibrium $\mu$ with support equal to*

$$\left\{ \left( x, (1 - x^{\frac{2}{P-1}})^{\frac{P-1}{2}} \right) \mid x \in [0, 1] \right\}, \tag{9}$$

*and where the distribution of $x$ has cdf equal to $\min(1, x^{2/(P-1)})$.*

Proposition 8 demonstrates that for different values of $P$, the support of the equilibrium $\mu$ follows different curves connecting $[1, 0]$ and $[0, 1]$. Note that these equilibria exhibit the relaxation of pure horizontal differentiation that we described earlier. Moreover, the curve is concave for $P = 2$, a line segment for $P = 3$, and convex for all $P \geq 4$. Indeed, as $P$ increases, the support converges to the union of the two coordinate axes.

### D.3 Closed-form equilibria in an infinite-producer limit

Motivated by the support collapsing onto the standard basis vectors for $P \to \infty$ in Proposition 8, we investigate equilibria in a "limiting marketplace" where $P \to \infty$. In the infinite-producer limit, we show that a *two-genre* equilibrium exists, regardless of the geometry of the 2 user vectors, and we characterize the equilibrium distribution $\mu$ (see Figure 3). Interestingly, these equilibria do not exhibit pure vertical differentiation or (the relaxation of) pure horizontal differentiation.

Formalizing the infinite-producer limit is subtle: the distribution of any single producer approaches a point mass at $0$, but the distribution of the *winning* producer turns out to be non-degenerate. To get intuition for this, let's revisit the one-dimensional setup of Example 1. The cumulative distribution function $F(p) = (p/N)^{\beta/(P-1)}$ of a single producer as $P \to \infty$ approaches $F(p) = 1$ for any $p > 0$—this corresponds to a point mass at $0$.[8] On the other hand, the cumulative distribution function of the *winning* producer $F^{\max}(p) = (p/N)^{\beta P/(P-1)}$ approaches $(p/N)^\beta$, which is a well-defined function.

When we formalize the infinite-producer limit for $N \geq 1$ users, we leverage the intuition that the distribution function of the winning producer is non-degenerate. In particular, we specify infinite-producer equilibria in terms of three properties—the *genres*, the *conditional quality distributions for each genre* (i.e. the distribution of the maximum quality $\|p\|$ along a genre, conditional on all of the producers choosing that genre), and the *weights* (i.e. the probability that a producer chooses each genre). We defer a formal treatment to Definition 1 in Section G.5.

In the infinite-producer limit, we show the following 2-genre distribution is an equilibrium. For ease of notation, we again assume these populations each consist of a *single* user (these results can be easily adapted to the case of $N/2$ users in each population).

**Theorem 3.** *[Informal version of Theorem 4] Suppose that there are 2 users located at two linearly independently vectors $u_1, u_2 \in \mathbb{R}_{\geq 0}^D$, let $\theta^* := \cos^{-1}\left(\frac{\langle u_1, u_2 \rangle}{\|u_1\|_2 \|u_2\|}\right)$ be the angle between them. Suppose we have cost function $c(p) = \|p\|_2^\beta$, $\beta > \beta^* = \frac{2}{1 - \cos(\theta^*)}$, and $P = \infty$ producers. Then, there exists an equilibrium with two genres:*

$$\left\{ [\cos(\theta^G + \theta_{min}), \sin(\theta^G + \theta_{min})], [\cos(\theta^* - \theta^G + \theta_{min}), \sin(\theta^* - \theta^G + \theta_{min})] \right\}$$

*where $\theta^G := \arg\max_{\theta \leq \theta^*/2} \left( \cos^\beta(\theta) + \cos^\beta(\theta^* - \theta) \right)$ and $\theta_{min} := \min\left( \cos^{-1}\left(\frac{\langle u_1, e_1 \rangle}{\|u_1\|}\right), \cos^{-1}\left(\frac{\langle u_2, e_1 \rangle}{\|u_2\|}\right) \right)$.*

*For each genre, the conditional quality distribution (i.e. the distribution of the maximum quality $\|p\|$ along a genre, conditional on all of the producers choosing that genre) has cdf given by a*

---

[8]The intuition is that the expected number of users that the producer wins at a symmetric equilibrium is $N/P$, which approaches $0$ in the limit; thus, the production cost that a producer can afford to expend must approach $0$ in the limit.

*countably-infinite piecewise function, where each piece is either constant or grows proportionally to* $\|p\|^{2\beta}$.

Theorem 3 reveals that finite-genre equilibria (that have more than one genre) re-emerge in the limit as $P \to \infty$, although they do not exist for any finite $P$ (see Figure 3). For users located at the standard basis vectors, Theorem 3 formalizes the intuition from Proposition 8 that the equilibrium converges to a distribution supported on the standard basis vectors. This means that at $P = \infty$, producers either entirely personalize their content to the first user or entirely personalize their content to the second user, but do not try to appeal to both users at the same time.

Interestingly, the set of genres is *not* equal to the set of two users unless users are orthogonal. As shown in Figure 3, the two-genres are located within the interior of the convex cone formed by the two users. This means that producers always attempt to cater their content to both users at the same time, although they either place a greater weight on one user or the other user, depending on which genre they choose. The location of these two genres changes for different values of $\beta$. When $\beta$ approaches the single-genre threshold, the genres both collapse onto the single-genre direction $\theta^*/2$. On the other hand, when $\beta$ approaches $\infty$, the genres converge to the two users.

Finally, the support of the equilibrium distribution consists of countably infinite disjoint line segments with interesting economic interpretations. First, observe that the cdf of the conditional quality distributions of each genre (see the last panel of Figure 3) has gaps in its support: it is a countably-infinite piecewise function, where each piece is either constant or grows proportionally to $q^{2\beta}$. The level of "bumpiness" of the cdf decreases as $\theta^*$ increases: for the limiting case of $\theta = \pi/2$, it converges to the smooth function $F^{\max}(q) = q^{2\beta}$. Moreover, the regions of zero density of each of the two genres are actually staggered, so that at most one of the genres can achieves a given utility for a given user. In particular, for each user $u_i$, it never holds that $\langle u_i, p \rangle = \langle u_i, p' \rangle$ for $p \neq p'$: that is, the utility level fully specifies the genre of the content. The closed-form expression of the density (see Theorem 4) formally establishes these properties.

## D.4 Overview of proof techniques

To prove our results in this section, our first step is establish a useful characterization of equilibria that enables us to separately account for the geometry of the users and the number of producers. This takes the form of necessary and sufficient conditions that decouple in terms of two quantities: a set of *marginal distributions* $H_i$, and the *support* $S \subseteq \mathbb{R}_{\geq 0}^N$.

**Lemma 1.** *Let* $\mathbf{U} = [u_1; u_2; \dots; u_N]$ *be the* $N \times D$ *matrix of users vectors. Given a set* $S \subseteq \mathbb{R}_{\geq 0}^N$ *and distributions* $H_1, \dots, H_N$ *over* $\mathbb{R}_{\geq 0}$, *suppose that the following conditions hold:*

*(C1) Every* $z^* \in S$ *is a maximizer of the equation:*

$$\max_{z \in \mathbb{R}_{\geq 0}^D} \sum_{i=1}^N H_i(z_i) - c_{\mathbf{U}}(z), \tag{10}$$

*where* $c_{\mathbf{U}}(z) := \min \left\{ c(p) \mid p \in \mathbb{R}_{\geq 0}^D, \mathbf{U}p = z \right\}$.

*(C2) There exists a random variable* $Z$ *with support* $S$, *such that the marginal distribution* $Z_i$ *has cdf equal to* $H_i(z)^{1/(P-1)}$.

*(C3)* $Z$ *is distributed as* $\mathbf{U}Y$ *with* $Y \sim \mu$, *for some distribution* $\mu$ *over* $\mathbb{R}_{\geq 0}^D$.

*Then, the distribution* $\mu$ *from (C3) is a symmetric mixed Nash equilibrium. Moreover, every symmetric mixed Nash equilibrium* $\mu$ *is associated with some* $(H_1, \dots, H_N, S)$ *that satisfy (C1)-(C3).*

In Lemma 1, the set $S$ captures the support of the realized user utilities $[\langle u_1, p \rangle, \dots, \langle u_N, p \rangle]$ for $p \sim \mu$. The distribution $H_i$ captures the distribution of the maximum utility $\max_{1 \leq j \leq P-1} \langle u_i, p_j \rangle$ for user $u_i$.

The conditions in Lemma 1 help us identify and analyze the equilibria in concrete instantiations, including in the 2 user vector setting that we focus on in this section.

- (C1) places conditions on $H_1$, $H_2$, and $S$ in terms of the induced cost function $c_{\mathbf{U}}$. We use the first-order and second-order conditions of equation (10) at $z = [z_1, z_2]$ to determine the necessary densities $h_1(z_1)$ and $h_2(z_2)$ of $H_1$ and $H_2$ for $z$ to be in the support $S$.

- (C2) restricts the relationship between $H_1$, $H_2$, and $S$ for a given value of $P$, which we instantiate in two different ways, depending on whether the support is a single curve or whether the distribution $\mu$ has finitely many genres.
- (C3) holds essentially without loss of generality when $u_1$ and $u_2$ are linearly independent.

The proofs of our results in this section boil down to leveraging these conditions.

# E   Proofs for Section 2

## E.1   Proof of Proposition 4

We restate and prove Proposition 4.

**Proposition 4.** *For any set of users and any $\beta \geq 1$, a pure strategy equilibrium does not exist.*

*Proof of Proposition 4.* Assume for sake of contradiction that the solution $p_1, \ldots, p_P$ is a pure strategy equilibrium. We divide into two cases based on whether there are ties. The cases are: (1) there exist $1 \leq j' \neq j \leq P$ and $i$ such that $\langle p_j, u_i \rangle = \langle p_{j'}, u_i \rangle$, (2) there does not exist $j, j'$ and $i$ such that $\langle p_j, u_i \rangle = \langle p_{j'}, u_i \rangle$.

**Case 1: there exist $1 \leq j' \neq j \leq P$ and $i$ such that $\langle p_j, u_i \rangle = \langle p_{j'}, u_i \rangle$.** Let producer $j$ and producer $j'$ be such that $\langle p_j, u_i \rangle = \langle p_{j'}, u_i \rangle$. The idea is that the producer $j$ can leverage the discontinuity in their profit function (1) at $p_j$. In particular, consider the vector $p_j + \epsilon u_i$. The number of users that they receive as $\epsilon \to_+ 0$ is *strictly greater* than at $p_j$. The cost, on the other hand, is continuous in $\epsilon$. This demonstrates that there exists $\epsilon > 0$ such that:

$$\mathcal{P}(p_j + \epsilon u_i; p_{-j}) > \mathcal{P}(p_j; p_{-j})$$

as desired. This is a contradiction.

**Case 2: there does not exist $j, j'$ and $i$ such that $\langle p_j, u_i \rangle = \langle p_{j'}, u_i \rangle$.** Since the sum of the expected number of users won by all of the producers is $N$, there exists a producer who wins a nonzero number of users in expectation. Let $j$ be such a producer. Using the assumption that there are no ties (i.e. there does not exist $j'$ and $i$ such that $\langle p_j, u_i \rangle = \langle p_{j'}, u_i \rangle$), we know that producer $j$ wins the following set of users:

$$\mathcal{N}_j := \{1 \leq i \leq N \mid \langle p_j, u_i \rangle > \langle p_{j'}, u_i \rangle \forall j' \neq j\}.$$

We see that $\mathcal{N}_j$ is nonempty by the assumption that producer $j$ wins a nonzero number of users in expectation. We now leverage that the profit function of producer $j$ is continuous at $p_j$. There exists $\epsilon > 0$ such that $\langle p_j(1 - \epsilon), u_i \rangle > \langle p_{j'}, u_i \rangle$ for all $j' \neq j$ and all $i \in \mathcal{N}_j$, so that:

$$\mathcal{P}(p_j(1 - \epsilon); p_{-j}) > \mathcal{P}(p_j; p_{-j})$$

as desired. This is a contradiction.

$\square$

## E.2   Proof of Proposition 1

We restate and prove Proposition 1.

**Proposition 1.** *For any set of users and any $\beta \geq 1$, a symmetric mixed equilibrium exists.*

*Proof of Proposition 1.* We apply a standard existence result of symmetric, mixed strategy equilibria in discontinuous games (see Corollary 5.3 of [Reny, 1999]). We adopt the terminology of that paper and refer the reader to [Reny, 1999] for a formal definition of the conditions. Note that the game is symmetric by assumption, since the producers have symmetric utility functions. It suffices to show that: (1) the producer action space is convex and compact and (2) the game is diagonally better-reply secure.

**Producer action space is convex and compact.** In the current game, the producer action space is not compact. However, we show that we can define a slightly modified game, where the producer action space is convex and compact, without changing the equilibrium of the game. For the remainder of the proof, we analyze this modified game.

In particular, each producer must receive at least 0 profit at equilibrium since $\mathcal{P}(\vec{0}; p_{-1}) \geq 0$ regardless of the actions $p_{-1}$ taken by other producers. If a producer chooses $p$ such that $\|p\| > N^{1/\beta}$, then their utility will be strictly negative. Thus, we can restrict to $\{p \in \mathbb{R}^D_{\geq 0} \mid \|p\| \leq 2N^{1/\beta}\}$ which is a convex compact set. We add a factor of 2 slack to guarantee that any best-response by a producer will be in the *interior* of the action space and not on the boundary.

**Establishing diagonal better reply security.** First, we show the payoff function $\mathcal{P}(\mu; [\mu, \ldots, \mu])$ (where $\mu$ is a distribution over the producer action space) is continuous in $\mu$. Here we slightly abuse notation since $\mathcal{P}$ is technically defined over pure strategies in (1). We implicitly extend the definition to mixed strategies by considering expected profit. Using the fact that each producer receives a $1/P$ fraction of users in expectation at a symmetric solution, we see that:

$$\mathcal{P}(\mu; [\mu, \ldots, \mu]) = \frac{N}{P} - \int \|p\|^\beta d\mu.$$

Since the underlying topology on the set of distributions $\mu$ is the weak* topology, this implies continuity of the payoff.

Now, we construct, for each relevant payoff in the closure of the graph of the game's diagonal payoff function, an action that diagonal payoff secures that payoff. More formally, let $(\mu^*, \alpha^*)$ be in the closure of the graph of the game's diagonal payoff function, and suppose that $(\mu^*, \ldots, \mu^*)$ is not an equilibrium. It suffices to show that a producer can secure a payoff of $\alpha > \alpha^*$ along the diagonal at $(\mu^*, \ldots, \mu^*)$. We construct $\mu^{\text{sec}}$ that secures a payoff of $\alpha > \alpha^*$ along the diagonal at $(\mu^*, \ldots, \mu^*)$.

Recall that $\alpha^* = \mathcal{P}(\mu^*, \ldots, \mu^*)$ by the continuity of the payoff function shown above. Since $(\mu^*, \ldots, \mu^*)$ is not an equilibrium, there exists $p \in \{p' \in \mathbb{R}^D_{\geq 0} \mid \|p'\| \leq N^{1/\beta}\}$ such that

$$\mathcal{P}(p; [\mu^*, \ldots, \mu^*]) > \mathcal{P}(\mu^*; [\mu^*, \ldots, \mu^*]) = \alpha^*.$$

Since we ultimately want to show that $p$ achieves high profit in an open neighborhood of $\mu^*$, we need to strengthen the above statement. We can achieve by this by appropriately perturbing $p$. First, we can perturb $p$ to $\tilde{p}$ such that for each $1 \leq i \leq N$, the distribution $\langle p', u_i \rangle$ where $p' \sim \mu^*$ does not have a point mass at $\langle \tilde{p}, u_i \rangle$, and such that:

$$\mathcal{P}(\tilde{p}; [\mu^*, \ldots, \mu^*]) = \sum_{i=1}^n \left(\mathbb{P}_{p' \sim \mu^*}[\langle \tilde{p}, u_i \rangle > \langle p', u_i \rangle]\right)^{P-1} - c(\tilde{p}) > \alpha^*.$$

Now, we further perturb $\tilde{p}$ to add $\epsilon$ slack to the constraint $\langle \tilde{p}, u_i \rangle > \langle p', u_i \rangle$. In particular, we observe that there exists $\epsilon > 0$ and $p^{\text{sec}} \in \mathbb{R}^D_{\geq 0}$ such that

$$\mathcal{P}(p^{\text{sec}}; [\mu^*, \ldots, \mu^*]) \geq \sum_{i=1}^n \left(\mathbb{P}_{p' \sim \mu^*}[\langle p^{\text{sec}}, u_i \rangle > \langle p', u_i \rangle + \epsilon \|u_i\|_2]\right)^{P-1} - c(p^{\text{sec}}) > \alpha^* + \epsilon. \quad (11)$$

We claim that $\mu^{\text{sec}}$ taken to be the point mass at $p^{\text{sec}}$ will secure a payoff of

$$\alpha = \frac{\mathcal{P}(p^{\text{sec}}; [\mu^*, \ldots, \mu^*]) + \alpha^*}{2} > \alpha^*$$

along the diagonal at $(\mu^*, \ldots, \mu^*)$. We define the event $A_i$ to be:

$$A_i = \{p' \mid \langle p^{\text{sec}}, u_i \rangle > \langle p', u_i \rangle\}.$$

In this notation, we can rewrite equation (11) as:

$$\mathcal{P}(p^{\text{sec}}; [\mu^*, \ldots, \mu^*]) \geq \sum_{i=1}^n (\mu^*(A_i^\epsilon))^{P-1} - c(p^{\text{sec}}) > \alpha^* + \epsilon.$$

Consider the metric on $\mathbb{R}_{\geq 0}^D$ given by the $\ell_2$ norm. For $\epsilon > 0$ let $B_\epsilon(\mu^*)$ denote the $\epsilon$-ball with respect to the Prohorov metric; using the definition of the weak* topology, we see that $B_\epsilon(\mu^*)$ is an open set with respect to the weak* topology. For each $1 \leq i \leq N$, we define the event $A_i$ as:

$$A_i^\epsilon = \{p' \mid \langle p^{\mathrm{sec}}, u_i \rangle > \langle p', u_i \rangle + \epsilon \|u_i\|_2\}$$

and define the event $A_i^\epsilon$ to be:

$$A_i = \{p' \mid \langle p^{\mathrm{sec}}, u_i \rangle > \langle p', u_i \rangle\}.$$

For every $p' \in A_i^\epsilon$, we see that $A_i$ contains the open neighborhood $B_\epsilon(p')$ with respect to the $\ell_2$ norm. By the definition of the Prohorov metric, we know that for all $\mu' \in B_\epsilon(\mu^*)$, it holds that

$$\mu'(A_i) \geq \mu^*(A_i^\epsilon) - \epsilon$$

This implies that

$$\mathcal{P}(p^{\mathrm{sec}}; [\mu', \ldots, \mu^*]) \geq \sum_{i=1}^n (\mu'(A_i))^{P-1} - c(p^{\mathrm{sec}}) \geq \underbrace{\sum_{i=1}^n (\mu^*(A_i^\epsilon) - \epsilon)^{P-1}}_{(A)} - c(p^{\mathrm{sec}}).$$

For each $1 \leq i \leq N$, let $A_i$ be the event that $\langle p^{\mathrm{sec}}, u_i \rangle > \langle p', u_i \rangle$. By the definition of the Prohorov metric, we see that $\mu'(A_i) \geq \mu^*(A_i) - \epsilon$. Moreover, it holds that:

$$\mathcal{P}(p^{\mathrm{sec}}; [\mu^*, \ldots, \mu^*]) = \sum_{i=1}^n (\mu^*(A_i))^{P-1} - c(p^{\mathrm{sec}}).$$

and moreover, for all $\mu' \in B_\epsilon(\mu^*)$, it holds that:

$$\mathcal{P}(p^{\mathrm{sec}}; [\mu', \ldots, \mu^*]) \geq \sum_{i=1}^n (\mu'(A_i))^{P-1} - c(p^{\mathrm{sec}}) \geq \underbrace{\sum_{i=1}^n (\mu^*(A_i) - \epsilon)^{P-1}}_{(A)} - c(p^{\mathrm{sec}}).$$

Using that (A) is continuous in $\epsilon$, we see that if $\epsilon$ is sufficiently small, then:

$$\mathcal{P}(p^{\mathrm{sec}}; [\mu', \ldots, \mu^*]) \geq \mathcal{P}(p^{\mathrm{sec}}; [\mu', \ldots, \mu^*]) - \frac{\mathcal{P}(p^{\mathrm{sec}}; [\mu^*, \ldots, \mu^*]) - \alpha^*}{3} > \alpha$$

for all $\mu' \in B_\epsilon(\mu^*)$, as desired. $\qquad\square$

### E.3 Proof of Proposition 5

In this proof, we consider the payoff function $\mathcal{P}(\mu_1; [\mu_2, \ldots, \mu_P])$ (where $\mu$ is a distribution over the producer action space) defined to be the expected profit attained if a producer plays $\mu_1$ when other producers play $\mu_2, \ldots, \mu_P$. Strictly speaking, this is an abuse of notation since $\mathcal{P}$ is technically defined over pure strategies in (1). We implicitly extend the definition to mixed strategies by considering *expected* profit.

*Proof of Proposition 5.* Let $\mu$ be a symmetric equilibrium, and assume for sake of contradiction that there is an atom at $p \in \mathbb{R}^d$ with probability mass $\alpha > 0$. It suffices to construct a vector $p'$ that achieves profit

$$\mathcal{P}(p'; [\mu, \ldots, \mu]) > \mathcal{P}(\vec{0}; [\mu, \ldots, \mu]) = \mathcal{P}(\mu; [\mu, \ldots, \mu]).$$

Consider the vector $p' = p + \epsilon u_1$ for some $\epsilon > 0$. For any given realization of actions by other producers, and for any given user, the vector $p'$ never wins the user with lower probability than the vector $p$. We construct an event and a user where the vector $p'$ wins the user with strictly higher probability than the vector $p$. Let $E$ be the event that all of the other producers choose the $p$ vector. This event happens with probability $\alpha^{P-1}$. Conditioned on $E$, the vector $p'$ wins user $u_1$; on the other hand, the vector $p$ wins user $u_1$ with probability $1/P$. Since the cost function is continuous in $\epsilon$, there exists $\epsilon$ such that $\mathcal{P}(p; [\mu, \ldots, \mu]) > \mathcal{P}(\vec{0}; [\mu, \ldots, \mu]) = \mathcal{P}(\mu; [\mu, \ldots, \mu])$. This is a contradiction.

$\qquad\square$

### E.4  Derivation of Example 1

To see that the cumulative distribution function is $F(p) = \min(1, p^{\beta/P-1})$, we use the fact that every equilibrium is by definition a single-genre equilibrium in 1 dimension and apply Lemma 3.

## F  Proofs for Section 3

In Section F.1, we prove Theorem 1, and in Section F.2, we prove the corollaries of Theorem 1 in Section 3 (with the exception of Corollary 3, whose proof we defer to Section G.3).

### F.1  Proof of Theorem 1

We begin with a proof sketch of Theorem 1. Since the single-genre equilibrium does not admit a straightforward closed-form solution, we must implicitly reason about its existence when proving Theorem 1. To do so, we draw a connection to minimax theory in optimization. Our main lemma shows that the existence of a single-genre equilibrium is equivalent to strong duality holding for the following minmax problem:

**Lemma 2** (Informal). *There exists a symmetric equilibrium $\mu$ with $|\mathrm{Genre}(\mu)| = 1$ if and only if:*

$$\inf_{y \in \mathcal{S}^\beta} \left( \sup_{y' \in \mathcal{S}^\beta} \sum_{i=1}^N \frac{y_i'}{y_i} \right) = \sup_{y' \in \mathcal{S}^\beta} \left( \inf_{y \in \mathcal{S}^\beta} \sum_{i=1}^N \frac{y_i'}{y_i} \right). \tag{12}$$

To prove Theorem 1 from Lemma 2, we analyze when strong duality holds. Note that while the objective in (12) is convex in $y$ and linear (concave) in $y'$, the constraints on $y$ and $y'$ through the set $\mathcal{S}^\beta$ can be non-convex. It turns out that we can eliminate the non-convexity in the constraint on $y$ for free, by reparameterizing to the space of content vectors $p \in \mathbb{R}_{\geq 0}^D$ with unit norm. On the other hand, to handle the non-convexity in the constraint on $y'$, we need to convexify the optimization program by replacing $\mathcal{S}^\beta$ with its convex hull $\bar{\mathcal{S}}^\beta$. By Sion's min-max theorem, we can flip sup and inf in this convexified version of the left-hand side of (12). The remaining technical step is to relate the resulting expression to the right-hand side of (12), which we defer to Appendix F.1.

To prove Lemma 2, we first characterize the cumulative distribution function of quality at a single-genre equilibria as $F(q) \propto q^\beta$ (Lemma 3). Then we show that $y$ corresponds to an equilibrium direction if and only if $\sup_{y' \in \mathcal{S}^\beta} \sum_{i=1}^N \frac{y_i'}{y_i} \leq N$, which means that there exists an equilibrium direction if and only if the left-hand side of (12) is at most $N$. We also show that the dual the right-hand side of (12) is always equal to $N$, which allows us to prove Lemma 2.

**A useful intermediate result.** Before diving into the proof of Lemma 2 and Theorem 1, we describe an intermediate result will be useful in the proof of Lemma 2. Suppose that there exists an equilibrium $\mu$ such that $\mathrm{Genre}(\mu) = \{p^*\}$ contains a single direction. Then $\mu$ is fully determined by the distribution over quality $\|p\|$ where $p \sim \mu$; therefore, let $F$ denote the cdf of $\|p\|$ for $p \sim \mu$. We can derive a closed-form expression for $F$; in fact, we show that it is identical to the cdf of the 1-dimensional setup in Example 1.

**Lemma 3.** *Suppose that $\mu$ is a symmetric equilibrium such that $\mathrm{Genre}(\mu)$ contains a single vector. Let $F$ be the cdf of the distribution over $\|p\|$ where $p \sim \mu$. Then, it holds that:*

$$F(r) = \min\left(1, \left(\frac{r^\beta}{N}\right)^{1/(P-1)}\right). \tag{13}$$

The intuition for Lemma 3 is that a single-genre equilibrium essentially reduces the producer's decision to a 1-dimensional space, and so inherits the structure of the 1-dimensional equilibrium.

To formalize the lemmas in this proof sketch, we will define a set $\mathcal{S}_{>0}$ which deletes all points with a zero coordinate from $\mathcal{S}$. More formally:

$$\mathcal{S}_{>0} := \left\{ \mathbf{U}p \mid \|p\| \leq 1, p \in \mathbb{R}_{\geq 0}^D \right\} \cap \mathbb{R}_{>0}^N.$$

For notational convenience, we also define:

$$\mathcal{B} := \left\{ p \in \mathbb{R}_{\geq 0}^D \mid \|p\| \leq 1 \right\},$$

$$\mathcal{B}_{>0} := \left\{ p \in \mathbb{R}^D_{\geq 0} \mid \|p\| \leq 1, \langle p, u_i \rangle > 0 \forall i \right\},$$

which are both convex sets. We further define:

$$\mathcal{D} := \left\{ p \in \mathbb{R}^D_{\geq 0} \mid \|p\| = 1 \right\}$$

and

$$\mathcal{D}_{>0} := \left\{ p \in \mathbb{R}^D_{\geq 0} \mid \|p\| = 1, \langle p, u_i \rangle > 0 \forall i \right\}.$$

Note that it follows from definition that:

$$\mathcal{S} = \{ \mathbf{U}p \mid p \in \mathcal{B} \}$$

$$\mathcal{S}_{>0} = \{ \mathbf{U}p \mid p \in \mathcal{B}_{>0} \}$$

**Outline for Appendix F.1.**    The proof will proceed by proving Lemma 3 and Lemma 2, and then proving Theorem 1 from these lemmas. In Section F.1.1, we prove a useful auxiliary lemma about single-genre equilibria; in Section F.1.2, we prove Lemma 3; in Appendix F.1.3, we formalize and prove Lemma 2; and in Section F.1.5, we prove Theorem 1.

### F.1.1   Auxiliary lemma

We show that at a single-genre equilibrium, it must hold that the direction vector has nonzero inner product with every user.

**Lemma 4.** *Suppose that $\mu$ is a symmetric equilibrium such that* $\mathrm{Genre}(\mu)$ *contains a single vector* $p^*$. *Then* $p^* \in span(u_1, \ldots, u_N)$ *(which also means that* $\langle p^*, u_i \rangle > 0$ *for all i.)*

*Proof.* Assume for sake of contradiction that $\langle p^*, u_i \rangle = 0$ for some $i$. Suppose that $p' \in \mathrm{supp}(\mu)$, and consider the vector $p' + \epsilon \frac{u_i}{\|u_i\|}$. We see that $p' + \epsilon \frac{u_i}{\|u_i\|}$ wins user $u_i$ with probability 1 whereas $p'$ wins user $u_i$ with probability $1/P$. The probability that $p + \epsilon u_i$ wins any other user is also at least the probability that $p'$ wins $u_i$. By leveraging this discontinuity, we see there exists $\epsilon$ such that $\mathcal{P}(p' + \epsilon \frac{u_i}{\|u_i\|}; [\mu, \ldots, \mu]) > \mathcal{P}(p'; [\mu, \ldots, \mu]) + (1 - \frac{1}{P})$ which is a contradiction.    □

### F.1.2   Proof of Lemma 3

We restate and prove Lemma 3.

**Lemma 3.** *Suppose that $\mu$ is a symmetric equilibrium such that* $\mathrm{Genre}(\mu)$ *contains a single vector. Let $F$ be the cdf of the distribution over $\|p\|$ where $p \sim \mu$. Then, it holds that:*

$$F(r) = \min \left( 1, \left( \frac{r^\beta}{N} \right)^{1/(P-1)} \right). \tag{13}$$

*Proof.* Next, we show that $F(r) = 0$ only if $r = 0$. Since the distribution $\mu$ is atomless (by Proposition 5), we can view the support as a closed set. Let $r_{\min}$ be the minimum magnitude element in the support of $\mu$. Since $\mu$ is atomless, this means that with probability 1, every producer will have magnitude greater than $r_{\min}$. This, coupled with Lemma 4, means that the producer the expected number of users achieved at $r_{\min}p$ is 0, and $\mathcal{P}(r_{\min}p; [\mu, \ldots \mu]) = -r_{\min}^\beta$. However, since $r_{\min}p \in \mathrm{supp}(\mu)$, it must hold that:

$$-r_{\min}^\beta = \mathcal{P}(r_{\min}p; [\mu, \ldots, \mu]) \geq \mathcal{P}(\vec{0}; [\mu, \ldots, \mu]) \geq 0.$$

This means that $r_{\min} = 0$.

Next, we show that the equilibrium profit at $(\mu, \ldots, \mu)$ is equal to 0. To see this, suppose that if the producer chooses $\vec{0}$. Since $\mu$ is atomless and since $\langle p^*, u_i \rangle > 0$ for all $i$ (by Lemma 4), we see that if a producer chooses $\vec{0} \in \mathrm{supp}(\mu)$, they receive 0 users in expectation. This means that $\mathcal{P}(\vec{0}; [\mu, \ldots, \mu]) = 0$ as desired.

Next, we show that $F(r) = \left( \frac{r^\beta}{N} \right)^{1/(P-1)}$ for any $rp^* \in \mathrm{supp}(\mu)$. To show this, notice that the producer must earn the same profit—here, zero profit—for any $p \in \mathrm{supp}(\mu)$. This means that for any $rp^* \in \mathrm{supp}(\mu)$, it must hold that $NF(r)^{P-1} - r^\beta = 0$. Solving, we see that $F(r) = \left( \frac{r^\beta}{N} \right)^{1/(P-1)}$.

Finally, we show that the support of $F$ is exactly $[0, N^{1/\beta}]$. First, we already showed that $r_{\min} = 0$ which means that 0 is the minimum magnitude element in the support. Moreover, $r = N^{1/\beta}$ must be the maximum magnitude element in the support since it is the unique value for which $F(r) = 1$. Now, we show that $\text{supp}(F)$ is equal to $[0, N^{1/\beta}]$. Note that the set $\text{supp}(F) \cup [N^{1/\beta}, \infty) \cup (-\infty, 0]$ is a finite union of closed sets and is thus closed. Let $S' := \mathbb{R} \setminus (\text{supp}(F) \cup [N^{1/\beta}, \infty) \cup (-\infty, 0])$; it suffices to prove that $S' = \emptyset$. Assume for sake of contradiction that $S' \neq \emptyset$. Since $S'$ is open, there exists $x \in (0, N^{1/\beta})$ and $\epsilon > 0$ such that $(x, x + \epsilon) \subseteq S'$. Let $r_1 = \inf_{y \in \text{supp}(F), y \leq x} y$ and let $r_2 = \sup_{y \in \text{supp}(F), y \geq x + \epsilon} y$. Note that both $r_1$ and $r_2$ are in $\text{supp}(F)$ (since it is closed), and $(r_1, r_2) \cap \text{supp}(F) = \emptyset$. By the structure of $F$, since $F(r_2) > F(r_1)$, this means that the cdf jumps from $F(x)$ to $F(x + \epsilon)$ anyway so there would be atoms (but there are no atoms by Proposition 5). This proves that the support is $[0, N^{1/\beta}]$.

In conclusion, we have shown that $F(r) = \left(\frac{r^\beta}{N}\right)^{1/(P-1)}$ for any $r \in [0, N^{1/\beta}]$. The $\min$ with 1 comes from the fact that $F(r) = 1$ for $r \geq N^{1/\beta}$. $\qquad\square$

### F.1.3 Formal Statement and Proof of Lemma 2

We begin with a proof sketch of Lemma 2. For $\mu$ to be an equilibrium, no alternative $q$ should do better than $p \sim \mu$, which yields the following necessary and sufficient condition after plugging into the profit function (1):

$$\sup_q \left( \sum_{i=1}^N \frac{1}{N} \left( \frac{\langle q, u_i \rangle}{\langle p^*, u_i \rangle} \right)^\beta - \|q\|^\beta \right) = \mathbb{E}_{p' \sim \mu} \left[ \sum_{i=1}^N \frac{1}{N} \left( \frac{\langle p', u_i \rangle}{\langle p^*, u_i \rangle} \right)^\beta - \|p'\|^\beta \right] \qquad (14)$$

The term $\frac{1}{N}(\cdot)^\beta$ is the probability $(F(\cdot))^{P-1}$ that $q$ outperforms the max of $P - 1$ samples from $\mu$. We next change variables according to $y_i = \langle p^*, u_i \rangle^\beta$ and $y_i' = \langle \frac{q}{\|q\|}, u_i \rangle^\beta$ and simplify to see that $\mu$ is an equilibrium if and only if $\sup_{y' \in \mathcal{S}^\beta} \sum_{i=1}^n \frac{y_i'}{y_i} = N$. Thus, there exists a single-genre equilibrium if and only if

$$\inf_{y \in \mathcal{S}^\beta} \sup_{y' \in \mathcal{S}^\beta} \sum_{i=1}^N \frac{y_i'}{y_i} = N. \qquad (15)$$

While the left-hand side of equation (15) is challenging to reason about directly, we show that the dual $\sup_{y' \in \mathcal{S}^\beta} \inf_{y \in \mathcal{S}^\beta} \sum_{i=1}^N \frac{y_i'}{y_i}$ is in fact equal to $N$.

With this proof sketch in mind, we are ready to formalize and prove Lemma 2.

**Lemma 5** (Formalization of Lemma 2). *There exists a symmetric equilibrium $\mu$ with $|\text{Genre}(\mu)| = 1$ if and only if:*

$$\inf_{p^* \in \mathcal{B}_{>0}} \sup_{y' \in \mathcal{S}^\beta} \sum_{i=1}^N \frac{y_i'}{(\langle p^*, u_i \rangle)^\beta} = \sup_{y' \in \mathcal{S}^\beta} \inf_{p^* \in \mathcal{B}_{>0}} \sum_{i=1}^N \frac{y_i'}{(\langle p^*, u_i \rangle)^\beta}. \qquad (16)$$

It turns out to be more convenient to use a (slightly less intuitive) variant of Lemma 5 to prove Theorem 1. We state and prove Lemma 6 below.

**Lemma 6.** *There exists a symmetric equilibrium $\mu$ with $|\text{Genre}(\mu)| = 1$ if and only if:*

$$\inf_{p^* \in \mathcal{B}_{>0}} \sup_{y' \in \mathcal{S}^\beta} \sum_{i=1}^N \frac{y_i'}{(\langle p^*, u_i \rangle)^\beta} \leq N. \qquad (17)$$

The main ingredient in the proof of Lemma 6 is the following characterization of a single-genre equilibrium in a given direction.

**Lemma 7.** *There is a symmetric equilibrium $\mu$ with $\text{Genre}(\mu) = \{p^*\}$ if and only if:*

$$\sup_{y' \in \mathcal{S}^\beta} \sum_{i=1}^N \frac{y_i'}{(\langle p^*, u_i \rangle)^\beta} \leq N. \qquad (18)$$

*Proof.* First, by Lemma 4, we see that the denominator is nonzero for every term in the sum, so equation (18) is well-defined.

If $\mu$ is a single-genre equilibrium, then the cdf of the magnitudes follows the form in Lemma 3. Thus, it suffices to identify necessary and sufficient conditions for that solution (that we call $\mu_{p^*}$) to be a symmetric equilibrium.

The solution $\mu_{p^*}$ is an equilibrium if and only if no alternative $q$ should do better than $p \sim \mu$. The profit level at $\mu_{p^*}$ is 0 by the structure of the cdf. Putting this all together, we see a necessary and sufficient for $\mu_{p^*}$ to be an equilibrium is:

$$\sup_{q \in \mathbb{R}^D_{\geq 0}} \left( \sum_{i=1}^{N} F \left( \frac{\langle q, u_i \rangle}{\langle p^*, u_i \rangle} \right)^{P-1} - \|q\|^\beta \right) \leq 0,$$

where the term $\frac{1}{N} \left( \cdot \right)^\beta$ is the probability $(F(\cdot))^{P-1}$ that $q$ outperforms the max of $P - 1$ samples from $\mu$. Using the structure of the cdf, we can write this as:

$$\sup_{q \in \mathbb{R}^D_{\geq 0}} \left( \sum_{i=1}^{N} \min \left( 1, \frac{1}{N} \left( \frac{\langle q, u_i \rangle}{\langle p^*, u_i \rangle} \right)^\beta \right) - \|q\|^\beta \right) \leq 0.$$

We can equivalently write this as:

$$\sup_{q \in \mathbb{R}^D_{\geq 0}} \left( \frac{1}{\|q\|^\beta} \sum_{i=1}^{N} \min \left( 1, \frac{1}{N} \left( \frac{\langle q, u_i \rangle}{\langle p^*, u_i \rangle} \right)^\beta \right) - 1 \right) \leq 0,$$

which we can equivalently write as

$$\sup_{q \in \mathcal{D}} \sup_{r > 0} \left( \frac{1}{r^\beta} \sum_{i=1}^{N} \min \left( 1, \frac{r^\beta}{N} \left( \frac{\langle q, u_i \rangle}{\langle p^*, u_i \rangle} \right)^\beta \right) - 1 \right) \leq 0.$$

For any direction $q$, if we disregard the first $\min$ with 1, the expression would be constant in $r$. With the minimum, the objective $\left( \frac{1}{r^\beta} \sum_{i=1}^{N} \min \left( 1, \frac{1}{N} \left( \frac{\langle q, u_i \rangle}{\langle p^*, u_i \rangle} \right)^\beta \right) - 1 \right)$ is weakly decreasing in $r$. Thus, $\sup_{r>0} \left( \frac{1}{r^\beta} \sum_{i=1}^{N} \min \left( 1, \frac{1}{N} \left( \frac{\langle q, u_i \rangle}{\langle p^*, u_i \rangle} \right)^\beta \right) - 1 \right)$ is attained as $r \to 0$. In fact, the maximum is attained at a value $r$ if $r \langle q, u_i \rangle < N^{1/\beta} \langle p^*, u_i \rangle$ for all $i$. This holds for *some* $r > 0$ since $\langle p^*, u_i \rangle > 0$ for all $i$ by Lemma 4. Thus we can equivalently formulate the condition as:

$$\sup_{q \in \mathcal{D}} \left( \left( \sum_{i=1}^{N} \frac{1}{N} \left( \frac{\langle q, u_i \rangle}{\langle p^*, u_i \rangle} \right)^\beta \right) - 1 \right) \leq 0,$$

which we can write as:

$$\sup_{q \in \mathcal{D}} \sum_{i=1}^{N} \left( \frac{\langle q, u_i \rangle}{(\langle p^*, u_i \rangle)} \right)^\beta \leq N.$$

This is equivalent to:

$$\sup_{q \in \mathcal{B}} \sum_{i=1}^{N} \left( \frac{\langle q, u_i \rangle}{(\langle p^*, u_i \rangle)} \right)^\beta \leq N.$$

A change of variables gives us the desired formulation. $\square$

Now, we can deduce Lemma 6.

*Proof of Lemma 6.* First, suppose that equation (17) does not hold. Then it must be true that:

$$\sup_{y' \in \mathcal{S}^\beta} \sum_{i=1}^{N} \frac{y'_i}{(\langle p^*, u_i \rangle)^\beta} > N$$

for every direction $p^* \in \mathcal{D}_{>0}$. This means that no direction in $\mathcal{D}_{>0}$ can be a single-genre equilibrium. We can further rule out directions in $\mathcal{D} \setminus \mathcal{D}_{>0}$ by applying Lemma 4.

Now, suppose that equation (17) does hold. It is not difficult to see that the optimum

$$\inf_{p^* \in \mathcal{B}_{>0}} \sup_{y' \in \mathcal{S}^\beta} \sum_{i=1}^{N} \frac{y_i'}{(\langle p^*, u_i \rangle)^\beta}$$

is attained at some direction $p^* \in \mathcal{D}_{>0}$. Applying Lemma 7, we see that there exists a single-genre equilibrium in the direction $p^*$. $\qquad \square$

### F.1.4 Proof of Lemma 5

To prove Lemma 5 from Lemma 6, we require the following additional lemma that helps us analyze the right-hand side of equation (16).

**Lemma 8.** *For any set $\mathcal{R} \subseteq \mathbb{R}_{>0}^N$, it holds that:*

$$\sup_{y' \in \mathcal{R}} \inf_{y \in \mathcal{R}} \sum_{i=1}^{N} \frac{y_i'}{y_i} = N.$$

*Proof.* By taking $y' = y$, we see that:

$$\sup_{y' \in \mathcal{R}} \inf_{y \in \mathcal{R}} \sum_{i=1}^{N} \frac{y_i'}{y_i} \leq N.$$

To show equality, notice by AM-GM that:

$$\sum_{i=1}^{N} \frac{y_i'}{y_i} \geq N \left( \prod_{i=1}^{n} \frac{y_i'}{y_i} \right)^{1/N} = N \left( \frac{\prod_{i=1}^{n} y_i'}{\prod_{i=1}^{N} y_i} \right)^{1/N}.$$

We can take $y' = \arg\max_{y'' \in \mathcal{R}} \prod_{i=1}^{n} y_i''$, and obtain a lower bound of $N$ as desired. (If the $\arg\max$ does not exist, then note that if we take $y'$ where $\prod_{i=1}^{n} y_i'$ is sufficiently close to the optimum $\sup_{y'' \in \mathcal{R}} \prod_{i=1}^{n} y_i''$, we have that $\inf_{y \in \mathcal{R}} \left( \frac{\prod_{i=1}^{n} y_i'}{\prod_{i=1}^{N} y_i} \right)^{1/N}$ is sufficiently close to 1 as desired.) $\qquad \square$

Now we are ready to prove Lemma 5.

*Proof of Lemma 5.* First, we see that:

$$\begin{aligned}
N &= \sup_{y' \in \mathcal{S}_{>0}^\beta} \inf_{y \in \mathcal{S}_{>0}^\beta} \sum_{i=1}^{N} \frac{y_i'}{y_i} \\
&= \sup_{y' \in \mathcal{S}^\beta} \inf_{y \in \mathcal{S}_{>0}^\beta} \sum_{i=1}^{N} \frac{y_i'}{y_i} \\
&= \sup_{y' \in \mathcal{S}^\beta} \inf_{p^* \in \mathcal{B}_{>0}} \sum_{i=1}^{N} \frac{y_i'}{(\langle p^*, u_i \rangle)^\beta},
\end{aligned}$$

where the first equality follows from Lemma 8.

Now, let's combine this with Lemma 6 to see that a necessary and sufficient condition for the existence of a single-genre equilibrium is:

$$\inf_{p^* \in \mathcal{B}_{>0}} \sup_{y' \in \mathcal{S}^\beta} \sum_{i=1}^{N} \frac{y_i'}{(\langle p^*, u_i \rangle)^\beta} \leq \sup_{y' \in \mathcal{S}^\beta} \inf_{p^* \in \mathcal{B}_{>0}} \sum_{i=1}^{N} \frac{y_i'}{(\langle p^*, u_i \rangle)^\beta} \qquad (19)$$

Weak duality tells us that $\inf_{p^* \in \mathcal{B}_{>0}} \sup_{y' \in \mathcal{S}^\beta} \sum_{i=1}^N \frac{y'_i}{(\langle p^*, u_i \rangle)^\beta} \geq$ $\sup_{y' \in \mathcal{S}^\beta} \inf_{p^* \in \mathcal{B}_{>0}} \sum_{i=1}^N \frac{y'_i}{(\langle p^*, u_i \rangle)^\beta}$, so equation (19) is equivalent to:

$$\inf_{p^* \in \mathcal{B}_{>0}} \sup_{y' \in \mathcal{S}^\beta} \sum_{i=1}^N \frac{y'_i}{(\langle p^*, u_i \rangle)^\beta} = \sup_{y' \in \mathcal{S}^\beta} \inf_{p^* \in \mathcal{B}_{>0}} \sum_{i=1}^N \frac{y'_i}{(\langle p^*, u_i \rangle)^\beta}.$$

$\square$

### F.1.5 Finishing the proof of Theorem 1

*Proof of Theorem 1.* Recall that by Lemma 6, a single genre equilibrium exists if and only if equation (17) is satisfied.

We can rewrite the left-hand side of equation (17) as follows:

$$\inf_{p^* \in \mathcal{B}_{>0}} \left( \sup_{y' \in \mathcal{S}^\beta} \sum_{i=1}^N \frac{y'_i}{\langle p^*, u_i \rangle^\beta} \right) = \inf_{p^* \in \mathcal{B}_{>0}} \left( \sup_{y' \in \bar{\mathcal{S}}^\beta} \sum_{i=1}^N \frac{y'_i}{\langle p^*, u_i \rangle^\beta} \right),$$

since the objective is linear in $y'$. Now, observing that the objective is convex in $p$ and concave in $y'$, we can apply Sion's min-max theorem[9] to see that:

$$\inf_{p^* \in \mathcal{B}_{>0}} \left( \sup_{y' \in \bar{\mathcal{S}}^\beta} \sum_{i=1}^N \frac{y'_i}{\langle p^*, u_i \rangle^\beta} \right) = \sup_{y' \in \bar{\mathcal{S}}^\beta} \left( \inf_{p^* \in \mathcal{B}_{>0}} \sum_{i=1}^N \frac{y'_i}{\langle p^*, u_i \rangle^\beta} \right) = \sup_{y' \in \bar{\mathcal{S}}^\beta} \left( \inf_{y \in \mathcal{S}^\beta_{>0}} \sum_{i=1}^N \frac{y'_i}{y_i} \right).$$

Thus, we have the following necessary and sufficient condition for a single-genre equilibrium to exist:

$$\sup_{y' \in \bar{\mathcal{S}}^\beta} \left( \inf_{y \in \mathcal{S}^\beta_{>0}} \sum_{i=1}^N \frac{y'_i}{y_i} \right) \leq N. \tag{20}$$

First, we show that if (4) does not hold, then there does not exist a single-genre equilibrium. Let $y' = \arg\max_{y'' \in \bar{\mathcal{S}}^\beta} \prod_{i=1}^n y''_i$. (The maximum exists because $\prod_{i=1}^n y''_i$ is a continuous function and $\bar{\mathcal{S}}^\beta$ is a compact set.) We see that:

$$\sum_{i=1}^N \frac{y'_i}{y_i} \geq N \left( \frac{\prod_{i=1}^n y'_i}{\prod_{i=1}^n y_i} \right)^{1/N} \geq N \left( \frac{\max_{y'' \in \bar{\mathcal{S}}^\beta} \prod_{i=1}^n y''_i}{\max_{y'' \in \mathcal{S}^\beta_{>0}} \prod_{i=1}^n y''_i} \right)^{1/N} = N \left( \frac{\max_{y'' \in \bar{\mathcal{S}}^\beta} \prod_{i=1}^n y''_i}{\max_{y'' \in \mathcal{S}^\beta} \prod_{i=1}^n y''_i} \right)^{1/N} > N,$$

which proves that:

$$\inf_{p^* \in \mathcal{B}_{>0}} \left( \sup_{y' \in \mathcal{S}^\beta} \sum_{i=1}^N \frac{y'_i}{\langle p^*, u_i \rangle^\beta} \right) = \sup_{y' \in \bar{\mathcal{S}}^\beta} \left( \inf_{y \in \mathcal{S}^\beta_{>0}} \sum_{i=1}^N \frac{y'_i}{y_i} \right) > N.$$

Thus equation (20) is not satisfied and a single-genre equilibrium does not exist as desired.

Next, we show that if (4) holds, then there exists a single-genre equilibrium. Let $y^* = \arg\max_{y'' \in \mathcal{S}^\beta} \prod_{i=1}^n y''_i = \arg\max_{y'' \in \mathcal{S}^\beta} \sum_{i=1}^n \log(y''_i)$. (The maximum exists because $\prod_{i=1}^n y''_i$ is a continuous function and $\mathcal{S}^\beta$ is a compact set.) By assumption, we see that $y^*$ is also the maximizer over $\bar{\mathcal{S}}^\beta$. We further see that $y^* \in \mathcal{S}^\beta_{>0}$. Using convexity of $\bar{\mathcal{S}}^\beta$, this means that for any $y' \in \bar{\mathcal{S}}^\beta$, it must hold that $\langle y' - y^*, \nabla \left( \sum_{i=1}^n \log(y^*_i) \right) \rangle \leq 0$. We can write this as:

$$\langle y' - y^*, \nabla \sum_{i=1}^n \frac{1}{y^*_i} \rangle \leq 0.$$

This can be written as:

$$\sum_{i=1}^n \frac{y'_i - y^*_i}{y^*_i} \leq 0,$$

---

[9] Note that $\bar{\mathcal{S}}^\beta$ is compact and convex and $\mathcal{B}_{>0}$ is convex (but not compact). We apply the non-compact formulation of Sion's min-max theorem in [Ha, 1981].

which implies that:

$$\sum_{i=1}^{n} \frac{y_i'}{y_i^*} \leq N.$$

Thus, we have that

$$\sup_{y' \in \bar{\mathcal{S}}^\beta} \left( \inf_{y \in \mathcal{S}_{>0}^\beta} \sum_{i=1}^{N} \frac{y_i'}{y_i^*} \right) \leq N,$$

and thus equation (20) is satisfied so a single-genre equilibrium does not exist as desired.

Next, we show that if all equilibria have multiple genres for some $\beta$, then all equilibria have multiple genres for all $\beta' \geq \beta$. $\beta' \leq \beta$. Notice that equation 4 can equivalently be restated as:

$$\max_{y \in \mathcal{S}} \prod_{i=1}^{N} y_i = \max_{y \in \bar{\mathcal{S}}^\beta} \left( \prod_{i=1}^{N} y_i \right)^{1/\beta}. \tag{21}$$

It thus suffices to show that:

$$\max_{y \in \bar{\mathcal{S}}^\beta} \left( \prod_{i=1}^{N} y_i \right)^{1/\beta} \leq \max_{y \in \bar{\mathcal{S}}^{\beta'}} \left( \prod_{i=1}^{N} y_i \right)^{1/\beta'}$$

for all $\beta' \geq \beta$. To see this, let $y$ denote the maximizer of $\max_{y \in \bar{\mathcal{S}}^\beta} \left( \prod_{i=1}^{N} y_i \right)^{1/\beta}$ (this is achieved since we are taking a maximum of a continuous function over a compact set). By definition, we see that $y$ can be written as a convex combination $\sum_{j=1}^{P} \lambda_j (x_i^j)^\beta$ where $x^1, \ldots, x^P$ denote vectors in $\mathcal{S}$ and where $\sum_{j=1}^{P} \lambda_j = 1$. In this notation, we see that:

$$\max_{y \in \bar{\mathcal{S}}^\beta} \left( \prod_{i=1}^{N} y_i \right)^{1/\beta} = \left( \prod_{i=1}^{N} \left( \sum_{j=1}^{P} \lambda_j (x_i^j)^\beta \right) \right)^{1/\beta}$$

By taking $y$ to be $\sum_{j=1}^{P} \lambda_j (x_i^j)^{\beta'}$, we see that:

$$\max_{y \in \bar{\mathcal{S}}^{\beta'}} \left( \prod_{i=1}^{N} y_i \right)^{1/\beta'} \geq \left( \prod_{i=1}^{N} \left( \sum_{j=1}^{P} \lambda_j (x_i^j)^{\beta'} \right) \right)^{1/\beta'}.$$

Notice that for any $1 \leq i \leq N$, it holds that:

$$\left( \sum_{j=1}^{P} \lambda_j (x_i^j)^{\beta'} \right) = \left( \sum_{j=1}^{P} \lambda_j ((x_i^j)^\beta)^{\beta'/\beta} \right) \geq \left( \sum_{j=1}^{P} \lambda_j ((x_i^j)^\beta) \right)^{\beta'/\beta},$$

where the last inequality follows from convexity of $f(c) = c^{\beta'/\beta}$ for $\beta' \geq \beta$. Putting this all together, we see that:

$$\max_{y \in \bar{\mathcal{S}}^{\beta'}} \left( \prod_{i=1}^{N} y_i \right)^{1/\beta'} \geq \left( \prod_{i=1}^{N} \left( \sum_{j=1}^{P} \lambda_j (x_i^j)^{\beta'} \right) \right)^{1/\beta'} \geq \left( \prod_{i=1}^{N} \left( \sum_{j=1}^{P} \lambda_j (x_i^j)^\beta \right) \right)^{1/\beta} = \max_{y \in \bar{\mathcal{S}}^\beta} \left( \prod_{i=1}^{N} y_i \right)^{1/\beta}$$

as desired. $\square$

### F.2 Proofs of corollaries of Theorem 1

We prove all of the corollaries of Theorem 1 in Section 3.2, except for Corollary 3 (proof deferred to Appendix G.2).

First, we prove Corollary 1, restated below.

**Corollary 1.** *The threshold $\beta^*$ is always at least 1. That is, if $\beta = 1$, there exists a single-genre equilibrium.*

*Proof.* When $\beta = 1$, we see that $\mathcal{S}^\beta = \mathcal{S}^1$ is a linear transformation of a convex set (the unit ball restricted to $\mathbb{R}^D_{\geq 0}$), so it is convex. This means that $\bar{\mathcal{S}}^\beta = \mathcal{S}^\beta$, and so (4) is trivially satisfied. By Theorem 1, there exists a single-genre equilibrium. $\qquad\square$

Next, we prove Corollary 2, restated below.

**Corollary 2.** *Let the cost function be $c(p) = \|p\|_q^\beta$. For any set of user vectors, it holds that $\beta^* \geq q$. If the user vectors are equal to the standard basis vectors $\{e_1, \dots, e_D\}$, then $\beta^*$ is equal to $q$.*

*Proof.* We split the proof into two steps: (1) showing that $\beta^* \geq q$ for any set of user vectors and (2) showing that $\beta^* \leq q$ for the standard basis vectors.

**Showing that $\beta^* \geq q$ for any set of users.** To show that $\beta^* \geq q$, by Theorem 1, it suffices to show that equation (4) is satisfied at $\beta = q$. Suppose that the right-hand side of (4):

$$\max_{y \in \bar{\mathcal{S}}^\beta} \prod_{i=1}^N y_i$$

is maximized at some $y^* \in \bar{\mathcal{S}}^\beta$. It suffices to construct $\tilde{y} \in \mathcal{S}^\beta$ such that

$$\prod_{i=1}^N \tilde{y}_i \geq \prod_{i=1}^N y_i^* \tag{22}$$

To construct $\tilde{y}$, we introduce some notation. By the definition of a convex hull, we can write $y^*$ as

$$y^* = \sum_{k=1}^m \lambda_k y^k,$$

where $y^1, \dots, y^m \in \mathcal{S}^\beta$ and where $\lambda_1, \dots, \lambda_m \in [0, 1]$ are such that $\sum_{k=1}^m \lambda_k = 1$. Let $p^1, \dots, p^m \in \mathbb{R}^D_{\geq 0}$ be such that $\|p^k\|_q \leq 1$ for all $1 \leq k \leq m$ and $y^k$ is given by the $\beta$-coordinate-wise powers of $\bar{U}p_k$. Now, we let $y = U\tilde{p}$ where the $d$th coordinate of $\tilde{p}$ is given by:

$$\tilde{p}_d := \left( \sum_{k=1}^m \lambda_k ((p^k)_d)^q \right)^{1/q}.$$

It follows from definition that:

$$\|\tilde{p}\|_q = \left( \sum_{d=1}^D \sum_{k=1}^m \lambda_k ((p^k)_d)^q \right)^{1/q} = \left( \sum_{k=1}^m \lambda_k \sum_{d=1}^D ((p^k)_d)^q \right)^{1/q} \leq \left( \sum_{k=1}^m \lambda_k \|p^k\|_q^q \right)^{1/q} \leq 1,$$

which means that $\tilde{y} \in \mathcal{S}^\beta$.

The remainder of the proof boils down to showing (22). It suffices to show that for every $1 \leq i \leq N$, it holds that $\tilde{y}_i \geq y_i^*$. Notice that:

$$y_i^* = \sum_{k=1}^m \lambda_k (y^k)_i = \sum_{k=1}^m \lambda_k \langle u_i, p^k \rangle^q = \sum_{k=1}^m \lambda_k \left( \sum_{d=1}^D (u_i)_d (p^k)_d \right)^q,$$

and

$$\tilde{y}_i = \langle u_i, \tilde{p} \rangle^q = \left( \sum_{d=1}^D (u_i)_d \tilde{p}_d \right)^q = \left( \sum_{d=1}^D (u_i)_d \left( \sum_{k=1}^m \lambda_k ((p^k)_d)^q \right)^{1/q} \right)^q.$$

Thus, it suffices to show the following inequality:

$$\sum_{d=1}^D (u_i)_d \left( \sum_{k=1}^m \lambda_k ((p^k)_d)^q \right)^{1/q} \geq \left( \sum_{k=1}^m \lambda_k \left( \sum_{d=1}^D (u_i)_d (p^k)_d \right)^q \right)^{1/q}. \tag{23}$$

The high-level idea is that the proof boils down to the triangle inequality for an appropriately chosen norm over $\mathbb{R}^m$. For $z \in \mathbb{R}^m$, we let:

$$\|z\|_\lambda := \left( \sum_{k=1}^m \lambda_k z^q \right)^{1/q}.$$

To see that this is a norm, note that $\left( \sum_{k=1}^m \lambda_k z^q \right)^{1/q} = \left( \sum_{k=1}^m (\lambda_k^{1/q} z)^q \right)^{1/q}$. The norm properties of this function are implied by the norm properties of $\| \cdot \|_q$. By the triangle inequality, we see that:

$$\sum_{d=1}^D (u_i)_d \left( \sum_{k=1}^m \lambda_k ((p^k)_d)^q \right)^{1/q} = \sum_{d=1}^D (u_i)_d \| [p_d^1, \ldots, p_d^m] \|_\lambda$$

$$\geq \| \sum_{d=1}^D (u_i)_d [p_d^1, \ldots, p_d^m] \|_\lambda$$

$$= \left( \sum_{k=1}^m \lambda_k \left( \sum_{d=1}^D (u_i)_d (p^k)_d \right)^q \right)^{1/q}$$

which implies equation (23).

**Showing that $\beta^* \leq q$ for the standard basis vectors.** By Theorem 1, it suffices to show, for any $\beta > q$, that equation (4) is not satisfied. First, we compute the left-hand side of equation (4):

$$\max_{y \in \mathcal{S}^\beta} \prod_{i=1}^N y_i = \left( \max_{x \in \mathbb{R}_{\geq 0}^D, \|x\|_q = 1} \prod_{i=1}^D x_i \right)^\beta = \left( \frac{1}{D} \right)^{\beta/q} < \left( \frac{1}{D} \right).$$

where the last line follows from AM-GM. Now, we compute the right-hand side:

$$\max_{y \in \bar{\mathcal{S}}^\beta} \prod_{i=1}^N y_i.$$

Consider $y^* = \left[ \frac{1}{D}, \ldots, \frac{1}{D} \right]$. Notice that $y$ is a convex combination of the standard basis vectors—all of which are in $\mathcal{S}$ and actually in $\mathcal{S}^\beta$ too—so $y \in \bar{\mathcal{S}}^\beta$. This means that

$$\max_{y \in \bar{\mathcal{S}}^\beta} \prod_{i=1}^N y_i \geq \prod_{i=1}^N y_i^* = \left( \frac{1}{D} \right).$$

This proves that:

$$\max_{y \in \mathcal{S}^\beta} \prod_{i=1}^N y_i < \max_{y \in \bar{\mathcal{S}}^\beta} \prod_{i=1}^N y_i,$$

so equation (4) is not satisfied as desired.

$\square$

We prove Corollary 4, restated below.

**Corollary 4.** *Let $\| \cdot \|_*$ denote the dual norm of $\| \cdot \|$, defined to be $\|p\|_* = \max_{\|p\|=1, p \in \mathbb{R}_{\geq 0}^D} \langle q, p \rangle$. Let $Z := \| \sum_{n=1}^N \frac{u_n}{\|u_n\|_*} \|_*$. Then,*

$$\beta^* \leq \frac{\log(N)}{\log(N) - \log(Z)}. \tag{5}$$

*Proof.* WLOG assume that the users to have unit dual norm. By Theorem 1, it suffices to show that:

$$\max_{y \in \mathcal{S}^\beta} \prod_{i=1}^N y_i < \max_{y \in \bar{\mathcal{S}}^\beta} \prod_{i=1}^N y_i.$$

First, let's lower bound the right-hand side. Consider the point $y = \frac{1}{N}\sum_{i'=1}^{N} z^{i'}$ where $z^{i'}$ is defined to be the $\beta$-coordinate-wise power of $\mathbf{U}\left(\arg\max_{||p||=1}\langle p, u_i\rangle\right)$. This means that

$$y_i \geq \frac{1}{N}z_i^i = \frac{1}{N}\left(\max_{||p||=1}\langle p, u_i\rangle\right)^{\beta} = \frac{||u_i||_*^{\beta}}{N} = \frac{1}{N}.$$

This means that:

$$\max_{y\in\mathcal{S}^{\beta}}\prod_{i=1}^{N} y_i \geq \frac{1}{N^N}.$$

Next, let's upper bound the left-hand side. By AM-GM, we see that:

$$\max_{y\in\mathcal{S}^{\beta}}\prod_{i=1}^{N} y_i = \max_{||p||=1, p\in\mathbb{R}_{\geq 0}^{D}}\left(\prod_{i=1}^{N}\langle p, u_i\rangle\right)^{\beta} \leq \left(\frac{\sum_{i=1}^{N}\langle p, u_i\rangle}{N}\right)^{N\beta} \leq \left(\frac{\langle p, \sum_{i=1}^{N} u_i\rangle}{N}\right)^{N\beta} \leq \frac{\left(||\sum_{i=1}^{N} u_i||_*\right)^{N\beta}}{N^{N\beta}}.$$

Putting this all together, we see that it suffices for:

$$\frac{1}{N^N} > \frac{\left(||\sum_{i=1}^{N} u_i||_*\right)^{N\beta}}{N^{N\beta}},$$

which we can rewrite as:

$$N^{\beta-1} > \left(||\sum_{i=1}^{N} u_i||_*\right)^{\beta}$$

which we can rewrite as:

$$N^{1-1/\beta} > ||\sum_{i=1}^{N} u_i||_*.$$

$\square$

We prove Corollary 5, restated below.

**Corollary 5.** *If there exists $\mu$ with $|\text{Genre}(\mu)| = 1$, then the corresponding producer direction maximizes Nash social welfare of the users:*

$$\text{Genre}(\mu) = \arg\max_{||p||=1|p\in\mathbb{R}_{\geq 0}^{D}}\sum_{i=1}^{N}\log(\langle p, u_i\rangle). \tag{6}$$

*Proof.* Corollary 5 follows as a consequence of the proof of Theorem 1. We apply Lemma 7 to see that if $\mu$ is a single-genre equilibrium with $\text{Genre}(\mu) = \{p^*\}$, then:

$$\sup_{y'\in\mathcal{S}^{\beta}}\sum_{i=1}^{N}\frac{y_i'}{(\langle p^*, u_i\rangle)^{\beta}} \leq N.$$

We see that:

$$N \geq \sup_{y'\in\mathcal{S}^{\beta}}\frac{y_i'}{(\langle p^*, u_i\rangle)^{\beta}} \geq N\sup_{y'\in\mathcal{S}^{\beta}}\left(\frac{\prod_{i=1}^{N} y_i'}{\prod_{i=1}^{N}(\langle p^*, u_i\rangle)^{\beta}}\right)^{1/N} \geq N\left(\frac{\sup_{y'\in\mathcal{S}^{\beta}}\prod_{i=1}^{N} y_i'}{\prod_{i=1}^{N}(\langle p^*, u_i\rangle)^{\beta}}\right)^{1/N}.$$

This implies that:

$$\prod_{i=1}^{N} y_i = \prod_{i=1}^{N}(\langle p^*, u_i\rangle)^{\beta} \geq \sup_{y'\in\mathcal{S}^{\beta}}\prod_{i=1}^{N} y_i',$$

where $y\in\mathcal{S}^{\beta}$ is defined so that $y_i = \langle p^*, u_i\rangle^{\beta}$. This implies that:

$$p^* \in \arg\max_{||p||\leq 1, p\in\mathbb{R}_{\geq 0}^{D}}\sum_{i=1}^{N}\log(\langle p, u_i\rangle) = \arg\max_{||p||=1, p\in\mathbb{R}_{\geq 0}^{D}}\sum_{i=1}^{N}\log(\langle p, u_i\rangle)$$

as desired. $\square$

# G   Proofs and Details for Section D

In Appendix G.1, we provide an overview of how we leverage Lemma 1 to analyze equilibria in the setting of two populations of users. In Appendix G.2, we prove Corollary 3. In Appendix G.3, we prove the results from Section D.1, and in Section G.4, we prove the results from Section D.2. In Appendix G.5, we formalize the infinite-producer limit, which we study in Section D.3, and in Appendix G.6, we prove results from Section D.3. In Appendix G.7, we prove several auxiliary lemmas that we used along the way.

## G.1   Overview of proof techniques

Before diving into proof techniques, we observe that it suffices to study a simpler setting with *two normalized users* and a rescaled cost function.

**Claim 1.** *A distribution $\mu$ is an equilibria for a marketplace with 2 populations of users of size $N/2$ located at vectors $u_1$ and $u_2$ and with producer cost function $c(p) = \|p\|_2^\beta$ if and only if $\mu$ is an equilibria for a marketplace with 2 users located at vectors $\frac{u_1}{\|u_1\|}$ and $\frac{u_2}{\|u_2\|}$ and with producer cost function $c(p) = \frac{2}{N}\|p\|_2^\beta$.*

Thus, we focus on marketplaces with 2 users located at vectors $u_1$ and $u_2$ such that $\|u_1\| = \|u_2\| = 1$ and with producer cost function $c(p) = \alpha\|p\|_2^\beta$ for $\alpha > 0$.

The proofs in this section boil down to leveraging conditions (C1)-(C3) in Lemma 1, restated below.

**Lemma 1.** *Let $\mathbf{U} = [u_1; u_2; \ldots; u_N]$ be the $N \times D$ matrix of users vectors. Given a set $S \subseteq \mathbb{R}_{\geq 0}^N$ and distributions $H_1, \ldots, H_N$ over $\mathbb{R}_{\geq 0}$, suppose that the following conditions hold:*

*(C1) Every $z^* \in S$ is a maximizer of the equation:*

$$\max_{z \in \mathbb{R}_{\geq 0}^D} \sum_{i=1}^N H_i(z_i) - c_{\mathbf{U}}(z), \tag{10}$$

*where $c_{\mathbf{U}}(z) := \min\{c(p) \mid p \in \mathbb{R}_{\geq 0}^D, \mathbf{U}p = z\}$.*

*(C2) There exists a random variable $Z$ with support $S$, such that the marginal distribution $Z_i$ has cdf equal to $H_i(z)^{1/(P-1)}$.*

*(C3) $Z$ is distributed as $\mathbf{U}Y$ with $Y \sim \mu$, for some distribution $\mu$ over $\mathbb{R}_{\geq 0}^D$.*

*Then, the distribution $\mu$ from (C3) is a symmetric mixed Nash equilibrium. Moreover, every symmetric mixed Nash equilibrium $\mu$ is associated with some $(H_1, \ldots, H_N, S)$ that satisfy (C1)-(C3).*

*Proof of Lemma 1.* The intuition is that the the set $S$ captures the support of the realized user values $[\langle u_1, p\rangle, \ldots, \langle u_N, p\rangle]$ for $p \sim \mu$ and the distribution $H_i$ captures the distribution of the maximum user value $\max_{1 \leq j \leq P-1}\langle u_i, p_j\rangle$ for user $u_i$.

To formalize this, we reparameterize from content vectors in $\mathbb{R}_{\geq 0}^D$ to realized user values in $\mathbb{R}_{\geq 0}^N$. That is, we transform the content vector $p \in \mathbb{R}_{\geq 0}^D$ into the vector of realized user values given by $z = \mathbf{U}p$. This reparameterization allows us to cleanly reason about the number of users that a producer wins: a producer wins a user $u_i$ if and only if they have the highest value in the $i$th coordinate of $z$. In this parametrization, the cost of production can be computed through an induced function $c_{\mathbf{U}}$ given by $c_{\mathbf{U}}(z) := \min\{c(p) \mid p \in \mathbb{R}_{\geq 0}^D, z = \mathbf{U}p\}$ if $z \in \{\mathbf{U}p \mid p \in \mathbb{R}_{\geq 0}^D\}$.

In this reparameterization, the producer profit takes a clean form. If producer 1 chooses $z \in \mathbb{R}^N$, and other producers follow a distribution $\mu_Z$ over $\mathbb{R}^N$, then the expected profit of producer 1 is:

$$\sum_{i=1}^N H_i(z_i) - c_{\mathbf{U}}(z),$$

where $H_i(\cdot)$ is the cumulative distribution function of the maximum realized user value over the other $P - 1$ producers, i.e. of the random variable $\max_{2 \leq j \leq P}(z_j)_i$ where $z_2, \ldots, z_P \sim \mu_Z$.

Recall that a distribution $\mu$ corresponds to a symmetric mixed Nash equilibrium if and only if every $z$ in the support $S := \mathrm{supp}(\mu_Z)$ is a maximizer of equation (10) (where $\mu_Z$ is the distribution over $\mathbf{U}p$ for $p \sim \mu$).

$\square$

### G.1.1 Leveraging (C1)

To leverage (C1), we use the first-order and second-order conditions for $z$ to be a maximizer of equation (10). In order to obtain useful closed-form expressions, we explicitly compute the induced cost function in terms of the angle $\theta^*$ between the user vectors.

**Lemma 9.** *Let there be 2 users located at* $u_1, u_2 \in \mathbb{R}^D_{\geq 0}$ *such that* $\|u_1\| = \|u_2\| = 1$, *and let* $\theta^* := \cos^{-1}(\langle u_1, u_2 \rangle) > 0$ *be the angle between the user vectors. Let the cost function be* $c(p) = \alpha\|p\|_2^\beta$ *for* $\alpha > 0$. *For any* $z \in \{\mathbf{U}p \mid p \in \mathbb{R}^D_{\geq 0}\}$, *the induced cost function is given by:*

$$c_\mathbf{U}(z) = \alpha \sin^{-\beta}(\theta^*) \left(z_1^2 + z_2^2 - 2z_1 z_2 \cos(\theta^*)\right)^{\frac{\beta}{2}}.$$

**First-order condition.** The first order condition implies that we can compute the densities $h_1$ and $h_2$ of $H_1$ and $H_2$ in terms of the $c_\mathbf{U}$. The densities $h_1(z_1)$ and $h_2(z_2)$ depend on the gradient $\nabla_z c_\mathbf{U}$ and *both* coordinates $z_1$ and $z_2$.

**Lemma 10.** *Let there be 2 users located at* $u_1, u_2 \in \mathbb{R}^D_{\geq 0}$ *such that* $\|u_1\| = \|u_2\| = 1$, *and let* $\theta^* := \cos^{-1}(\langle u_1, u_2 \rangle) > 0$ *be the angle between the user vectors. Let the cost function be* $c(p) = \alpha\|p\|_2^\beta$ *for* $\alpha > 0$. *For any* $z \in \{\mathbf{U}p \mid p \in \mathbb{R}^D_{\geq 0}\}$, *the first-order condition of equation (10) can be written as:*

$$\begin{bmatrix} h_1(z_1) \\ h_2(z_2) \end{bmatrix} = \nabla_z(c_\mathbf{U}(z)).$$

*More specifically, it holds that:*

$$\begin{bmatrix} h_1(z_1) \\ h_2(z_2) \end{bmatrix} = \beta\alpha \sin^{-\beta}(\theta^*) \left(z_1^2 + z_2^2 - 2z_1 z_2 \cos(\theta^*)\right)^{\frac{\beta}{2}-1} \begin{bmatrix} z_1 - z_2 \cos(\theta^*) \\ z_2 - z_1 \cos(\theta^*) \end{bmatrix},$$

*and if we represent* $z = \mathbf{U}[r\cos(\theta), r\sin(\theta)]$, *then it also holds that:*

$$\begin{bmatrix} h_1(z_1) \\ h_2(z_2) \end{bmatrix} = \beta\alpha r^{\beta-1} \begin{bmatrix} \frac{\sin(\theta^*-\theta)}{\sin(\theta^*)} \\ \frac{\sin(\theta)}{\sin(\theta^*)} \end{bmatrix}.$$

**Second-order condition.** When we also take advantage of the second-order condition, we can identify the "direction" that the support must point at $z \in S$ terms of the location of $z$, the cost function parameter $\beta$, and the angle $\theta^*$ between the two populations of users.

**Lemma 11.** *Let there be 2 users located at* $u_1, u_2 \in \mathbb{R}^D_{\geq 0}$ *such that* $\|u_1\| = \|u_2\| = 1$, *and let* $\theta^* := \cos^{-1}(\langle u_1, u_2 \rangle) > 0$ *be the angle between the user vectors. Let the cost function be* $c(p) = \alpha\|p\|_2^\beta$ *for* $\alpha > 0$. *If* $z$ *is of the form* $[r\cos(\theta), r\cos(\theta^*-\theta)]$ *for* $\theta \in [0, \theta^*]$, *then the sign of* $\frac{\partial^2 c_\mathbf{U}(z)}{\partial z_1 \partial z_2}$ *is equal to the sign of:*

$$\frac{\beta-2}{\beta}\cos(\theta^*-2\theta) - \cos(\theta^*).$$

**Lemma 12.** *Let there be 2 users located at* $u_1, u_2 \in \mathbb{R}^D_{\geq 0}$ *such that* $\|u_1\| = \|u_2\| = 1$, *and let* $\theta^* := \cos^{-1}(\langle u_1, u_2 \rangle) > 0$ *be the angle between the user vectors. Let the cost function be* $c(p) = \alpha\|p\|_2^\beta$ *for* $\alpha > 0$. *Suppose that condition (C1) is satisfied for* $(H_1, H_2, S)$. *If* $S$ *contains a curve of the form* $\{(z_1, g(z_1)) \mid x \in I\}$ *for any open interval* $I$ *and any differentiable function* $g$, *then for any* $z_1 \in I$, *it holds that:*

$$g'(z_1) \cdot \left(\frac{\beta-2}{\beta}\cos(\theta^*-2\theta) - \cos(\theta^*)\right) \leq 0.$$

Lemmas 11 and 12 demonstrate that if $\left(\frac{\beta-2}{\beta}\cos(\theta^*-2\theta) - \cos(\theta^*)\right) > 0$, then the curve $g$ must be decreasing, and if $\left(\frac{\beta-2}{\beta}\cos(\theta^*-2\theta) - \cos(\theta^*)\right) < 0$, then the curve $g$ must be increasing. This characterizes the "direction" of the curve in terms of the location $z_1$.

### G.1.2 Leveraging (C3)

For the case of 2 users with cost function $c(p) = \|p\|_2^\beta$, the condition (C3) always holds, as long as condition (C1) holds. Since the two vectors $u_1$ and $u_2$ are linearly independent, the matrix $\mathbf{U}$ is invertible, so we can define $\mu$ to be the distribution given by $\mathbf{U}^{-1}Z$. The only remaining condition comes $p$ being restricted to $\mathbb{R}_{\geq 0}^D$ rather than $\mathbb{R}^D$. This means that $S$ must be contained in the convex cone generated by $[1, \cos(\theta^*)]$ and $[\cos(\theta^*), 1]$. This restriction on $S$ is already implicitly implied by (C1): it is not difficult to see that all maximizers of (10) will be contained in this convex cone.

### G.1.3 Leveraging (C2)

To leverage (C2), we obtain a functional equation that restricts the relationship between $H_1$, $H_2$, and $S$ for a given value of $P$, and we instantiate this in two ways. First, when the support is a curve $(z_1, g(z_1))$, the marginal distributions $Z_1$ and $Z_2$ are related by a change of variables formula given by $Z_2 \sim g(Z_1)$. This translates into a condition on $H_1$ and $H_2$ that depends on the derivative $g'$ and the number of producers $P$. Second, if the equilibrium were to contain finitely many genres, there would be a pair of functional equations relating the cdfs $H_1$ and $H_2$, the distribution over quality within each genre, and the number of producers $P$. We describe each of these settings in more detail below.

**Case 1: support is a single curve.** The first case where we instantiate (C2) is when $S$ is equal to $\{(z_1, g(z_1)) \mid x \in M\}$ where $M$ is a (well-behaved) subset of $\mathbb{R}_{\geq 0}$. Let $h_1^*$ and $h_2^*$ be the densities of the marginal distributions $Z_1$ and $Z_2$ respectively. Since $Z_2 \sim g(Z_1)$, the change of variables formula implies that the densities $h_1^*$ and $h_2^*$ are related as follows:

$$h_1^*(z_1) = h_2^*(g(z_1))|g'(z_1)|, \tag{24}$$

In order to use equation (24), we need to translate it into a condition on the distributions $H_1$ and $H_2$. Let $h_1$ and $h_2$ be the densities of $H_1$ and $H_2$ respectively. Then equation (24) can reformulated as:

$$\frac{h_1(x)}{(H_1(x))^{\frac{P-2}{P-1}}} = \frac{h_2(g(x))}{(H_2(g(x)))^{\frac{P-2}{P-1}}}|g'(x)|. \tag{25}$$

Equation (25) reveals that the constraint induced by the number of producers $P$ can be messy in general, since it involves both the densities $h_1$ and $h_2$ and the cdfs $H_1$ and $H_2$. Intuitively, these complexities arise because $H_i^*$ and $H_i$ are related by a $(P-1)$th degree polynomial (put differently, the maximum of $P-1$ i.i.d. draws of a random variable does not generally have a clean structure). Nonetheless, equation (25) does simplify into a tractable form in special cases. For example, if $P = 2$, then the dependence on $H_1$ and $H_2$ vanishes. As another example, if $g$ is *increasing*, then $H_1(x) = H_2(g(x))$ for any $P \geq 2$, so the dependence on $H_1$ and $H_2$ again vanishes.

**Case 2: two-genre equilibria.** The second case where we instantiate (C2) is when $S$ is a subset of the union of two lines: that is,

$$S \subseteq \{(z_1, c_1 \cdot z_1) \mid z_1 \in \mathbb{R}_{\geq 0}\} \cup \{(z_1, c_2 \cdot z_1) \mid z_1 \in \mathbb{R}_{\geq 0}\},$$

where $\cos(\theta^*) \leq c_1, c_2 \leq \frac{1}{\cos(\theta^*)}$. Since linear transformations preserve lines through the origin, this means that the support of the distribution $\mu$ of $U^{-1}Z$ is also contained in the union of two lines through the origin: thus $|\text{Genre}(\mu)| \leq 2$.

A distribution $Z$ can be entirely specified by the probabilities $\alpha_1 + \alpha_2$ that it places on each of the two lines and the conditional distribution of $Z_1$ along each of the lines (this in particular determines the conditional distribution of $Z_2$ along the lines). More specifically, the probabilities $\alpha_1 + \alpha_2$ will correspond to

$$\alpha_1 := \mathbb{P}_Z[Z \in \{(z_1, c_1 \cdot z_1) \mid z_1 \in \mathbb{R}_{\geq 0}\}]$$
$$\alpha_2 := \mathbb{P}_Z[Z \in \{(z_1, c_2 \cdot z_1) \mid z_1 \in \mathbb{R}_{\geq 0}\}],$$

and $F_1$ and $F_2$ will correspond to the cdfs of the conditional distributions

$$F_1 \sim Z_1 \mid Z \in \{(z_1, c_1 \cdot z_1\}$$
$$F_2 \sim Z_1 \mid Z \in \{(z_1, c_2 \cdot z_1\}$$

respectively. The (unique) distribution $Z$ associated with $\alpha_1, \alpha_2, F_1, F_2$ satisfies (C2) if and only if the following pairs of functional equations are satisfied:

$$(\alpha_1 F_1(z_1) + \alpha_2 F_2(z_1)) = (H_1(z_1))^{\frac{1}{P-1}} \text{ and } \left(\alpha_1 F_1(c_1^{-1} z_2) + \alpha_2 F_2(c_2^{-1} z_2)\right) = (H_2(z_2))^{\frac{1}{P-1}}.$$
(26)

The functional equations can be solved to determine if there is a valid solution.

## G.2 Proof of Corollary 3

We prove Corollary 3, restated below:

**Corollary 3.** *Suppose that there are $N$ users split equally between two linearly independently vectors $u_1, u_2 \in \mathbb{R}_{\geq 0}^D$, and let $\theta^* := \cos^{-1}\left(\frac{\langle u_1, u_2 \rangle}{\|u_1\|_2 \|u_2\|}\right)$. If the cost function is $c(p) = \|p\|_2^\beta$, then,*

$$\beta^* = \frac{2}{1 - \cos(\theta^*)}.$$

*Proof.* By Claim 1, we can assume that there are 2 normalized users $\|u_1\| = \|u_2\|$. We further assume WLOG that $u_1 = e_1$.

We claim that if there is a single-genre equilibrium, it must be in the direction of $[\cos(\theta^*/2), \sin(\theta^*/2)]$. By Corollary 5, if there is a single-genre equilibrium in a direction $p$, then it must maximize $\log(\langle p, u_1 \rangle) + \log(\langle p, u_1 \rangle)$. Let's let $p = [\cos(\theta), \sin(\theta)]$. Then, we see that:

$$\log(\langle p, u_1 \rangle) + \log(\langle p, u_2 \rangle) = \log(\cos(\theta)) + \log(\cos(\theta^* - \theta)) = \log\left(\frac{\cos(\theta^*) + \cos(\theta^* - 2\theta)}{2}\right),$$

which is uniquely maximized at $\theta = \theta^*/2$ as desired.

We first show that $\beta^* \leq \frac{2}{1-\cos(\theta^*)}$. Assume for sake of contradiction that there is a single-genre equilibrium. The above argument shows that it must be in the direction of $[\cos(\theta^*/2), \sin(\theta^*/2)]$. By Lemma 3, we know that the support of the equilibrium distribution is a line segment. If $\beta > \frac{2}{1-\cos(\theta^*)}$, we see that

$$\frac{\beta - 2}{\beta} \cos(\theta^* - 2\theta) - \cos(\theta^*) = 1 - \frac{2}{\beta} - \cos(\theta^*) < 0.$$

By Lemma 1 and Lemma 12, we see that the single-genre line $(z, g(z))$ must have $g'(z_1) \leq 0$ in its support, which is a contradiction.

We next show that $\beta^* \leq \frac{2}{1-\cos(\theta^*)}$. It suffices to show that the single-genre distribution in the direction of $[\cos(\theta^*/2), \sin(\theta^*/2)]$ with cdf given by $F(q) = \left(\frac{q^\beta}{2}\right)^{1/(P-1)}$. We apply Claim 1; it suffices to verify condition (C1). Notice that

$$H_1(w) = H_2(w) = \left(\frac{w^\beta}{2\cos^\beta(\theta^*/2)}\right).$$

Thus, equation (10) can be written as:

$$\max_z \left(\min(1, \frac{z_1^\beta}{2\cos^\beta(\theta^*/2)}) + \min(1, \frac{z_2^\beta}{2\cos^\beta(\theta^*/2)}) - c_{\mathbf{U}}(z)\right).$$

It suffices to show that that for all $z$, it holds that:

$$z_1^\beta + z_2^\beta - 2\cos^\beta(\theta^*/2)\left(\frac{z_1^2 + z_2^2 - 2z_1 z_2 \cos(\theta^*)}{\sin^2(\theta^*)}\right)^\beta \leq 0.$$

Let $z = [r\cos(\theta), r\cos(\theta^* - \theta)]$. Then this reduces to:

$$\cos^\beta(\theta) + \cos^\beta(\theta^* - \theta) \leq 2\cos^\beta(\theta^*/2) \leq 0.$$

We observe that $\cos^\beta(\theta) + \cos^\beta(\theta^* - \theta)$ is maximized at $\theta = \theta^*/2$, which proves the desired statement.

$\square$

### G.3 Proofs for Section D.1

We prove Proposition 6, restated below:

**Proposition 6.** *Suppose that there are $N$ users split equally between two linearly independently vectors $u_1, u_2 \in \mathbb{R}^2_{\geq 0}$, and let $\theta^* := \cos^{-1}\left(\frac{\langle u_1, u_2 \rangle}{\|u_1\|_2 \|u_2\|}\right)$ be the angle between the user vectors. Let the cost function be $c(p) = \|p\|_2^\beta$, and let $P \geq 2$. Let $\mu$ be a symmetric Nash equilibrium such that the distributions $\langle u_1, p \rangle$ and $\langle u_2, p \rangle$ over $\mathbb{R}_{\geq 0}$ are absolutely continuous. As long as $\beta \neq 2$ or $\theta^* \neq \pi/2$, the support of $\mu$ does not contain an $\ell_2$-ball of radius $\epsilon$ for any $\epsilon > 0$.*[10]

*Proof of Proposition 6.* Assume for sake of contradiction that the support of $\mu$ contains an $\ell_2$-ball of radius $\epsilon_1 > 0$. We apply Lemma 1 and show that condition (C1) is violated. Since $\mu$ contains a ball of $\epsilon_1$-radius ball, we know that the distribution $Z$ over $\mathbf{U}p$ over $p \sim \mu$ contains an $\ell_2$ ball of radius $\epsilon_2 > 0$. Let this ball be $B$. Notice that $Z_1$ and $Z_2$ are absolutely continuous by assumption, $Z_1$ and $Z_2$ have bounded support, and the function $m \mapsto m^{P-1}$ is Lipschitz on any bounded interval: this means that $H_1$ and $H_2$ are also absolutely continuous. This means that densities exist a.e. For $(z_1, z_2) \in B$, we can apply the first-order condition in Lemma 10 to obtain that:

$$h_1(z_1) = \frac{\partial c_{\mathbf{U}}(z)}{\partial z_1}$$

We see that this needs to be satisfied for $z = [z_1, m]$ where $m \in (z_2 - \epsilon', z_2 + \epsilon')$. This means that the mapping $m \mapsto \frac{\partial c_{\mathbf{U}}([z_1, m])}{\partial z_1}$ needs to be a constant on $m \in (z_2 - \epsilon', z_2 + \epsilon')$. This means that the derivative of this mapping with respect to $z_2$ needs to be $0$, so:

$$\frac{\partial^2 c_{\mathbf{U}}([z_1, z_2])}{\partial z_1 \partial z_2} = 0 \tag{27}$$

for all $z \in B$.

We apply Lemma 11 to show that equation (27) cannot be zero on all of $B$. For all $z$ that satisfy equation (27), Lemma 11 implies if we represent $z$ as $\mathbf{U}[r\cos(\theta), r\sin(\theta)]$, then

$$\frac{\beta - 2}{\beta} \cos(\theta^* - 2\theta) = \cos(\theta^*).$$

If equation (27) holds for all $z \in B$, then it must hold at all $\theta$ within some nonempty interval. This is a contradiction as long as $\beta \neq 2$ or $\theta^* \neq \pi/2$.

For the special case where $\beta = 2$ and $\theta^* = \pi/2$, $\qquad\qquad\qquad\qquad\qquad\qquad\qquad\qquad\square$

We next prove Theorem 2, restated below:

**Theorem 2.** *Suppose that there are $N$ users split equally between two linearly independently vectors $u_1, u_2 \in \mathbb{R}^D_{\geq 0}$, and let $\theta^* := \cos^{-1}\left(\frac{\langle u_1, u_2 \rangle}{\|u_1\|_2 \|u_2\|}\right)$ be the angle between the user vectors. Let the cost function be $c(p) = \|p\|_2^\beta$. Let $\mu$ be a a distribution on $\mathbb{R}^d$ such that the distributions $\langle u_1, p \rangle$ and $\langle u_2, p \rangle$ over $\mathbb{R}_{\geq 0}$ over $\mathbb{R}_{\geq 0}$ for $p \sim \mu$ are absolutely continuous and twice continuously differentiable within their supports. There are two regimes based on $\beta$ and $\theta^*$:*

1. *If $\beta < \beta^* = \frac{2}{1 - \cos(\theta^*)}$ and if $\mu$ is a symmetric mixed equilibrium, then $\mu$ satisfies $|\mathrm{Genre}(\mu)| = 1$.*

2. *If $\beta > \beta^* = \frac{2}{1 - \cos(\theta^*)}$, if $|\mathrm{Genre}(\mu)| < \infty$, and if the conditional distribution of $\|p\|$ along each genre is continuously differentiable, then $\mu$ is not an equilibrium.*

We split into two propositions: together, these propositions directly imply Theorem 2.

**Proposition 9.** *Consider the setup in Theorem 2. If $\beta < \beta^* = \frac{2}{1 - \cos(\theta^*)}$ and $\mu$ is a symmetric mixed equilibrium, then $\mu$ satisfies $|\mathrm{Genre}(\mu)| = 1$.*

---

[10]The case of $\beta = 2$ and $\theta^* = \pi/2$ is degenerate and permits a range of possible equilibria.

**Proposition 10.** *Consider the setup in Theorem 2. If $\beta > \beta^* = \frac{2}{1-\cos(\theta^*)}$, if $|\mathrm{Genre}(\mu)| < \infty$, and if the conditional distribution of $\|p\|$ along each genre is continuous differentiable, then $\mu$ is not an equilibrium.*

To prove Proposition 9, we leverage the machinery given by Lemma 1 as follows. Condition (C1) helps us show that the support $S$ can be specified by $(w, g(w))$ for an increasing function $w$: in particular, Lemma 10 enables us to show that $S$ must be one-to-one, and Lemma 12 enables us to pin down the sign of $g'$. Using condition (C2), which simplifies since $g$ is increasing, we show a functional equation in terms of $g$ that has a unique solution at the single-genre equilibrium. We formalize this below.

*Proof of Proposition 9.* By Claim 1, it suffices to focus on the case of 2 normalized users. By Lemma 1, it suffices to study $(H_1, H_2, S)$ that satisfy (C1), (C2), and (C3).

Let $\mathrm{supp}(H_1) = I_1$ and let $\mathrm{supp}(H_2) = I_2$. Note that since the distributions are twice continuously differentiable, we know that the densities $h_1$ and $h_2$ exist and are continuously differentiable a.e on $I_1$ and $I_2$ respectively. We break the proof into several steps.

**Step 1: there exists a one-to-one function $g$ such that $S = \{(w, g(w)) \mid w \in I_1\}$ and where $g$ is continuously differentiable and strictly increasing.** We first show that $\frac{\partial^2 c_{\mathbf{U}}(z)}{\partial z_1 \partial z_2} < 0$ everywhere. By Lemma 11, it suffices to show that $\frac{\beta-2}{\beta}\cos(\theta^* - 2\theta) - \cos(\theta^*) < 0$. To see this, notice that

$$\frac{\beta - 2}{\beta}\cos(\theta^* - 2\theta) - \cos(\theta^*) < 0 \leq \frac{\beta - 2}{\beta} - \cos(\theta^*) = 1 - \cos(\theta^*) - \frac{2}{\beta} < 0$$

because $\beta < \frac{2}{1-\cos(\theta^*)}$.

We now show that the support $S$ is equal to $\{(w, g(w)) \mid w \in I_1\}$ for some one-to-one function $g : I_1 \to I_2$. To show this, it suffices to show that the support does contain both $(z_1, z_2)$ and $(z_1, z_2')$ for $z_2 \neq z_2'$ (and, analogously, the support does not contain both $(z_1', z_2)$ and $(z_1, z_2)$ for $z_1 \neq z_1'$). Notice that for any fixed value of $z_1$, the function $z_2 \mapsto \frac{\partial c_{\mathbf{U}}([z_1, z_2])}{\partial z_1}$ is strictly decreasing. If $(z_1, z_2)$ and $(z_1, z_2')$ are both in the support, then by Lemma 10, it must be true that:

$$h_1(z_1) = \frac{\partial c_{\mathbf{U}}([z_1, z_2])}{\partial z_1} = \frac{\partial c_{\mathbf{U}}([z_1, z_2'])}{\partial z_1}.$$

However, since $z_2 \mapsto \frac{\partial c_{\mathbf{U}}([z_1, z_2])}{\partial z_1}$ is strictly decreasing, this means that $z_2 = z_2'$ as desired.

We can thus implicitly define the function $g$ by the (unique) value such that:

$$Q(w, g(w)) - h_1(w) = 0$$

where

$$Q(z_1, z_2) := \frac{\partial c_{\mathbf{U}}([z_1, z_2])}{\partial z_1}.$$

Uniqueness follows from the fact that $Q$ is a strictly decreasing function in its second argument, since $\frac{\partial Q(w, g(w))}{\partial z_2} = \frac{\partial^2 c_{\mathbf{U}}([w, g(w)])}{\partial z_1 \partial z_2} < 0$ as we showed previously. Since $h_1(w)$ is continuously differentiable and since:

$$\frac{\partial Q(w, g(w))}{\partial z_2} \neq 0$$

for $w \in I_1$, we can apply the implicit function theorem to see that $g(w)$ is continuously differentiable for $w \in I_1$.

We next show that $g$ is increasing on $I_1$. Within the interior of $I_1$, by Lemma 12 along with the fact that $\frac{\beta-2}{\beta}\cos(\theta^* - 2\theta) - \cos(\theta^*) < 0$ everywhere, we see that $g$ is a strictly increasing function on each contiguous portion of $I_1$. It thus suffices to show $I_1$ is an interval and that there are no gaps. If there is a gap, there must be a gap for both $z_1$ *and* $z_2$ at the same point $z$ since the support is on-to-one and closed. However, if $z$ is right above the gap, the producer would obtain higher utility by choosing $(1 - \epsilon)z$ for sufficiently small $\epsilon$ to ensure that $(1 - \epsilon)z$ is within the gap on both coordinates. This means that $I_1$ is an interval, which proves $g$ is an increasing function.

**Step 2: differential equation.** We show that

$$g'(w)g(w) - g'(w)w\cos(\theta^*) = w - g(w)\cos(\theta^*), \tag{28}$$

for all $w \in \text{supp}(H_1)$.

First, we derive the the condition that we described in equation (25) and further simplify it using that $g$ is increasing. Let $H_1^*(w) = H_1(w)^{\frac{1}{P-1}}$ and $H_2^*(w) = H_2(w)^{\frac{1}{P-1}}$. The densities $h_1^*$ and $h_2^*$ take the following form:

$$h_1^*(w) = (H_1^*)'(w) = \frac{1}{P-1}h_1(w)H_1(w)^{-\frac{P-2}{P-1}}$$

$$h_2^*(w) = (H_2^*)'(w) = \frac{1}{P-1}h_2(w)H_2(w)^{-\frac{P-2}{P-1}}.$$

In order for there to exist a distribution $\mu$ that satisfies condition (C2), it must hold that $H_1^*(w) = H_2^*(g(w))$ because $g$ is increasing. (This also means that $H_1(w) = H_2(g(w))$.) This means that $h_1^*(w) = h_2^*(g(w))g'(w)$ and $H_1(w) = H_2(g(w))$. Plugging this into the above expressions for $h_1^*$ and $h_2^*$, this means that:

$$h_1(w) = (P-1)h_1^*(w)H_1(w)^{\frac{P-2}{P-1}} = (P-1)g'(w)h_2^*(g(w))H_2(g(w))^{\frac{P-2}{P-1}} = h_2(w)g'(w).$$

This means that

$$g'(w) = \frac{h_1(w)}{h_2(w)} = \frac{w - g(w)\cos(\theta^*)}{g(w) - w\cos(\theta^*)},$$

where the last line follows from Lemma 10. This gives us the desired differential equation.

**Step 3: solving the differential equation.** We claim that the only valid solution to the differential equation (28) is $g(w) = w$. To see this, let $f(w) = \frac{g(w)}{w}$. This means that $wf(w) = g(w)$ and thus $f(w) + wf'(w) = g'(w)$. Plugging this into equation (28) and simplifying we obtain a separable differential equation. The solutions to this differential equation are $f(w) = 1$ and the following:

$$f_K^*(w) = K - \log(w) = \frac{1}{2}\left((1 + \cos(\theta^*))\log(1 + f(w)) - (1 + \cos(\theta^*))\log(1 - f(w))\right)$$

for some constant $K$. Notice that for $f_K^*$ to even be well-defined, we know that $f_K^*(w) < 1$ everywhere.

Assume for sake of contradiction that there exists an equilibrium with support given by $\{(w, g(w)) \mid w \in I\}$ for $g(w) \neq w$. Then we know that $g(w) = f_K^*(w) \cdot x$ for some $K$. In order for this solution to even be well-defined, it would imply that $f_K^*(w) < 1$ everywhere. This implies that $g(w) < w$, for all $w \in I_1$. However, we know that the function $g^{-1}$ must satisfy the differential equation too (and $g^{-1}(w) \neq w$), so by an analogous argument, we know that $g^{-1}(w) < w$ for all $w \in I_2$, which means that $w < g(w)$. This is a contradiction.

We can thus conclude that since $g(x) = x$, we have that $|\text{Genre}(\mu)| = 1$ as desired.

$\square$

To prove Proposition 10, we also leverage the machinery in Lemma 1. We use Lemma 10 to rule out all finite-genre equilibria except for two-genre equilibria. We can show that $H_1(w)$ and $H_2(w)$ grow proportionally to $w^\beta$. Then, we can implement this knowledge of $H_1$ and $H_2$ into the finite genre formulation of condition (C2) in equation (26) and show that no solutions to the functional equation exist for finite $P$. We formalize this below.

*Proof of Proposition 10.* By Claim 1, it suffices to focus on the case of 2 normalized users. We further assume WLOG that $u_1 = e_1$ and $u_2 = [\cos(\theta^*), \sin(\theta^*)]$. Since $\beta > \frac{2}{1 - \cos(\theta^*)}$, we know by Corollary 3 that there is no single-genre equilibrium. Assume for sake of contradiction that there exists a *finite*-genre equilibrium $\mu$ with $|\text{Genre}(\mu)| \geq 2$. By Lemma 1, we know that there exists $H_1, H_2$ and $S$ associated with $\mu$ that satisfy (C1)-(C3). Our proof boils down to two steps:

- *Step 1:* We show that $\text{Genre}(\mu) = \{\theta_1, \theta_2\}$ for some $\theta_1 < \theta^*/2 < \theta_2$.

- *Step 2:* We show that no two-genre distribution $\mu$ exists.

**Step 1.** Let us first translate the concept of genres to the reparameterized space. First, we consider the following set:

$$\text{Genre}_Z(S) := \left\{ \frac{1}{c_{\mathbf{U}}(z)} [z_1, z_2] \mid z \in S \right\}.$$

Since vectors in $\text{Genre}_Z(S)$ are of the form $[\cos(\theta), \cos(\theta^* - \theta)]$ by the normaalization by $c_{\mathbf{U}}(z)$, we can actually define a set of *angles*:

$$\text{Genre}_\Theta(S) := \left\{ \cos^{-1}(z_1) \mid [z_1, z_2] \in \text{Genre}_Z(S) \right\}.$$

We see that $\theta \in \text{Genre}_\Theta(S)$ if and only if $[\cos(\theta), \cos(\theta^* - \theta)] \in \text{Genre}_Z(S)$ if and only if $[\cos(\theta), \sin(\theta)] \in \text{Genre}(\mu)$. Elements of $\text{Genre}_\Theta(S)$ thus exactly corresponds to genres of $\text{Genre}(\mu)$.

We first observe that every $\theta \in \text{Genre}_\Theta(S)$ is in $(0, \theta^*)$. By (C1) of Lemma 1, the set $S$ must be contained in the convex cone of $[1, \cos(\theta^*)]$ and $[\cos(\theta^*, 1]$, which implies that $\theta \in [0, \theta^*]$. It thus suffices to show that $\theta \neq 0$ and $\theta \neq \theta^*$. We show that $\theta \neq 0$ (the case of $\theta \neq \theta*$ follows from an analogous argument). In this case, we see that there must be some set of the form $\{[r, r\cos(\theta^*)] \mid r \in \mathbb{R}_{\geq 0}\}$ that is subset of $S$. If $\theta^* = \pi/2$, then this would mean the distribution given by $H_2$ would have a point mass at 0, which is clearly not possible at equilibrium. Otherwise, if $\theta^* < \pi/2$, we apply (C1) and Lemma 10, and we see that $h_2(r\cos(\theta^*)) = 0$. However, this is a contradiction, since there is positive probability mass on some line segment on this genre by assumption.

Now, we observe that the support of the cdfs $H_1$ and $H_2$ must be bounded *intervals* of the form $[0, z_1^{\max}]$ and $[0, z_2^{\max}]$. First, we show that $\max(\text{supp}(H_1)), \max(\text{supp}(H_1)) < \infty$. By (C1), we see that a producer must achieve nonzero profit (since they always so $c_{\mathbf{U}}(z) \leq 2$, which means that $z_1, z_2 \leq \frac{2}{\alpha}$ as desired. This means that we can set $z_1^{\max} = \max(\text{supp}(H_1))$ and $z_2^{\max} = \max(\text{supp}(H_2))$. Next, we show that the supports of $H_1$ and $H_2$ contain the full intervals $[0, z_1^{\max}]$ and $[0, z_2^{\max}]$, respectively. Assume for sake of contradiction that the support of $H_1$ does not contain some interval $(x, x + \epsilon)$ for $\epsilon > 0$ within $[0, z_1^{\max}]$. Let $\epsilon$ be defined so that $z_1 = x + \epsilon \in \text{supp}(H_1)$. However, this means that there exists $z_2$ such that $[z_1, z_2] \in S$ and, moreover, $[z_1, z_2]$ must be located on a genre $\theta \in (0, \theta^*)$. We can thus reduce $z_1$ and hold $z_2$ fixed, while keeping $H_1(z_1) + H_2(z_2)$ fixed, and reducing the cost $c_{\mathbf{U}}(z)$, which violates the fact that $[z_1, z_2]$ is a maximizer of (10). An analogous argument shows that the support of $H_2$ is the full interval $[0, z_2^{\max}]$.

Next, we show that for $\theta, \theta' \in \text{Genre}_\Theta(S)$, it must hold that

$$\frac{\sin(\theta^* - \theta)}{\cos^{\beta-1}(\theta)} = \frac{\sin(\theta^* - \theta')}{\cos^{\beta-1}(\theta')} \quad \text{and} \quad \frac{\sin(\theta)}{\cos^{\beta-1}(\theta^* - \theta)} = \frac{\sin(\theta')}{\cos^{\beta-1}(\theta^* - \theta')} \tag{29}$$

To prove this, suppose that $|\text{Genre}_Z(S)| = G$ and label the genres by the indices $1, \ldots, G$ arbitrarily. For $z_1 \in \text{supp}(H_1)$ let $T(z_1) \subseteq \{1, \ldots, G\}$ be the set of genres $j$ where there exists $z_2$ such that $(z_1, z_2) \in S$ and $[z_1, z_2]$ points in the direction of $[\cos(\theta_j), \cos(\theta^* - \theta_j)]$. By Lemma 10, for all $i \in T(z_1)$, it must hold that:

$$h_1(z_1) = \beta z_1^{\beta-1} \alpha \cdot \frac{\sin(\theta^* - \theta_i)}{\sin(\theta^*)} \cdot \frac{1}{\cos(\theta_i)^{\beta-1}}.$$

This means that for $i, i' \in T(z_1)$, it holds that

$$\frac{\sin(\theta^* - \theta_i)}{\cos^{\beta-1}(\theta_i)} = \frac{\sin(\theta^* - \theta_{i'})}{\cos^{\beta-1}(\theta_{i'})}.$$

We now generalize this argument to arbitrary genres $\theta, \theta' \in \text{Genre}_\Theta(S)$. Consider $1 \leq i, i' \leq G$. Even though $\theta_i$ and $\theta_{i'}$ may not be in the same set $T(z_1)$, we show that there must be some "path" connecting $\theta_i$ and $\theta_{i'}$. To formalize this, for each genre $1 \leq i \leq G$, let $S_i = \{z_1 \in \text{supp}(H_1) \mid i \in T(z_i)\}$. Let's define an undirected graph vertices $[G]$ and an edge $(i_1, i_2)$ if and only if $S_{i_1} \cap S_{i_2} \neq \emptyset$. The argument from the previous paragraph showed that if there an edge between $i$ and $i'$, then $\frac{\sin(\theta^* - \theta_i)}{\cos^{\beta-1}(\theta_i)} = \frac{\sin(\theta^* - \theta_{i'})}{\cos^{\beta-1}(\theta_{i'})}$. Moreover, if there exists a path from $i$ to $i'$ in this graph, then we can chain together equalities along each edge in the path to prove $\frac{\sin(\theta^* - \theta_i)}{\cos^{\beta-1}(\theta_i)} = \frac{\sin(\theta^* - \theta_{i'})}{\cos^{\beta-1}(\theta_{i'})}$. The only remaining case is that there is no path from $i$ to $i'$. However, this would mean that the

vertices $[G]$ can be divided into a partition $P_1, \ldots, P_n$ for $n > 1$ such that there is no edge across partitions. Note that $\cup_{1 \le i \le G} S_i = \mathrm{supp}(H_1)$, which we already proved is equal to $[0, z_1^{\max}]$. Thus, this would mean that the disjoint, closed sets $\cup_{i \in P_1} S_i, \ldots, \cup_{i \in P_n} S_i$ have union equal to $[0, z_1^{\max}]$, which is not possible Sierpinski [1918]. Thus we have shown that $\frac{\sin(\theta^* - \theta_i)}{\cos^{\beta-1}(\theta_i)} = \frac{\sin(\theta^* - \theta_{i'})}{\cos^{\beta-1}(\theta_{i'})}$ for any $1 \le i, i' \le G$ and an analogous argument shows that $\frac{\sin(\theta_i)}{\cos^{\beta-1}(\theta^* - \theta_i)} = \frac{\sin(\theta_{i'})}{\cos^{\beta-1}(\theta^* - \theta_{i'})}$. This proves equation (29).

We next show that there exist exactly 2 genres given by $\theta_1 < \theta_2$. Using Lemma 11, we see that for any $\theta$, there are at most two values of $\theta' \ne \theta_1$ such that equation (29) can hold. Moreover, by Lemma 12, one of these values lies within the region where $g'$ would have to be negative (which is not possible). Thus, there are at most two genres, and Lemma 11 further tells us that they lie on opposite sides of $\theta^*/2$.

**Step 2.** Condition (C2) gives us functional equations that the distribution $\mu$ must satisfy for $P < \infty$. More specifically, let $F_1$ be the cdf of the magnitude of the genre given by $\theta_1$, and let $F_2$ be the cdf of the magnitude of the genre given by $\theta_2$. Then we obtain the following functional equations:

$$\left( \alpha_1 F_1 \left( \frac{z_1}{\cos(\theta_1)} \right) + \alpha_2 F_2 \left( \frac{z_1}{\cos(\theta_2)} \right) \right)^{P-1} = H_1(z_1)$$

$$\left( \alpha_1 F_1 \left( \frac{z_2}{\cos(\theta^* - \theta_1)} \right) + \alpha_2 F_2 \left( \frac{z_2}{\cos(\theta^* - \theta_2)} \right) \right)^{P-1} = H_2(z_1).$$

For these functional equations to be useful, we need to compute the cdfs $H_1$ and $H_2$. This will involve some notation: as in the previous step, let the genres be $\{\theta_1, \theta_2\}$ where $\theta_1 < \theta^*/2 < \theta_2$. Let $r_1^{\max} := \max(\mathrm{supp}(F_1))$ be the maximum value in the support of $F_1$ and let $r_2^{\max} := \max(\mathrm{supp}(F_2))$ be the maximum value in the support of $F_2$. We define:

$$i_1 := \arg\max_{i \in \{1,2\}} r_i \cos(\theta_i) \qquad i_2 := \arg\max_{i \in \{1,2\}} r_i \cos(\theta^* - \theta_i)$$

which correspond to which genre produces the highest value of $z_1$ and $z_2$ respectively.

We apply Lemma 10 to see that for all $z_1$ and $z_2$ in the support of $H_1$ and $H_2$, it holds that:

$$h_1(z_1) = \beta z_1^{\beta-1} \alpha \cdot \frac{\sin(\theta^* - \theta_{i_1})}{\sin(\theta^*)} \cdot \frac{1}{\cos(\theta_{i_1})^{\beta-1}}$$

$$h_1(z_2) = \beta z_2^{\beta-1} \alpha \cdot \frac{\sin(\theta_{i_2})}{\sin(\theta^*)} \cdot \frac{1}{\cos(\theta^* - \theta_{i_2})^{\beta-1}}.$$

We can integrate with respect to $z_1$ and $z_2$ to obtain that $H_1(z_1) = c_1 z_1^\beta$ and $H_1(z_2) = c_2 z_2^\beta$, such that:

$$c_1 = \alpha \cdot \frac{\sin(\theta^* - \theta_{i_1})}{\sin(\theta^*)} \cdot \frac{1}{\cos(\theta_{i_1})^{\beta-1}} \tag{30}$$

$$c_2 = \alpha \cdot \frac{\sin(\theta_{i_2})}{\sin(\theta^*)} \cdot \frac{1}{\cos(\theta^* - \theta_{i_2})^{\beta-1}}. \tag{31}$$

WLOG assume that $c_1 \ge c_2$ for the remainder of the analysis.

Using this specification of $H_1$ and $H_2$, we can write the functional equations as

$$\alpha_1 F_1 \left( \frac{z_1}{\cos(\theta_1)} \right) + \alpha_2 F_2 \left( \frac{z_1}{\cos(\theta_2)} \right) = c_1^{\frac{1}{P-1}} z_1^{\frac{\beta}{P-1}}$$

$$\alpha_1 F_1 \left( \frac{z_2}{\cos(\theta^* - \theta_1)} \right) + \alpha_2 F_2 \left( \frac{z_2}{\cos(\theta^* - \theta_2)} \right) = c_2^{\frac{1}{P-1}} z_2^{\frac{\beta}{P-1}}.$$

By taking a derivative with respect to $z_1$ and $z_2$, we see that for any $z_1$ within the support of $H_1$ and $z_2$ within the support of $H_2$, it holds that:

$$\frac{\alpha_1}{\cos(\theta_1)} f_1 \left( \frac{z_1}{\cos(\theta_1)} \right) + \frac{\alpha_2}{\cos(\theta_2)} f_2 \left( \frac{z_1}{\cos(\theta_2)} \right) = c_1^{\frac{1}{P-1}} \frac{\beta}{P-1} z_1^{\frac{\beta}{P-1}-1}. \tag{32}$$

$$\frac{\alpha_1}{\cos(\theta^* - \theta_1)} f_1\left(\frac{z_2}{\cos(\theta^* - \theta_1)}\right) + \frac{\alpha_2}{\cos(\theta^* - \theta_2)} f_2\left(\frac{z_2}{\cos(\theta^* - \theta_2)}\right) = c_2^{\frac{1}{P-1}} \frac{\beta}{P-1} z_2^{\frac{\beta}{P-1}-1}.$$

(33)

We prove that these functional equations have no valid solution. To show this, we prove that any solution to equations (32) and (33) would have negative density somewhere. Where the negative density occurs depends on $i_1$ and $i_2$.

We thus do casework on $i_1$ and $i_2$. In this analysis, we will use the notation $z_1^{\max}$ to denote $\max(\text{supp}(H_1))$ and $z_2^{\max}$ to denote $\max(\text{supp}(H_2))$. Note that by definition, $z_1^{\max} = r_{i_1}\cos(\theta_{i_1})$ and $z_2^{\max} = r_{i_2}\cos(\theta^* - \theta_{i_2})$.

First, we reduce the number of cases needed by using the fact that $c_1 \geq c_2$ (which we assumed earlier WLOG). In particular, this turns out to imply that $i_2 \neq 1$. More precisely, we show:

$$r_1^{\max}\cos(\theta^* - \theta_1) < r_2^{\max}\cos(\theta^* - \theta_2) \tag{34}$$

To show this, assume for sake of contradiction that $r_1^{\max}\cos(\theta^* - \theta_1) \geq r_2^{\max}\cos(\theta^* - \theta_2)$. Then we'd have that

$$z_2^{\max} = r_1^{\max}\cos(\theta^* - \theta_1) < r_1^{\max}\cos(\theta_1) \leq z_1^{\max}$$

which would imply that $c_1 < c_2$, which is a contradiction.

We thus split into 2 cases based on $i_1$.

- **Case 1**: $r_2^{\max}\cos(\theta_2) < r_1^{\max}\cos(\theta_1)$

- **Case 2**: $r_1^{\max}\cos(\theta_1) \leq r_2^{\max}\cos(\theta_2)$

Let's first handle **Case 1**. Since $z_1^{\max} = r_1^{\max}\cos(\theta_1) > r_2^{\max}\cos(\theta_2)$, we see that

$$\frac{z_1^{\max}}{\cos(\theta_2)} > r_2^{\max}$$

is not in the support of $F_2$. This means that the density $f_2$ of $F_2$ at $\frac{z_1^{\max}}{\cos(\theta_2)}$ is equal to 0 and, moreover, there exists $z_1^* < z_1^{\max}$ sufficiently close to $z_1^{\max}$ such that $z_1^*$ is in the support of $H_1$ and $\frac{z_1^*}{\cos(\theta_2)}$ is not in the support of $F_2$. At $z_1^*$, by equation (32), we see that:

$$\frac{\alpha_1}{\cos(\theta_1)} f_1\left(\frac{z_1^*}{\cos(\theta_1)}\right) = \frac{\alpha_1}{\cos(\theta_1)} f_1\left(\frac{z_1^*}{\cos(\theta_1)}\right) + \frac{\alpha_2}{\cos(\theta_2)} f_2\left(\frac{z_1^*}{\cos(\theta_2)}\right) = c_1^{\frac{1}{P-1}} \frac{\beta}{P-1} (z_1^*)^{\frac{\beta}{P-1}-1}.$$

Now, let's let $z_2^*$ be such that:

$$z_2^* := z_1^* \frac{\cos(\theta^* - \theta_1)}{\cos(\theta_1)}.$$

At $z_2^*$, we see that the left-hand side of equation (33) satisfies

$$\frac{\alpha_1}{\cos(\theta^* - \theta_1)} f_1\left(\frac{z_2^*}{\cos(\theta^* - \theta_1)}\right) + \frac{\alpha_2}{\cos(\theta^* - \theta_2)} f_2\left(\frac{z_2^*}{\cos(\theta^* - \theta_2)}\right)$$

$$\geq \frac{\alpha_1}{\cos(\theta^* - \theta_1)} f_1\left(\frac{z_2^*}{\cos(\theta^* - \theta_1)}\right)$$

$$= \frac{\cos(\theta_1)}{\cos(\theta^* - \theta_1)} \left(\frac{\alpha_1}{\cos(\theta_1)} f_1\left(\frac{z_1^*}{\cos(\theta_1)}\right)\right)$$

$$= \frac{\cos(\theta_1)}{\cos(\theta^* - \theta_1)} \left(c_1^{\frac{1}{P-1}} \frac{\beta}{P-1} (z_1^*)^{\frac{\beta}{P-1}-1}\right)$$

$$= \frac{\cos(\theta_1)}{\cos(\theta^* - \theta_1)} \left(c_1^{\frac{1}{P-1}} \frac{\beta}{P-1} \left(z_2^* \frac{\cos(\theta_1)}{\cos(\theta^* - \theta_1)}\right)^{\frac{\beta}{P-1}-1}\right)$$

$$= c_1^{\frac{1}{P-1}} (z_2^*)^{\frac{\beta}{P-1}-1} \frac{\beta}{P-1} \left(\frac{\cos(\theta_1)}{\cos(\theta^* - \theta_1)}\right)^{\frac{\beta}{P-1}}$$

$$> c_2^{\frac{1}{P-1}} \frac{\beta}{P-1} (z_2^*)^{\frac{\beta}{P-1}-1},$$

where the last inequality uses that $c_1 \geq c_2$ (which we assumed WLOG earlier) and $\theta_1 < \theta^*/2$. However, this is a contradiction since (33) must hold.

Let's next handle **Case 2**. By equation (34), we know that $z_2^{\max} = r_2^{\max} \cos(\theta^* - \theta_2) > r_1^{\max} \cos(\theta^* - \theta_1)$, so there exists $z_2$ such that $z_2 \in (r_1^{\max} \cos(\theta^* - \theta_1), z_2^{\max})$. At this value of $z_2$, we see by equation (33) that

$$\frac{\alpha_2}{\cos(\theta^* - \theta_2)} f_2 \left( \frac{z_2}{\cos(\theta^* - \theta_2)} \right) = c_2^{\frac{1}{P-1}} \frac{\beta}{P-1} z_2^{\frac{\beta}{P-1}-1}.$$

By assumption, we have that $z_1^{\max} = r_2^{\max} \cos(\theta_2) = \frac{z_2^{\max} \cos(\theta_2)}{\cos(\theta^* - \theta_2)}$ so $z_1 = \frac{z_2 \cos(\theta_2)}{\cos(\theta^* - \theta_2)}$ is in the support of $H_1$. By equation (32), for $z_1 = \frac{z_2 \cos(\theta_2)}{\cos(\theta^* - \theta_2)}$:

$$\frac{\alpha_2}{\cos(\theta_2)} f_2 \left( \frac{z_1}{\cos(\theta_2)} \right) \geq \frac{\alpha_1}{\cos(\theta_1)} f_1 \left( \frac{z_1}{\cos(\theta_1)} \right) + \frac{\alpha_2}{\cos(\theta_2)} f_2 \left( \frac{z_1}{\cos(\theta_2)} \right) = c_1^{\frac{1}{P-1}} \frac{\beta}{P-1} z_1^{\frac{\beta}{P-1}-1}.$$

Putting this all together, we see that:

$$
\begin{aligned}
c_1^{\frac{1}{P-1}} \frac{\beta}{P-1} z_1^{\frac{\beta}{P-1}-1} &\leq \frac{\alpha_2}{\cos(\theta_2)} f_2 \left( \frac{z_1}{\cos(\theta_2)} \right) = \frac{\alpha_2 \cos(\theta^* - \theta_2)}{\cos(\theta_2)} f_2 \left( \frac{z_2}{\cos(\theta^* - \theta_2)} \right) \\
&= c_2^{\frac{1}{P-1}} \frac{\cos(\theta^* - \theta_2)}{\cos(\theta_2)} \frac{\beta}{P-1} z_2^{\frac{\beta}{P-1}-1} \\
&= c_2^{\frac{1}{P-1}} \frac{\cos(\theta^* - \theta_2)}{\cos(\theta_2)} \frac{\beta}{P-1} \left( z_1 \frac{\cos(\theta^* - \theta_2)}{\cos(\theta_2)} \right)^{\frac{\beta}{P-1}-1} \\
&= c_2^{\frac{1}{P-1}} \frac{\beta}{P-1} z_1^{\frac{\beta}{P-1}-1} \left( \frac{\cos(\theta^* - \theta_2)}{\cos(\theta_2)} \right)^{\frac{\beta}{P-1}}
\end{aligned}
$$

This implies that:

$$\frac{c_1}{c_2} \leq \left( \frac{\cos(\theta^* - \theta_2)}{\cos(\theta_2)} \right)^{\beta}.$$

However, by equations (30) and (31), we also see that:

$$\frac{c_1}{c_2} = \frac{\sin(\theta^* - \theta_2)}{\sin(\theta_2)} \frac{\cos(\theta^* - \theta_2)^{\beta-1}}{\cos(\theta_2)^{\beta-1}} = \frac{\tan(\theta^* - \theta_2)}{\tan(\theta_2)} \frac{\cos(\theta^* - \theta_2)^{\beta}}{\cos(\theta_2)^{\beta}} > \frac{\cos(\theta^* - \theta_2)^{\beta}}{\cos(\theta_2)^{\beta}}, \quad (35)$$

where the last step uses that $\theta^* - \theta_2 < \theta_2$. This is a contradiction.

$\square$

### G.4 Proofs for Section D.2

We prove Proposition 7, restated below.

**Proposition 7.** *Suppose that there are 2 users located at the standard basis vectors $e_1, e_2 \in \mathbb{R}^2$, and the cost function is $c(p) = \|p\|_2^{\beta}$. For $P = 2$ and $\beta \geq \beta^* = 2$, there is an equilibrium $\mu$ supported on the quarter-circle of radius $(2\beta^{-1})^{1/\beta}$, where the angle $\theta \in [0, \pi/2]$ has density $f(\theta) = 2\cos(\theta)\sin(\theta)$.*

Conceptually speaking, the machinery given by Lemma 1 enables us to systematically identify the equilibrium in the concrete market instance of Proposition 7. Condition (C1) is simple along the quarter circle: by Lemma 10, the densities $h_1(u)$ and $h_2(v)$ are *proportional* to $u$ and $v$. Since the support of a single curve and $P = 2$, condition (C2) can be simplified to a clean condition on the densities $h_1$ and $h_2$ given by (24).

To actually prove Proposition 7, we only need to *verify* that the equilibrium $\mu$ in Proposition 7 which is easier.

*Proof.* By Lemma 1, it suffices to prove that (C1)-(C3) hold for $H_1$, $H_2$, and $S$ associated with the distribution $\mu$ in the statement of the proposition. Conditions (C2) and (C3) follow by construction of $\mu$, so it suffices to prove (C1).

First, we claim that

$$H_1(z_1) = \left(\frac{2}{\beta}\right)^{-2/\beta} z_1^2, \text{ and } H_2(z_2) = \left(\frac{2}{\beta}\right)^{-2/\beta} z_2^2.$$

We show that $H_2(z_2) = \left(\frac{2}{\beta}\right)^{-2/\beta} z_2^2$ (an analogous argument applies to $H_1$). We see that $H_2$ is supported on $\left[0, \left(\frac{2}{\beta}\right)^{1/\beta}\right]$ by construction, so it suffices to show that

$$h_2(z_2) = 2\left(\frac{2}{\beta}\right)^{-2/\beta} z_2$$

on this interval. Since $z_2 = \left(\frac{2}{\beta}\right)^{1/\beta} \sin(\theta)$, by the change of variables formula for $P = 2$, we see that

$$h_2(z_2) \left(\frac{2}{\beta}\right)^{1/\beta} \cos(\theta) = f(\theta) = 2\sin(\theta)\cos(\theta).$$

We can solve and obtain:

$$h_2(z_2) = 2\left(\frac{2}{\beta}\right)^{-1/\beta} \sin(\theta) = 2\left(\frac{2}{\beta}\right)^{-2/\beta} z_2,$$

as desired.

Now, we prove (C1). Applying Lemma 9, we see that:

$$H_1(z_1) + H_2(z_2) - c_{\mathbf{U}}(z) = \left(\min\left(\left(\frac{2}{\beta}\right)^{-2/\beta} z_1^2, 1\right) + \min\left(\left(\frac{2}{\beta}\right)^{-2/\beta} z_2^2, 1\right)\right) - (z_1^2 + z_2^2)^{\beta/2}.$$

Thus, equation (10) can be written as:

$$\max_{z_1, z_2 \geq 0} \left(\left(\min\left(\left(\frac{2}{\beta}\right)^{-2/\beta} z_1^2, 1\right) + \min\left(\left(\frac{2}{\beta}\right)^{-2/\beta} z_2^2, 1\right)\right) - (z_1^2 + z_2^2)^{\beta/2}\right) \quad (36)$$

We wish to show equation (36) is maximized whenever $z \in S$. Since $z_1^2 + z_2^2 = \left(\frac{2}{\beta}\right)^{2/\beta}$ for any $z \in S$, this follows from Lemma 13. $\qquad\square$

We prove Proposition 8, restated below.

**Proposition 8.** *Suppose that there are 2 users located at the standard basis vectors $e_1, e_2 \in \mathbb{R}^2$, with cost function $c(p) = \|p\|_2^\beta$. For $\beta = 2$, there is a multi-genre equilibrium $\mu$ with support equal to*

$$\left\{\left(x, (1 - x^{\frac{2}{P-1}})^{\frac{P-1}{2}}\right) \mid x \in [0, 1]\right\}, \quad (9)$$

*and where the distribution of $x$ has cdf equal to $\min(1, x^{2/(P-1)})$.*

Again, the machinery given by Lemma 1 enables us to systematically identify the equilibrium in the concrete market instance of Proposition 8. Since we need to consider $P \neq 2$, the condition (C2) does not take as clean of a form: as shown by (25), it depends on both the densities $h_1$ and $h_2$ along with the cdfs $H_1$ and $H_2$. Nonetheless, in the special case of $\beta = 2$, we can compute the cdf in closed-form: Lemma 10 implies that the density $h_1(z_1)$ is entirely specified by $z_1$ and does not depend on $z_2$, so we can integrate over the density to explicitly compute the cdf. We can obtain the equilibria in Proposition 8 as a solution to a differential equation.

To prove Proposition 7, we again only need to *verify* that the equilibrium $\mu$ in Proposition 8 which is easier.

*Proof of Proposition 8.* By Lemma 1, it suffices to prove that (C1)-(C3) hold for $H_1$, $H_2$, and $S$ for the distribution $\mu$ given in the statement of the proposition. Conditions (C2) and (C3) follow by construction of $\mu$, so it suffices to prove (C1).

First, we claim that $H_1(z_1) = z_1^2$ and $H_1(z_1) = z_2^2$. We see that since the cdf of $p_1$ for $p \sim \mu$ is $z_1$, we know that $H_1(z_1) = z_1^2$ by construction. For $z_2$, first we note that the cdf of $p_2$ for $p_2 \sim \mu$ is given by:

$$\mathbb{P}_{p_2 \sim \mu}[p_2 \leq p_2'] = \mathbb{P}_{p_1 \sim \mu}\left[p_1 \geq (1 - (p_2')^{\frac{2}{P-1}})^{\frac{P-1}{2}}\right] = 1 - (1 - (p_2')^{\frac{2}{P-1}}) = (p_2')^{\frac{2}{P-1}}.$$

By definition, this means that $H_2(z_2) = z_2^2$ as desired.

Now, we prove (C1). Applying Lemma 9, we see that:

$$H_1(z_1) + H_2(z_2) - c_{\mathbf{U}}(z) = \left(\min\left(z_1^2, 1\right) + \min\left(z_2^2, 1\right)\right) - (z_1^2 + z_2^2).$$

Thus, equation (10) can be written as:

$$\max_{z_1, z_2 \geq 0} \left(\min\left(z_1^2, 1\right) + \min\left((z_2^2, 1)\right) - (z_1^2 + z_2^2)^{\beta/2}\right) \tag{37}$$

We wish to show equation (37) is maximized whenever $z \in S$. Since $z_1^2 + z_2^2 = 1$ for any $z \in S$, this follows from Lemma 13 applied to $\beta = 2$. □

### G.5 Formalization of the infinite-producer limit

Since our characterization result (Theorem 3) focuses on finite-genre equilibria, we restrict our formal definition of the infinite-producer limit to case of finite genres for technical convenience.

We arrive at a formalism by taking a limit of the conditions in Lemma 1 as $P \to \infty$. Let $\mu$ be a finite-genre distribution over $\mathbb{R}_{\geq 0}^D$. We can specify $\mu$ by the three attributes: the genres $d_1, \ldots, d_G$, the distributions $F_g$ over $\mathbb{R}_{\geq 0}$ corresponding to the distribution of $\|p\|$ for $p$ drawn from $\mu$ conditioned on $p$ pointing in the direction of $d_g$, and the weights $\alpha_g$ corresponding to the probability that $p \sim \mu$ points in the direction of $d_g$. In particular, $\mu$ can be described as follows: with probability $\alpha_g$, choose the vector $q_g d_g$ where $q_g$ is drawn from a distribution with cdf $F_g$. We see that the corresponding function $H_i$ from Lemma 1 will be equal to:

$$H_i(z_i) = \left(\sum_{g=1}^{G} \alpha_g F_g\left(\frac{\langle u_i, p\rangle}{\langle u_i, d_g\rangle}\right)\right)^{P-1}.$$

Note that the conditions (C2) and (C3) are essentially satisfied by construction; condition (C1) requires that

$$\max_{p \in \mathbb{R}_{\geq 0}^D} \left(\sum_{i=1}^{N} H_i(\langle u_i, p\rangle) - c(p)\right)$$

is maximized for any $p \in \mathrm{supp}(\mu)$. This can be rewritten as requiring that any $p^* \in \mathrm{supp}(\mu)$ satisfies:

$$p^* \in \arg\max_{p \in \mathbb{R}_{\geq 0}^D} \left(\sum_{i=1}^{N} \left(\sum_{g=1}^{G} \alpha_g F_g\left(\frac{\langle u_i, p\rangle}{\langle u_i, d_g\rangle}\right)\right)^{P-1} - c(p)\right).$$

Let's rewrite this in terms of winning producers: more formally, let $F_g^{\max}(\cdot) = (F_g(\cdot))^{P-1}$ denote the cumulative distribution function of the *maximum* quality in a genre, conditioned on all producers choosing that genre. We call the distributions $F_1^{\max}, \ldots, F_G^{\max}$ the *conditional quality distributions*. Then we obtain the following:

$$p^* \in \arg\max_{p \in \mathbb{R}_{\geq 0}^D} \left(\sum_{i=1}^{N} \left(\sum_{g=1}^{G} \alpha_g \left(F_g^{\max}\left(\frac{\langle u_i, p\rangle}{\langle u_i, d_g\rangle}\right)\right)^{1/(P-1)}\right)^{P-1} - c(p)\right). \tag{38}$$

Taking a limit as $P \to \infty$, we see that

$$\left(\sum_{g=1}^{G} \alpha_g F_g\left(\frac{\langle u_i, p\rangle}{\langle u_i, d_g\rangle}\right)^{1/(P-1)}\right)^{P-1} \to \prod_{g=1}^{G} \left(F_g\left(\frac{\langle u_i, p\rangle}{\langle u_i, d_g\rangle}\right)\right)^{\alpha_i}.$$

Thus, equation 38 (informally speaking) approaches the following condition in the limit:

$$p^* \in \underset{p \in \mathbb{R}_{\geq 0}^D}{\arg\max} \left( \sum_{i=1}^N \left( \sum_{g=1}^G \alpha_g F_g \left( \frac{\langle u_i, p \rangle}{\langle u_i, d_g \rangle} \right)^{1/(P-1)} \right)^{P-1} - c(p) \right). \tag{39}$$

Motivated by equation 39, we specify $\mu$ by three attributes—the *genres* $d_1, \ldots, d_G$, the *conditional quality* distributions $F_g^{\max}$ over $\mathbb{R}_{\geq 0}$, and the *weights* $\alpha_g$ corresponding to the probability that $p \sim \mu$ points in the direction of a given genre—as follows.

**Definition 1** (Finite-genre equilibria for $P = \infty$). *Let $u_1, \ldots, u_N \in \mathbb{R}_{\geq 0}^D$ be a set of users and let $c(p) = \|p\|_2^\beta$ be the cost function. A set of genres $d_1, \ldots, d_G \in \mathbb{R}_{\geq 0}^D$ such that $\|d_i\|_2 = 1$ for all $1 \leq g \leq G$, a set of conditional quality distributions $F_1, F_2, \ldots, F_G$ over $\mathbb{R}_{\geq 0}$, and a set of weights $\alpha_1, \ldots, \alpha_G \geq 0$ such that $\sum_{g=1}^G \alpha_g = 1$ forms a finite-genre equilibrium if the following condition holds for*

$$p^* \in \underset{p \in \mathbb{R}_{\geq 0}^D}{\arg\max} \left( \sum_{i=1}^N \left( \prod_{g=1}^G \left( F_g^{max} \left( \frac{\langle u_i, p \rangle}{\langle u_i, d_g \rangle} \right) \right)^{\alpha_i} \right) - c(p) \right) \tag{40}$$

*for any $p^* = q_i d_i$ such that $1 \leq i \leq G$ and $q_i \in supp(F_i)$.*

Using the formalization in Definition 1 of equilibria for $P = \infty$, we investigate the case of two homogeneous populations of users, and we characterize two-genre equilibria.

**Theorem 4.** *[Formal version of Theorem 3] Suppose that there are 2 users located at two linearly independently vectors $u_1, u_2 \in \mathbb{R}_{\geq 0}^D$, let $\theta^* := \cos^{-1}\left( \frac{\langle u_1, u_2 \rangle}{\|u_1\|_2 \|u_2\|_2} \right) < 0$ be the angle between them. Suppose we have cost function $c(p) = \|p\|_2^\beta$, $\beta > \beta^* = \frac{2}{1 - \cos(\theta^*)}$, and $P = \infty$ producers. Then, the genres $d_1, d_2$, conditional quality distributions $F_1^{max} = F^{max}$ and $F_2^{max} = F^{max}$, and weights $\alpha_1 = \alpha_2 = 2$ form an equilibrium (as per Definition 1), where*

$$\{d_1, d_2\} := \left\{ [\cos(\theta^G + \theta_{min}), \sin(\theta^G + \theta_{min})], [\cos(\theta^* - \theta^G + \theta_{min}), \sin(\theta^* - \theta^G + \theta_{min})] \right\}$$

*such that $\theta^G := \arg\max_{\theta \leq \theta^*/2} \left( \cos^\beta(\theta) + \cos^\beta(\theta^* - \theta) \right)$ and $\theta_{min} := \min\left( \cos^{-1}\left( \frac{\langle u_1, e_1 \rangle}{\|u_1\|} \right), \cos^{-1}\left( \frac{\langle u_2, e_1 \rangle}{\|u_2\|} \right) \right)$, and where*

$$F^{max}(q) := \begin{cases} C_2^{(2n+2)\beta} & \text{if } q \in C_1^{1/\beta} C_2^{2n+1} [C_2, 1] \text{ for } n \geq 0 \\ C_1^{-2} C_2^{-2n\beta} q^{2\beta} & \text{if } q \in C_1^{1/\beta} C_2^{2n} [C_2, 1] \text{ for } n \geq 0 \\ 1 & \text{if } q \geq C_1^{1/\beta}, \end{cases},$$

*such that the constants are defined by $C_1 := \frac{\sin(\theta^*) \cos(\theta^G)}{\sin(\theta^* - \theta_G)}$ and $C_2 := \frac{\cos(\theta^* - \theta^G)}{\cos(\theta^G)}$.*

### G.6 Proofs for Section D.3

To recover the equilibrium in the infinite-producer limit, we need to show that there exists a two-genre equilibrium and find this equilibrium. We can apply machinery that is conceptually similar to Lemma 1 enables us to systematically identify the particular equilibrium within the family of two-genre equilibrium. The first-order condition (Lemma 10) given by condition (C1) helps identify the location of the genre directions, and this further enables us to compute the cdfs $H_1$ and $H_2$. At this stage, the proof boils down to solving for the conditional quality distributions $F_1$ and $F_2$. We obtain an infinite-producer limit of the functional equations in (26) which can be solved directly.

To actually prove Theorem 4, we again only need to *verify* that the equilibrium $\mu$ in Theorem 4 which is easier.

*Proof of Theorem 4.* WLOG, we assume that $\|u_1\| = \|u_2\| = 1$. It suffices to verify that the genres, conditional quality distributions, and weights satisfy (40). Motivated by Lemma 1, we define:

$$H_1(z_1) = \sqrt{F_1^{\max}\left(\frac{z_1}{\langle u_1, d_1\rangle}\right) F_2^{\max}\left(\frac{z_1}{\langle u_1, d_2\rangle}\right)}$$

$$H_2(z_2) = \sqrt{F_1^{\max}\left(\frac{z_2}{\langle u_1, d_1\rangle}\right) F_2^{\max}\left(\frac{z_2}{\langle u_1, d_2\rangle}\right)}.$$

We define the support $S$ to be

$$S := \{[\langle u_1, qd_1\rangle, \langle u_2, qd_1\rangle] \mid q_1 \in \operatorname{supp}(F_1^{\max})\} \cup \{[\langle u_1, qd_1\rangle, \langle u_2, qd_1\rangle] \mid q_2 \in \operatorname{supp}(F_2^{\max})\}.$$

Using this notation, we can rewrite (40) as requiring that:

$$\max_z \left(H_1(z_1) + H_2(z_2) - c_{\mathbf{U}}(z)\right) \tag{41}$$

is maximized for every $z \in S$.

First, we show that

$$\sin(\theta^G)\cos^{\beta-1}(\theta^G) = \sin(\theta^* - \theta^G)\cos^{\beta-1}(\theta^* - \theta^G) \tag{42}$$

This immediately follows from using that $\theta^G \in \arg\max_\theta \left(\cos^\beta(\theta) + \cos^\beta(\theta^* - \theta)\right)$ and applying the first-order condition.

For the remainder of the proof, we define:

$$c := \frac{\sin(\theta^* - \theta^G)}{\sin(\theta^*)\cos^{\beta-1}(\theta^G)} = \frac{\sin(\theta^G)}{\sin(\theta^*)\cos^{\beta-1}(\theta^* - \theta^G)},$$

**Computing $H_1$ and $H_2$.** We show that:

$$H_1(z_1) = \min\left(cz_1^\beta, 1\right) \text{ and } H_2(z_2) = \min\left(1, cz_2^\beta\right).$$

We show that

$$H_1(z_1) = \min\left(cz_1^\beta, 1\right), \tag{43}$$

and observe that the expression for $H_2$ follows from an analogous argument. By definition, we see that:

$$H_1(z_1) = \sqrt{F_1\left(\frac{z_1}{\langle u_1, d_1\rangle}\right) F_2\left(\frac{z_1}{\langle u_1, d_2\rangle}\right)}$$

$$= \sqrt{F\left(\frac{z_1}{\langle u_1, d_1\rangle}\right) F\left(\frac{z_1}{\langle u_1, d_2\rangle}\right)}.$$

We know that either (1) $\langle u_1, d_1\rangle = \langle u_2, d_2\rangle = \cos(\theta^G)$ and $\langle u_1, d_2\rangle = \langle u_2, d_1\rangle = \cos(\theta^* - \theta^G)$, or (2) $\langle u_1, d_2\rangle = \langle u_2, d_1\rangle = \cos(\theta^G)$ and $\langle u_1, d_1\rangle = \langle u_2, d_2\rangle = \cos(\theta^* - \theta^G)$. WLOG, we assume that (1) holds. This means that:

$$H_1(z_1) = \sqrt{F^{\max}\left(\frac{z_1}{\langle u_1, d_1\rangle}\right) F^{\max}\left(\frac{z_1}{\langle u_1, d_2\rangle}\right)}$$

$$= \sqrt{F^{\max}\left(\frac{z_1}{\cos(\theta^G)}\right) F^{\max}\left(\frac{z_1}{\cos(\theta^* - \theta^G)}\right)}$$

Let's reparameterize and let:

$$q_1 = \frac{z_1}{\cos(\theta^* - \theta^G)}.$$

This means that:

$$H_1(q_1\cos(\theta^* - \theta^G)) = \sqrt{F^{\max}(q_1)F^{\max}\left(q_1\frac{\cos(\theta^* - \theta^G)}{\cos(\theta^G)}\right)}.$$

Equation (43) reduces to

$$\sqrt{F^{\max}(q_1)F^{\max}\left(q_1\frac{\cos(\theta^*-\theta^G)}{\cos(\theta^G)}\right)} = \min\left(1, c\cos(\theta^*-\theta^G)^\beta q_1^\beta\right).$$

which simplifies to

$$\sqrt{F^{\max}(q_1)F^{\max}\left(q_1\frac{\cos(\theta^*-\theta^G)}{\cos(\theta^G)}\right)} = \min\left(1, \frac{\sin(\theta^G)\cos(\theta^*-\theta^G)}{\sin(\theta^*)}q_1^\beta\right)$$

which simplifies to

$$\sqrt{F^{\max}(q_1)F^{\max}(q_1C_2)} = \min\left(1, C_3^{-1}q_1^\beta\right) \tag{44}$$

We verify equation (44) by doing casework on $q_1$. Note that $C_1^{1/\beta} = C_3^{1/\beta}C_2$. If $q_1 \geq C_3^{1/\beta}C_2^{-1/\beta}$, then we see that $F^{\max}(q_1) = F^{\max}(q_1C_2) = 1$ and the equation holds. In fact, if $q_1 \geq C_3^{1/\beta}C_2^{1-1/\beta}$, then we see that $F^{\max}(q_1) = 1$ and

$$F^{\max}(q_1C_2) = C_3^{-2}C_2^{2-2\beta}(q_1C_2)^{2\beta} = C_3^{-2}C_2^2 q_1^{2\beta}$$

, so equation (44) is satisfied. Otherwise, if $q_1 = C_3^{1/\beta}C_2 C_2^{2n}\gamma$ for $n \geq 0$ and $\gamma \in [C_2, 1]$, then

$$F^{\max}(q_1) = C_3^{-2}C_2^{-2\beta-2n\beta}q^{2\beta}$$

and

$$F^{\max}(q_1C_2) = C_2^{(2n+2)\beta},$$

so:

$$\sqrt{F^{\max}(q_1)F^{\max}(q_1C_2)} = \sqrt{C_3^{-2}C_2^{-2\beta-2n\beta}C_2^{(2n+2)\beta}} = \sqrt{C_3^{-2}C_2^2 q^{2\beta}} = C_3^{-1}C_2 q_1^\beta$$

as desired. Finally, if $q_1 = C_1^{1/\beta}C_2^{1-1/\beta}C_2^{2n+1}\gamma$ for $n \geq 0$ and $\gamma \in [C_2, 1]$, then

$$F^{\max}(q_1) = C_2^{(2n+2)\beta}$$

and

$$F^{\max}(q_1C_2) = C_3^{-2}C_2^{-(2n+4)\beta}q^{2\beta},$$

so:

$$\sqrt{F^{\max}(q_1)F^{\max}(q_1C_2)} = \sqrt{C_2^{(2n+2)\beta}C_3^{-2}C_2^{(2n+4)\beta}q^{2\beta}C_2^{2\beta}} = C_3^{-1}q^\beta.$$

This proves the desired formulas for $H_1$ and an analogous argument applies to $H_2$.

**Showing equation (41) is maximized at every $z \in S$.** We need to show that for every $z \in S$, it holds that:

$$H_1(z_1) + H_2(z_2) - c_{\mathbf{U}}(z) = \max_{z'}(H_1(z_1') + H_2(z_2') - c_{\mathbf{U}}(z')).$$

Plugging in our expressions above, our goal is to show:

$$\min(1, cz_1^\beta) + \min(1, cz_2^\beta) - c_{\mathbf{U}}(z) = \max_{z'}(H_1(z_1') + H_2(z_2') - c_{\mathbf{U}}(z'))$$

for every $z \in S$.

We split into two steps: first, we show that

$$\min(1, cz_1^\beta) + \min(1, cz_2^\beta) - c_{\mathbf{U}}(z) = 0 \tag{45}$$

for every $z \in S$, and next we show that:

$$\max_{z'}(H_1(z_1') + H_2(z_2') - c_{\mathbf{U}}(z')) \leq 0. \tag{46}$$

To show (45), let's first consider $[z_1, z_2] = [r\cos(\theta^G), r\cos(\theta^G - \theta^*)] \in S$. Then we see that:

$$\min(1, cz_1^\beta) + \min(1, cz_2^\beta) - c_{\mathbf{U}}(z) = cz_1^\beta + cz_2^\beta - c_{\mathbf{U}}(z)$$
$$= r^\beta\left(c\cos^\beta(\theta^G) + c\cos^\beta(\theta^* - \theta^G) - 1\right)$$

Thus, it suffices to show that:

$$\cos^\beta(\theta^G) + \cos^\beta(\theta^* - \theta^G) = \frac{1}{c}. \tag{47}$$

We now show equation (47):

$$\cos^\beta(\theta^G) + \cos^\beta(\theta^* - \theta^G) =_{(A)} \frac{\cos(\theta^G)\cos^{\beta-1}(\theta^* - \theta^G)\sin(\theta^* - \theta^G)}{\sin(\theta^G)} + \cos^\beta(\theta^* - \theta^G)$$

$$= \frac{\cos^{\beta-1}(\theta^* - \theta^G)}{\sin(\theta^G)}\left(\cos(\theta^G)\sin(\theta^* - \theta^G) + \cos(\theta^* - \theta^G)\sin(\theta^G)\right)$$

$$= \frac{\cos^{\beta-1}(\theta^* - \theta^G)}{\sin(\theta^G)}\sin(\theta^*)$$

$$= \frac{1}{c}.$$

where (A) follows from applying equation (42). Let's now consider let's first consider $[z_1, z_2] = [r\cos(\theta^G - \theta^*), r\cos(\theta^*)] \in S$. Then, we see that

$$\min(1, cz_1^\beta) + \min(1, cz_2^\beta) - c_U(z) = cz_1^\beta + cz_2^\beta - c_U(z) = r^\beta\left(c\cos^\beta(\theta^G) + c\cos^\beta(\theta^* - \theta^G) - 1\right) = 0,$$

where the last equality follows from equation (47). This establishes equation (46).

Now, we show equation (46). Let's represent $z'$ as $U[r'\cos(\theta), r'\sin(\theta)]$. Then this becomes:

$$c(r')^\beta\cos^\beta(\theta) + c(r')^\beta\cos^\beta(\theta^* - \theta) \le (r')^\beta.$$

Dividing by $r'^\beta$, we obtain:

$$\cos^\beta(\theta) + \cos^\beta(\theta^* - \theta) \le \frac{1}{c}.$$

To show this, observe that:

$$\cos^\beta(\theta) + \cos^\beta(\theta^* - \theta) \le \cos^\beta(\theta^G) + \cos^\beta(\theta^* - \theta^G) = \frac{1}{c}.$$

where the first inequality follows from the fact that $\theta^G$ is a maximizer of $\cos^\beta(\theta) + \cos^\beta(\theta^* - \theta)$ by definition, and the second equality follows from equation (47). This establishes equation (46).

This proves that equation (47) is maximized at every $z \in S$, and thus the conditions of Definition 1 are satisfied. □

### G.7 Proofs of auxiliary lemmas

We state and prove Lemma 13, a lemma which we used in the proofs of Proposition 8 and Proposition 7.

**Lemma 13.** *For any $\beta \ge 2$, the expression*

$$\max_{z_1, z_2 \ge 0}\left(\left(\min\left(\left(\frac{2}{\beta}\right)^{-2/\beta}z_1^2, 1\right) + \min\left(\left(\frac{2}{\beta}\right)^{-2/\beta}z_2^2, 1\right)\right) - (z_1^2 + z_2^2)^{\beta/2}\right)$$

*is maximized for any $(z_1, z_2)$ such that $z_1^2 + z_2^2 = \left(\frac{2}{\beta}\right)^{2/\beta}$.*

*Proof.* First, for $z_1, z_2$ such that $z_1^2 + z_2^2 = \left(\frac{2}{\beta}\right)^{2/\beta}$, we have that

$$\left(\frac{2}{\beta}\right)^{-2/\beta}\left(z_1^2 + z_2^2\right) - (z_1^2 + z_2^2)^{\beta/2} = 1 - \frac{2}{\beta}.$$

It thus suffices to prove that:

$$\left(\min\left(\left(\frac{2}{\beta}\right)^{-2/\beta}z_1^2, 1\right) + \min\left(\left(\frac{2}{\beta}\right)^{-2/\beta}z_2^2, 1\right)\right) - (z_1^2 + z_2^2)^{\beta/2} \le 1 - \frac{2}{\beta}$$

for any $z_1, z_2 \geq 0$. It suffices to prove the stronger statement that:

$$\left(\frac{2}{\beta}\right)^{-2/\beta} (z_1^2 + z_2^2) - (z_1^2 + z_2^2)^{\beta/2} \leq 1 - \frac{2}{\beta}$$

Let $c = z_1^2 + z_2^2$; then we can rewrite the desired condition as:

$$\max_{c \geq 0} \left( \left(\frac{2}{\beta}\right)^{-2/\beta} c^2 - c^\beta \right) \leq 1 - \frac{2}{\beta}.$$

A first-order condition tells us for $\beta \geq 2$, that $\left(\frac{2}{\beta}\right)^{-2/\beta} c^2 - c^\beta$ is maximized at $c = \left(\frac{2}{\beta}\right)^{1/\beta}$, which proves the desired statement. $\qquad\square$

We prove Lemma 9.

*Proof of Lemma 9.* It suffices to show that if $z_1 = \langle u_1, p \rangle$ and $z_2 = \langle u_2, p \rangle$, then:

$$\|p\|^2 = \frac{z_1^2 + z_2^2 - 2z_1 z_2 \cos(\theta^*)}{\sin^2(\theta^*)} \tag{48}$$

WLOG, let $u_1 = e_1$ and let $u_2 = [\cos(\theta^*), \sin(\theta^*)]$. We see that:

$$\frac{z_1^2 + z_2^2 - 2z_1 z_2 \cos(\theta^*)}{\sin^2(\theta^*)} = \frac{p_1^2 + (p_1 \cos(\theta^*) + p_2 \sin(\theta^*))^2 - 2p_1(p_1 \cos(\theta^*) + p_2 \sin(\theta^*)) \cos(\theta^*)}{\sin^2(\theta^*)}$$

$$= \frac{p_1^2 \sin^2(\theta^*) + p_2^2 \sin^2(\theta^*)}{\sin^2(\theta^*)}$$

$$= p_1^2 + p_2^2$$

$$= \|p\|_2^2,$$

which proves equation (48). $\qquad\square$

We prove Lemma 10.

*Proof of Lemma 10.* Since $\mu$ is a symmetric mixed equilibrium, $z$ must be a maximizer of equation (10). The equation

$$\begin{bmatrix} h_1(z_1) \\ h_2(z_2) \end{bmatrix} = \nabla_z(c_\mathbf{U}(z))$$

is the first-order condition and thus holds for every $z$ is in the support of $\mu$.

Next, we show that:

$$\nabla_z(c_\mathbf{U}(z)) = \beta \alpha^\beta \sin^{-\beta}(\theta^*) \left( (z_1^2 + z_2^2 - 2z_1 z_2 \cos(\theta^*))^{\frac{\beta}{2}-1} \right) \begin{bmatrix} z_1 - z_2 \cos(\theta^*) \\ z_2 - z_1 \cos(\theta^*) \end{bmatrix}.$$

By applying Lemma 9, we see that:

$$\nabla_z(c_\mathbf{U}(z)) = \nabla_z \left( \alpha^\beta \sin^{-2\beta}(\theta^*) (z_1^2 + z_2^2 - 2z_1 z_2 \cos(\theta^*))^{\frac{\beta}{2}} \right)$$

$$= \alpha^\beta \sin^{-\beta}(\theta^*) \cdot \nabla_z \left( (z_1^2 + z_2^2 - 2z_1 z_2 \cos(\theta^*))^{\frac{\beta}{2}} \right)$$

$$= \beta \alpha^\beta \sin^{-\beta}(\theta^*) \left( (z_1^2 + z_2^2 - 2z_1 z_2 \cos(\theta^*))^{\frac{\beta}{2}-1} \right) \begin{bmatrix} z_1 - z_2 \cos(\theta^*) \\ z_2 - z_1 \cos(\theta^*) \end{bmatrix},$$

as desired.

Finally, we show that

$$\nabla_z(c_\mathbf{U}(z)) = \beta \alpha^\beta \sin^{-\beta}(\theta^*) (z_1^2 + z_2^2 - 2z_1 z_2 \cos(\theta^*))^{\frac{\beta}{2}-1} \begin{bmatrix} z_1 - z_2 \cos(\theta^*) \\ z_2 - z_1 \cos(\theta^*) \end{bmatrix}.$$

We see that:

$$\nabla_z(c_{\mathbf{U}}(z)) = \beta\alpha^\beta \sin^{-\beta}(\theta^*)\left(\left(z_1^2 + z_2^2 - 2z_1z_2\cos(\theta^*)\right)^{\frac{\beta}{2}-1}\right)\begin{bmatrix} z_1 - z_2\cos(\theta^*) \\ z_2 - z_1\cos(\theta^*) \end{bmatrix}$$

$$= \beta\alpha^\beta r^{\beta-2}\begin{bmatrix} \frac{z_1 - z_2\cos(\theta^*)}{\sin^2(\theta^*)} \\ \frac{z_2 - z_1\cos(\theta^*)}{\sin^2(\theta^*)} \end{bmatrix}$$

$$= \beta\alpha^\beta r^{\beta-1}\begin{bmatrix} \frac{\cos(\theta) - \cos(\theta^*-\theta)\cos(\theta^*)}{\sin^2(\theta^*)} \\ \frac{\cos(\theta^*-\theta) - \cos(\theta)\cos(\theta^*)}{\sin^2(\theta^*)} \end{bmatrix}$$

$$= \beta\alpha^\beta r^{\beta-1}\begin{bmatrix} \frac{\cos(\theta*-(\theta^*-\theta)) - \cos(\theta^*-\theta)\cos(\theta^*)}{\sin^2(\theta^*)} \\ \frac{\sin(\theta^*)\sin(\theta)}{\sin^2(\theta^*)} \end{bmatrix}$$

$$= \beta\alpha^\beta r^{\beta-1}\begin{bmatrix} \frac{\sin(\theta^*)\sin(\theta^*-\theta)}{\sin^2(\theta^*)} \\ \frac{\sin(\theta)}{\sin(\theta^*)} \end{bmatrix}$$

$$= \beta\alpha^\beta r^{\beta-1}\begin{bmatrix} \frac{\sin(\theta^*-\theta)}{\sin(\theta^*)} \\ \frac{\sin(\theta)}{\sin(\theta^*)} \end{bmatrix},$$

as desired.

$\square$

We prove Lemma 11.

*Proof of Lemma 11.* By construction, we see that $z \in \left\{\mathbf{U}p \mid p \in \mathbb{R}^D_{\geq 0}\right\}$. We can apply Lemma 10 to see that

$$\frac{\partial^2 c_{\mathbf{U}}(z)}{\partial z_1 \partial z_2} = \frac{\partial^2}{\partial z_1 \partial z_2}\left(\sin^{-2\beta}(\theta^*)\left(z_1^2 + z_2^2 - 2z_1z_2\cos(\theta^*)\right)^{\frac{\beta}{2}}\right)$$

$$= \frac{\partial}{\partial z_2}\left(\beta\alpha^\beta \sin^{-\beta}(\theta^*)\left(z_1^2 + z_2^2 - 2z_1z_2\cos(\theta^*)\right)^{\frac{\beta}{2}-1}(z_1 - z_2\cos(\theta^*))\right)$$

$$= \beta\alpha^\beta \sin^{-\beta}(\theta^*)\frac{\partial}{\partial z_2}\left(\left(z_1^2 + z_2^2 - 2z_1z_2\cos(\theta^*)\right)^{\frac{\beta}{2}-1}(z_1 - z_2\cos(\theta^*))\right).$$

This is the same sign as:

$$\frac{\partial}{\partial z_2}\left(\left(z_1^2 + z_2^2 - 2z_1z_2\cos(\theta^*)\right)^{\frac{\beta}{2}-1}(z_1 - z_2\cos(\theta^*))\right)$$

$$= (\beta - 2)\left(z_1^2 + z_2^2 - 2z_1z_2\cos(\theta^*)\right)^{\frac{\beta}{2}-2}(z_1 - z_2\cos(\theta^*))(z_2 - z_1\cos(\theta^*)) - \left(z_1^2 + z_2^2 - 2z_1z_2\cos(\theta^*)\right)^{\frac{\beta}{2}-1}\cos(\theta^*)$$

$$= \left(z_1^2 + z_2^2 - 2z_1z_2\cos(\theta^*)\right)^{\frac{\beta}{2}-2}\left((\beta-2)(z_1 - z_2\cos(\theta^*))(z_2 - z_1\cos(\theta^*)) - \cos(\theta^*)\left(z_1^2 + z_2^2 - 2z_1z_2\cos(\theta^*)\right)\right)$$

This is the same sign as:

$$(\beta - 2)(z_1 - z_2\cos(\theta^*))(z_2 - z_1\cos(\theta^*)) - \cos(\theta^*)\left(z_1^2 + z_2^2 - 2z_1z_2\cos(\theta^*)\right).$$

Let's represent $z$ as $[r\cos(\theta), r\cos(\theta^* - \theta)]$. The above expression is the same sign as:

$$(\beta - 2)(\cos(\theta) - \cos(\theta^* - \theta)\cos(\theta^*))(\cos(\theta^* - \theta) - \cos(\theta)\cos(\theta^*)) - \cos(\theta^*)\sin^2(\theta^*)$$

$$= (\beta - 2)(\sin(\theta^*)\sin(\theta^* - \theta))(\sin(\theta)\sin(\theta^*)) - \cos(\theta^*)\sin^2(\theta^*)$$

$$= \sin^2(\theta^*)\left((\beta - 2)\sin(\theta^* - \theta)\sin(\theta) - \cos(\theta^*)\right).$$

This is the same sign as:

$$(\beta-2)\sin(\theta^*-\theta)\sin(\theta) - \cos(\theta^*) = (\frac{\beta}{2}-1)(\cos(\theta^*-2\theta) - \cos(\theta^*)) - \cos(\theta^*) = \left(\frac{\beta}{2}-1\right)(\cos(\theta^*-2\theta)) - \frac{\beta}{2}\cos(\theta^*).$$

This is the same sign as:

$$\frac{\beta - 2}{\beta}\cos(\theta^* - 2\theta) - \cos(\theta^*).$$

$\square$

We prove Lemma 12.

*Proof of Lemma 12.* By Lemma 11, we see that $\left(\frac{\beta-2}{\beta}\cos(\theta^* - 2\theta) - \cos(\theta^*)\right)$ has the same sign as $\frac{\partial^2 c_{\mathbf{U}}(z)}{\partial z_1 \partial z_2}$. Thus it suffices to show that $g'(z_1) \cdot \frac{\partial^2 c_{\mathbf{U}}(z)}{\partial z_1 \partial z_2} \leq 0$. When $\frac{\partial^2 c_{\mathbf{U}}(z)}{\partial z_1 \partial z_2} = 0$, the condition in the proposition statement is trivially satisfied. We thus assume for the remainder of the proof that $\frac{\partial^2 c_{\mathbf{U}}(z)}{\partial z_1 \partial z_2} \neq 0$.

The second-order condition for $z$ to be a maximizer of equation (10) is the following:

$$\begin{bmatrix} h_1'(z_1) & 0 \\ 0 & b \quad h_2'(z_2) \end{bmatrix} - \nabla^2 c_{\mathbf{U}}(z) \preceq 0. \tag{49}$$

Let's apply Lemma 10, to see that:

$$h_1(x) = \frac{\partial c_{\mathbf{U}}([x, g(x)])}{\partial z_1}.$$

Since this holds in a neighborhood of $z_1$, we see that:

$$h_1'(z_1) = \frac{\partial^2 c_{\mathbf{U}}(z)}{\partial z_1^2} + g'(z_1)\frac{\partial^2 c_{\mathbf{U}}(z)}{\partial z_1 \partial z_2}.$$

An analogous argument, coupled with the inverse function theorem, shows that:

$$h_2'(z_2) = \frac{\partial^2 c_{\mathbf{U}}(z)}{\partial z_2^2} + \frac{1}{g'(z_1)}\frac{\partial^2 c_{\mathbf{U}}(z)}{\partial z_1 \partial z_2}.$$

Plugging this into equation (49), we obtain:

$$\begin{aligned} 0 \succeq & \begin{bmatrix} h_1'(z_1) & 0 \\ 0 & b \quad h_2'(z_2) \end{bmatrix} - \nabla^2 c_{\mathbf{U}}(z) \\ = & \begin{bmatrix} \frac{\partial^2 c_{\mathbf{U}}(z)}{\partial z_1^2} + g'(z_1)\frac{\partial^2 c_{\mathbf{U}}(z)}{\partial z_1 \partial z_2} & 0 \\ 0 & \frac{\partial^2 c_{\mathbf{U}}(z)}{\partial z_2^2} + \frac{1}{g'(z_1)}\frac{\partial^2 c_{\mathbf{U}}(z)}{\partial z_1 \partial z_2} \end{bmatrix} - \nabla^2 c_{\mathbf{U}}(z) \\ = & \begin{bmatrix} g'(z_1)\frac{\partial^2 c_{\mathbf{U}}(z)}{\partial z_1 \partial z_2} & -\frac{\partial^2 c_{\mathbf{U}}(z)}{\partial z_1 \partial z_2} \\ -\frac{\partial^2 c_{\mathbf{U}}(z)}{\partial z_1 \partial z_2} & \frac{1}{g'(z_1)}\frac{\partial^2 c_{\mathbf{U}}(z)}{\partial z_1 \partial z_2} \end{bmatrix} \\ = & \frac{\partial^2 c_{\mathbf{U}}(z)}{\partial z_1 \partial z_2} \begin{bmatrix} g'(z_1) & -1 \\ -1 & \frac{1}{g'(z_1)} \end{bmatrix}. \end{aligned}$$

When $\frac{\partial^2 c_{\mathbf{U}}(z)}{\partial z_1 \partial z_2} = 0$, the condition in the proposition statement is trivially satisfied. Since we've assumed that $\frac{\partial^2 c_{\mathbf{U}}(z)}{\partial z_1 \partial z_2} \neq 0$, the eigenvectors are $[1, g'(u)]$ which has eigenvalue 0 and $[-g'(u), 1]$ which has eigenvalue

$$\frac{(g'(z_1))^2 + 1}{g'(z_1)} \cdot \frac{\partial^2 c_{\mathbf{U}}(z)}{\partial z_1 \partial z_2}.$$

The sign of that eigenvalue is equal to the sign of $g'(z_1) \cdot \frac{\partial^2 c_{\mathbf{U}}(z)}{\partial z_1 \partial z_2}$. Since the matrix must be negative semidefinite, we see that $g'(z_1) \cdot \frac{\partial^2 c_{\mathbf{U}}(z)}{\partial z_1 \partial z_2} \leq 0$.

$\square$

# H   Proofs for Section 4

We prove Proposition 2, restated below.

**Proposition 2.** *Suppose that*

$$\max_{\|p\| \leq 1} \min_{1 \leq i \leq N} \left\langle p, \frac{u_i}{\|u_i\|} \right\rangle < N^{-P/\beta}. \tag{8}$$

*Then for any symmetric equilibrium $\mu$, the profit $\mathcal{P}^{eq}(\mu)$ is strictly positive.*

*Proof.* Without loss of generality, we assume user vectors have unit norm $\|u_i\|$. Given an equilibrium $\mu$, we will construct an explicit vector $p$ that generates positive profit. This proves that the equilibrium profit is positive because no vector can achieve higher than the equilibrium profit. The vector $p$ is of the form $\left(Q\left(\max_{p'\in\text{supp}(\mu)}\|p'\|\right)+\epsilon\right)\cdot u_{i^*}$ for some $i^*\in[1,N]$.

Cluster the set of unit vectors $p$ into $N$ groups $G_1,\ldots,G_N$, based on the user for whom they generate the lowest value. That is, each vector $p$ belongs to the group $G_i$ where $u_i=\text{argmin}_{1\le i'\le N}\langle p,u_{i'}\rangle$. This means that if all producers choose (unit vector) directions in $G_i$, then the maximum value received by user $u_i$ is

$$\max_{1\le j\le P}\langle p_j,u_i\rangle \le \max_{\|p\|\le 1}\min_{1\le i\le N}\langle p,u_i\rangle = Q. \tag{50}$$

Let $G_{i^*}$ be the group with highest probability of appearing in $\mu$, that is $i^*\in\text{arg}\max_i\mathbb{P}_{v\sim\mu}\left[\frac{v}{\|v\|}\in G_i\right]$.

Let $E$ be the event that all of the other $P-1$ producers choose directions in $G_{i^*}$. The event $E$ happens with probability at least $\mathbb{P}_{v\sim\mu}\left[\frac{v}{\|v\|}\in G_{i^*}\right]\ge(1/N)^{P-1}$. Since the value received by the user is linear in the magnitude of the producer action, we see that the maximum possible value that could be received by user $u_i$ from the other producers is $Q\left(\max_{p'\in\text{supp}(\mu)}\|p'\|\right)$. On the other hand, the action $p$ results in value $\left(Q\left(\max_{p'\in\text{supp}(\mu)}\|p'\|\right)+\epsilon\right)$ for $u_{i^*}$, so it wins $u_{i^*}$ with probability 1 on the event $E$. This means that the expected profit obtained by $p$ is at most

$$\left(\frac{1}{N}\right)^{P-1}-\left(Q\left(\max_{p'\in\text{supp}(\mu)}\|p'\|\right)+\epsilon\right)^{\beta}.$$

Taking a limit as $\epsilon\to_+0$, we obtain the profit can be set arbitrarily close to:

$$\left(\frac{1}{N}\right)^{P-1}-\left(Q\left(\max_{p'\in\text{supp}(\mu)}\|p'\|\right)\right)^{\beta}. \tag{51}$$

It suffices to bound $\max_{p'\in\text{supp}(\mu)}\|p'\|$. The action $p''\in\text{arg}\max_{p'\in\text{supp}(\mu)}\|p'\|$ produces a profit of at most $N-\left(\max_{p\in\text{supp}(\mu)}\|p\|\right)^{\beta}$. Thus, $\left(\max_{p\in\text{supp}(\mu)}\|p\|\right)^{\beta}\le N$, so $\left(\max_{p\in\text{supp}(\mu)}\|p\|\right)\le N^{1/\beta}$.

Plugging this into (51), we see that there exist actions that produces profit arbitrarily close to

$$\left(\frac{1}{N}\right)^{P-1}-NQ^{\beta}.$$

Thus, a strictly positive profit will be obtained if:

$$Q<\left(\frac{1}{N}\right)^{P/\beta},$$

as desired. $\qquad\square$

We prove Proposition 3, restated below.

**Proposition 3.** *If $\mu$ is a single-genre equilibrium, then the profit $\mathcal{P}^{eq}(\mu)$ is equal to 0.*

*Proof.* Since $\mu$ is an equilibrium, all choices $p$ in the support of $\mu$ achieve profit equal to the equilibrium profit. We apply Lemma 3 to see that the cdf of $\mu$ is $F(p)=\min\left(1,\left(\frac{p^{\beta}}{N}\right)^{1/(P-1)}\right)$, which shows that $p=0$ is in $\text{supp}(\mu)$. For this choice of $p$, the cost is 0, but the producer also never wins any users, so the profit is also zero, as claimed. $\qquad\square$