# OpenReview forum: "Supply-Side Equilibria in Recommender Systems"
_NeurIPS.cc/2023/Conference — NeurIPS 2023 poster_

### Official Review · Reviewer_rJ3F · 2023-07-05

**Soundness:** 3 good
**Presentation:** 3 good
**Contribution:** 2 fair
**Rating:** 5
**Confidence:** 3

**Summary:**

The paper presents a theoretical study regarding the equilibrium of supply-side competition in recommender systems. In particular, the analysis examined when and how the recommender system influence producers' creation of online contents and will specialization occur under the effect of recommender systems.

**Strengths:**

1. Examining the supply-side competition under the influence of recommender system is an interesting topic to explore.

2. The paper presents theoretical analysis for the equilibrium of supply-side competition and specialization.

**Weaknesses:**

1. The theoretical analysis is based on a very simplified recommendation model, i.e, inner-product between the user and item vectors which is basically from the idea of matrix factorization. However, modern recommender systems have advanced much more beyond MF, and real-world systems are driven by deep neural networks and large language models. As a result, it's not clear if the theoretical analysis really sheds light on real-world recommender systems.

2. There is no experimental analysis, even simulated analysis is missing.

**Questions:**

1. To what extent the theoretical analysis reflects real-world recommender systems that are driven by deep learning and LLMs.

2. Is there any experimental analysis to show the predicted effects, at least based on simulation.

**Limitations:**

The paper does not discuss limitations.

---

> ### Author Rebuttal · Authors · 2023-08-10
>
> We thank the reviewer for their feedback and respond to their questions below. We present new results that address some of their concerns: in particular, **a new empirical analysis on the MovieLens-100K dataset** that validates and goes beyond our theoretical findings (see the General Response).
>
> **“Is there any experimental analysis to show the predicted effects”**
>
> We provide a new empirical analysis of supply-side equilibria of nonnegative matrix factorization on the MovieLens-100K dataset. These experiments beyond qualitative insights that validate our theoretical results and also provide intuition for the structure of supply-side equilibria that goes beyond our theoretical results.
>
> - We compute the direction of the single-genre equilibrium embedding (using Corollary 5) for several different cost functions (see Figures N1-N2 in the General Response pdf). (Recall that this direction is equal to the genre chosen by producers when there is no specialization at equilibrium.) Figures N1-N2 confirm our theoretical findings that the location of the genre does not seem to permit a clean closed-form characterization (Corollary 5) and depends on subtleties of the cost function. Moreover,  Figures N1-N2 go beyond our theoretical results to illustrate that the genre realized in a marketplace can be heavily influenced by some (but not other) aspects of producer costs.
> - We compute the boundary where specialization starts to occur, using the upper bound from Corollary 4 (see Figure N3 in the General Response pdf). As $D$ increases, specialization is more likely to occur in a given marketplace. The intuition is that as $D$ increases, the user embeddings become more heterogeneous, which increases the likelihood of specialization, as suggested by our theoretical results in Section 3. Figure N3 further indicates that the platform may influence the level of specialization by setting the embedding dimension $D$ to be higher or lower. We also show how whether specialization occurs subtly depends on the cost function structure, which also aligns with our theoretical results from Section 3.
>
> We refer the reviewer to the General Response for details of the empirical setup and results.
>
> **“The theoretical analysis is based on a very simplified recommendation model, i.e, inner-product between the user and item vectors which is basically from the idea of matrix factorization…To what extent the theoretical analysis reflects real-world recommender systems that are driven by deep learning and LLMs?”**
>
> Our formalism is not limited to simple matrix factorization algorithms. If we interpret user and movie vectors as “embeddings” *learned* by the algorithm, our formalism can also capture any deep learning-based recommendation systems that learn embeddings for the users and movies. In particular, the process by which these embeddings are learned can be arbitrarily complex, as long as the recommendation system ultimately evaluates user value based on the inner product of the learned embeddings.
>
> As to language-model based recommendation systems, it is not clear if our model is a perfect fit for these newer systems. Understanding the supply-side equilibria of language model-based recommendation systems would be a very interesting direction for future work. We hope that our analysis of specialization inspires future work on this topic.
>
> As a broader point about the stylized nature of our model, we refer the reviewer to “Nature of our contribution” in the General response. We don’t think that incorporating all of the complexities of real-world recommender systems into our model would have improved our analysis of specialization. If a model has too many changing components and degrees of freedom, it is hard to pinpoint exactly which component is responsible for a given phenomena. In general, complex models are often less robust, whereas results from simple models tend to at least qualitatively generalize beyond their assumptions. Our stylized model is one of the simplest that enables us to study specialization (see Appendix A.1).

---

> > ### Author Response · Authors · 2023-08-18
> >
> > Reviewer rJ3F: thank you again for your review. We wanted to bring your attention to **our new empirical analysis on the MovieLens dataset in the rebuttal**, which we believe might address your concerns about the lack of empirical validation of our results. We also respond to your other questions in the rebuttal. If you have follow-up questions, we are happy to further clarify any aspect of the paper or the rebuttal in the remaining few days of the discussion period.

---

### Official Review · Reviewer_sbgm · 2023-07-05

**Soundness:** 3 good
**Presentation:** 3 good
**Contribution:** 3 good
**Rating:** 6
**Confidence:** 1

**Summary:**

This paper aims to understand the equilibria of the digital content producer side competition in the recommender platforms. It specifically studies the potential of specialization, where different producers create different type of content. The paper proposed to model value of product as the inner product of user and item embeddings, where the personalization, product cost, as well as producer profit are derived from the embedding model. Based on this setup, the paper studies the equilibria of the producer side competition. It provides conditions under which specialization occurs and the form it takes. It claims that specialization can reduce the competitiveness of the marketplace by decreasing competition within each genre.

**Note** It should be noted that I am not deeply familiar with the field of algorithmic game theory. As a result, I may lack an in-depth understanding of the conceptual links between various aspects of the theory used in the paper and the broader field. Therefore, I may not be adequately equipped to fully assess the novelty and correctness of the technical content. While effort has been made to understand and summarize the paper, the detailed insights on the specific game-theoretic models and their implications could benefit from a review by a scholar who is more specialized in this field.

**Strengths:**

* The study of the influence of recommender system on the digital content production and diversity of the content is an important topic.
* The insights drawn from the study of  specialization could potentially provide new directions for recommender model design.


**Weaknesses:**

* The current model presented in the paper seems to be specifically tailored towards digital goods such as music, movies, and news. It is uncertain how this model might translate to other types of recommender systems such as e-commerce, dating, or job recommendation platforms, which also hold significant relevance in today's digital marketplace. Although the authors have proposed directions for future work, further investigation is required to expand this model to other sectors and recommendation scenarios, enhancing the universal applicability of the research.
* While the model provides valuable theoretical insights, it could be strengthened by empirical validation. Observations or experiments on real-world recommender systems could further test and substantiate the authors' claims.

**Questions:**

* Given the theoretical framework proposed in the paper, how can this study be practically utilized, particularly for practitioners who are actively involved in the development of new recommender system models? The practical implications and real-world applications of the theoretical findings would be of interest to the broader research community and industry practitioners. Clarification on this point would further enhance the utility and impact of the work.

**Limitations:**

The limitations is discussed in the paper.

---

> ### Author Rebuttal · Authors · 2023-08-10
>
> We thank the reviewer for their feedback and respond to their questions below. We present new results that address some of their concerns: in particular, **a new empirical analysis on the MovieLens-100K dataset** that validates and goes beyond our theoretical findings (see the General Response).
>
> **“While the model provides valuable theoretical insights, it could be strengthened by empirical validation.”**
>
> We provide a new empirical analysis of supply-side equilibria of nonnegative matrix factorization on the MovieLens-100K dataset. These experiments beyond qualitative insights that validate our theoretical results and also provide intuition for the structure of supply-side equilibria that goes beyond our theoretical results.
> - We compute the direction of the single-genre equilibrium embedding (using Corollary 5) for several different cost functions (see Figures N1-N2 in the General Response pdf). (Recall that this direction is equal to the genre chosen by producers when there is no specialization at equilibrium.) Figures N1-N2 confirm our theoretical findings that the location of the genre does not seem to permit a clean closed-form characterization (Corollary 5) and depends on subtleties of the cost function. Moreover,  Figures N1-N2 go beyond our theoretical results to illustrate that the genre realized in a marketplace can be heavily influenced by some (but not other) aspects of producer costs.
> - We compute the boundary where specialization starts to occur, using the upper bound from Corollary 4 (see Figure N3 in the General Response pdf). As $D$ increases, specialization is more likely to occur in a given marketplace. The intuition is that as $D$ increases, the user embeddings become more heterogeneous, which increases the likelihood of specialization, as suggested by our theoretical results in Section 3. Figure N3 further indicates that the platform may influence the level of specialization by setting the embedding dimension $D$ to be higher or lower. We also show how whether specialization occurs subtly depends on the cost function structure, which also aligns with our theoretical results from Section 3.
>
> We refer the reviewer to the General Response for details of the empirical setup and results.
>
> **“The current model presented in the paper seems to be specifically tailored towards digital goods such as music, movies, and news. It is uncertain how this model might translate to other types of recommender systems such as e-commerce, dating, or job recommendation platforms, which also hold significant relevance in today's digital marketplace.”**
>
>  As the reviewer notes, the current model presented in the paper is tailored towards digital content. This is intentional: as discussed in the introduction, the supply-side effects of digital content recommendation platforms are poorly understood, and understanding these particular supply-side effects is the focus of our paper. Other types of recommender systems, such as for dating or for jobs, have different structure, in terms of interaction between participants, incentive and strategic behavior, and the platform’s objective. We believe that grouping these two types of marketplaces into one single model would hinder the derivation of  conclusive economic insights.
>
> **“Given the theoretical framework proposed in the paper, how can this study be practically utilized, particularly for practitioners who are actively involved in the development of new recommender system models?”**
>
> One insight from our results is that the practitioners can influence the long-run equilibrium content landscape through their choice of embedding dimension. As shown in our empirical analysis (Figure N3), the embedding dimension impacts the cost function exponent at which specialization starts to occur in a marketplace. What this means is that a practitioner can increase the embedding dimension to increase the level of specialization in the marketplace (whether every producer produces similar genres or very different ones). More generally, our work shows that platforms can influence the amount of specialization through their recommender systems.
>
> As to whether the practitioners should opt to induce specialization or not, this depends on several factors. Our results highlight two consequences of specialization—(1) content diversity, and (2) positive producer profit—which should both impact the platform’s decision.
> - Specialization leads to content diversity, which impacts the long-term satisfaction of users and thus the long-run revenue of the platform. On the positive side, content diversity can provide users with content tailored to their interests, which may help attract and retain a wider user base. On the other hand, content diversity and the consumption of niche content may inadvertently drive filter bubbles, polarization, or other negative user experiences.
> - Specialization can also lead producers to earn a positive profit at equilibrium (Section 4). This has negative implications for content quality which might reduce the long-term satisfaction of users (and thus lower user retention). On the other hand, this has positive implications for producers, which might improve producer retention.
>
> The platform can use knowledge of its specific marketplace to balance the positive and negative effects, and determine whether specialization improves revenue in a given marketplace.

---

> > ### Author Response · Authors · 2023-08-18
> >
> > Reviewer sbgm: thank you again for your review. We wanted to bring your attention to **our new empirical analysis on the MovieLens dataset in the rebuttal**, which we believe might address your concerns about the lack of empirical validation of our results. We also respond to your other questions in the rebuttal. If you have follow-up questions, we are happy to further clarify any aspect of the paper or the rebuttal in the remaining few days of the discussion period.

---

> > > ### Comment · Reviewer_sbgm · 2023-08-19
> > >
> > > Thanks for the answers, they have addressed my concerns.

---

### Official Review · Reviewer_2uvz · 2023-07-07

**Soundness:** 3 good
**Presentation:** 3 good
**Contribution:** 2 fair
**Rating:** 6
**Confidence:** 4

**Summary:**

This work studies the supply-side equilibria in content recommender platforms. The authors proposed a game-theoretic model to describe content creators' competition and derive necessary and sufficient conditions under which the specialization over genres occurs or does not occur at the equilibrium.

**Strengths:**

The motivation is well justified, and the proposed problem setting is interesting in general.
The technique used for the main result is novel and solid, and the insight behind the theoretical results is well-explained.

**Weaknesses:**

The proposed problem itself and the solution are intriguing, but the specific problem setting considered seems to be oversimplified. In particular, the utility function defined in Eq.(1) is symmetric for different creators, and only the symmetric mixed NE is considered. I understand this is the only feasible solution concept with an existence guarantee in this situation; still, it is not a realistic model to characterize the outcome of creators' competition in practice. In the real world, creators should have heterogeneous preferences and costs, which should be reflected in their distinct strategy sets or cost functions. And also, it is hard to believe in any real market, all creators will eventually form a homogeneous belief about the production strategy (i.e., a symmetric mixed NE).

In terms of the theoretical results, my understanding is that the specialization phenomenon hinges on the joint property of the user population structure and parameter $\beta$. While I appreciate the insight about how user distribution might affect the emergence of specialization, the discussion of $\beta$ seems too restrictive to the specific form of the cost term. The current term $\|p\|^{\beta}$ is too simple to capture the nature of the cost, for example, 1. why the marginal cost has to depend on an exponential factor rather than a multiplicative factor? 2. why the cost only depends on the norm of $p$? What if we generalize the cost term in the following ways:

1. $c_j(p) = a \|p\|^{\beta}$
2. $c_j(p) = a_j \|p\|^{\beta}$
3. $c_j(p) = a \|p-p_j\|^{\beta}$
4. $c_j(p) = a_j \|p-p_j\|^{\beta}$

Can we still derive results that share similar insights using the current technique? If such an extension is promising, I would highly recommend this paper.

Minor:
L.254: should be "linearly independent vectors"

**Questions:**

Please see the questions raised in weaknesses.

1. in corollary 3, what does it mean by "N users split equally between two linearly independent vectors?"

---

> ### Author Rebuttal · Authors · 2023-08-10
>
> We thank the reviewer for their feedback and respond to their questions below. In response to the reviewers, we also showed **a new empirical analysis on the MovieLens-100K dataset** that both validates and go beyond our theoretical findings (see the General Response).
>
> **“What if we generalize the cost term in the following ways?”**
>
> We describe the applicability of our cost functions to the functional forms proposed by the reviewers.
>
> - First, our model and results directly accommodate the scaled cost function $c_S(p) = a ||p||^{\beta}$. We can define a new norm $||x||_{S} = ||x \cdot a^{1/\beta}||$, so that $c_S(p) =a ||p||^{\beta} = ||p||_S^{\beta}$. Our characterization from Theorem 1 specifies when specialization occurs for $c_S(p)$: specialization occurs for $c_S(p)$ if and only if specialization occurs for the non-scaled cost function $c(p) = ||p||^{\beta}$.
>
> - Our model and results can also accommodate the translated cost function $c_T(p) = a ||p - q||^{\beta}$. We can handle the scalar factor of $a$ by a similar argument to the above, so WLOG let’s assume that $a =1$. We claim that $X \sim \mu$ is a symmetric mixed equilibrium for $c_T(p) = ||p-q||^{\beta}$ if and only if $(X - q)$ where $X \sim \mu$ is a symmetric mixed equilibrium for $c(p) = ||p||^{\beta}$. The intuition is that we can write $\langle u, p - q \rangle = \langle u, p \rangle - \langle u,q \rangle$, which just shifts a user’s utility by the same constant factor for all producers so it can be disregarded. This gives us an equivalence between the equilibria in these two setups; however, to apply Theorem 1, we’d need to slightly change the definition of genre. We instead define $Genre(\mu)$ to be set of directions $\frac{p-q}{||p-q||}$ for $p \in \text{supp}(\mu)$. That is, the genres object would instead capture the directions along which the producers *change* from the starting point of $q$, rather than the final direction of the producers. (Note: we don’t run into any technical difficulties with the nonnegative orthant constraints, because those turn out to only be required for user vectors and not for producer vectors in our results.)
>
> As described below, we cannot directly capture heterogeneous costs in our model which precludes some of the functional forms proposed by the reviewer (see additional discussion below).
>
> **“The current term $||p||^{\beta}$  is too simple to capture the nature of the cost. For example, why the cost only depends on the norm of p?”**
>
> We allow for any norm (not just the $\ell_2$-norm) within our cost function, which captures a very broad family of functions. For example, we can take the norm $||\cdot||$ to be $\ell_q$ norm for any $q \ge 1$ as well as any weighted cost norm $||x||$ defined to be the $\ell_2$ norm of $[x_1 \cdot \alpha_1, \ldots, x_D \cdot \alpha_D]$ (we study both in our new experiments on the MovieLens-100K dataset, as described in the General Response). In more detail, weighted costs are parameterized by a $D$-dimensional vector $\alpha \in R_{\ge 0}^D$ of weights such that $\sum_{i=1}^D \alpha_i = 1$. The cost function $c_{\alpha}$ is defined to be $c_{\alpha}(p) := ||[p_1 \cdot \alpha_1, \ldots, p_D \cdot \alpha_D]||^{\beta}$ where the norm is the $\ell_2$ norm. This cost function captures that certain dimensions might be cheaper vs. more expensive for the producer to improve. We could also accomodate Mahalanobis distance, matrix norms, etc. within our model.
>
> **“The utility function defined in Eq.(1) is symmetric for different creators, and only the symmetric mixed NE is considered.”**
>
> Our model and results do require that the cost functions are homogeneous. One could have easily extended our model to allow for producers to have heterogeneous cost functions, but we focused on a single cost function to simplify the technical analysis. In fact, the technical analysis for homogeneous cost functions already required several novel technical innovations (see Appendix B.1). We would qualitatively expect that the tendency towards specialization would only be amplified if producers could have heterogeneous cost functions.
>
> That being said, our model does implicitly captures heterogeneity in *producer behaviors* via the randomness in the symmetric mixed equilibrium. Under this randomness, each producer independently samples a content vector from the equilibrium distribution, so different producers create different content in any given realization. In fact, this heterogeneity in producer behavior motivated us to formalize specialization in terms of the support of the symmetric mixed equilibrium distribution (lines 179-188).
>
> **“In corollary 3, what does it mean by "N users split equally between two linearly independent vectors?":**
>
> We assume that $u_1 = .. u_{N/2} = x_1$ and $u_{N/2+1} = … = u_N = x_2$, for some linearly independent vectors $x_1$ and $x_2$.

---

> > ### Comment · Reviewer_2uvz · 2023-08-17
> > **Response**
> >
> > Thanks for the response. After carefully digesting the response, my biggest concern stands: the framework is not able to capture the heterogeneous situation, which I believe is a very important aspect to consider in practice. Although I understand that mixed NE does allow players to take different actions, the insight that everybody eventually converges to the same mixed strategy seems unrealistic and overly simplified to me.
> >
> > However, despite the concern I have, I do appreciate the novelty of the model and the acute analysis. That said, I believe this work does have the potential to serve as a nice starting point to study the specialization effect of producer competition. Therefore, I decide to raise my score from 5 to 6.

---

### Official Review · Reviewer_ham5 · 2023-07-08

**Soundness:** 2 fair
**Presentation:** 3 good
**Contribution:** 2 fair
**Rating:** 5
**Confidence:** 5

**Summary:**

In this paper, the authors investigate the supply-side equilibria in personalized content recommender systems. They propose a game-theoretic model that captures the multi-dimensional decisions of producers and the heterogeneous preferences of users. They analyze the conditions for specialization to occur and the impact of specialization on market competitiveness. The paper provides insights into how recommender systems shape the diversity and quality of content created by producers.

**Strengths:**

(1) The paper addresses an important and timely topic – the impact of recommender systems on the supply side of the digital goods market. It sheds light on how producers make decisions to maximize their appearance in recommendations, and how this affects the diversity and competitiveness of the marketplace.

(2) The proposed game-theoretic model is well-designed and captures the multi-dimensional decision space of producers and the heterogeneity of user preferences. This model allows for a nuanced analysis of specialization and its consequences in recommender systems.

(3) The paper provides rigorous theoretical analysis, deriving necessary and sufficient conditions for specialization to occur. It also presents concrete settings with two populations of users to characterize the distribution of content at equilibrium.

**Weaknesses:**

(1) The implementation details of the proposed model are not provided. It would be helpful for readers to have a clear understanding of how the model can be replicated and reproduced.

(2) The evaluation of the proposed model is limited. The paper does not compare the results with any existing baselines or alternative approaches. It would be valuable to see how the proposed model performs compared to other methods in the field.

**Questions:**

My questions are mentioned above. You can provide the feedback in the rebuttal phase.

---

> ### Author Rebuttal · Authors · 2023-08-10
>
> We thank the reviewer for their feedback and respond to their questions below. We present new results that address some of their concerns: in particular, a **new empirical analysis on the MovieLens-100K dataset** that validates and goes beyond our theoretical findings (see the General Response).
>
> **“The evaluation of the proposed model is limited.”**
>
> We provide a new empirical analysis of supply-side equilibria of nonnegative matrix factorization on the MovieLens-100K dataset. These experiments beyond qualitative insights that validate our theoretical results and also provide intuition for the structure of supply-side equilibria that goes beyond our theoretical results.
>
> - We compute the direction of the single-genre equilibrium embedding (using Corollary 5) for several different cost functions (see Figures N1-N2 in the General Response pdf). (Recall that this direction is equal to the genre chosen by producers when there is no specialization at equilibrium.) Figures N1-N2 confirm our theoretical findings that the location of the genre does not seem to permit a clean closed-form characterization (Corollary 5) and depends on subtleties of the cost function. Moreover,  Figures N1-N2 go beyond our theoretical results to illustrate that the genre realized in a marketplace can be heavily influenced by some (but not other) aspects of producer costs.
>
> - We compute the boundary where specialization starts to occur, using the upper bound from Corollary 4 (see Figure N3 in the General Response pdf). As $D$ increases, specialization is more likely to occur in a given marketplace. The intuition is that as $D$ increases, the user embeddings become more heterogeneous, which increases the likelihood of specialization, as suggested by our theoretical results in Section 3. Figure N3 further indicates that the platform may influence the level of specialization by setting the embedding dimension $D$ to be higher or lower. We also show how whether specialization occurs subtly depends on the cost function structure, which also aligns with our theoretical results from Section 3.
>
> We refer the reviewer to the General Response for details of the empirical setup and results.
>
> **“The implementation details of the proposed model are not provided. It would be helpful for readers to have a clear understanding of how the model can be replicated and reproduced.”**
>
> As a primarily theoretical contribution (except for the new experiments on the MovieLens dataset), we believe that we have fully specified our model in Section 2, and included all the theoretical proofs in the attached supplement. As such, we believe that our results are fully replicable and verifiable by readers–if there is a particular detail that is unclear, we would appreciate if the reviewer could point it out so we can clarify it.
>
> **“The paper does not compare the results with any existing baselines or alternative approaches. It would be valuable to see how the proposed model performs compared to other methods in the field.”**
>
> It is not clear what “baselines” would mean in our model. Our goal in this paper is not an algorithm or a method. Rather, as is typical with many papers in the economics and machine learning literature, we develop a mathematical model and analyze the model to provide economic insights about real-world marketplaces. Our specific goal in this paper is to study content creator incentives in recommender systems, and in particular, the economic phenomena of specialization. For more context about the style of our contribution, we refer the reviewer to the “Nature of contribution” section of the General Response.

---

> > ### Author Response · Authors · 2023-08-18
> >
> > Reviewer ham5: thank you again for your review. We wanted to bring your attention to **our new empirical analysis on the MovieLens dataset in the rebuttal**, which we believe might address your concerns about the lack of empirical validation of our results. We also respond to your other questions in the rebuttal. If you have follow-up questions, we are happy to further clarify any aspect of the paper or the rebuttal in the remaining few days of the discussion period.

---

### Author Rebuttal · Authors · 2023-08-10

Thanks to the reviewers for their feedback. We provide a **new empirical analysis of our theoretical findings on the MovieLens-100K dataset**. We then clarify the nature of our contribution of proposing and analyzing a mathematical model to study an economic phenomenon. (We respond individually to the reviewers below.)

## Empirical analysis on MovieLens dataset

Several reviewers asked about an empirical analysis of our theoretical findings on real-world datasets. We provide an empirical analysis of supply-side equilibria using the MovieLens-100K dataset and recommendations based on nonnegative matrix factorization (NMF). These experiments provide qualitative insights that validate our theoretical results and provide intuition going beyond our theoretical results.

We construct user embeddings of dimension $D = 2, 3, 5, 10, 50$ by running NMF with $D$ factors. We consider two families of producer cost functions:

- (C1) Let $\alpha \in R_{\ge 0}^D$ be a weight vector such that $\sum_{i=1}^D \alpha_i = 1$. The cost function $c_{\alpha}(p) := ||[p_1 \cdot \alpha_1, \ldots, p_D \cdot \alpha_D]||^{\beta}$ captures that certain dimensions might be cheaper vs. more expensive to improve along.

- (C2) Let $q \ge 1$ and let $c_q(p) := ||p||_q^{\beta}$.

**Single-genre equilibrium direction**: We compute the direction of the single-genre equilibrium embedding (using Corollary 5) for cost functions in (C1) (see Figure N1 in pdf) and in (C2) (Figure N2 in pdf). (Recall that this direction is equal to the genre chosen by producers when there is no specialization at equilibrium.) We observe the following:
- For (C1), the genre varies significantly with the weights $\alpha$ (see Figure N1). The magnitude of the genre coordinate is higher along the cheaper dimension.
- For (C2), the genre does not change significantly with the norm parameter $q$ (see Figure N2).
- In both cases, the genre typically does not coincide with the arithmetic mean of the users.

Figures N1-N2 align with our theoretical findings that the genre location doesn’t permit a clean closed-form characterization (Corollary 5) and depends on subtleties of the cost function. Figures N1-N2 also go beyond our theoretical results to show that the genre can be heavily influenced by some (but not other) aspects of producer costs.

**Boundary where specialization starts to occur**: We investigate the cost function exponent $\beta^*$ where specialization starts to occur (line 224, Theorem 1). We compute an upper bound $\beta^u$ on $\beta^*$ (using Corollary 4) for cost functions in (C1) and different embedding dimensions $D$ (Figure N3). We observe the following:
- As $D$ increases, the value of $\beta^u$ *decreases*, and specialization is more likely to occur (see Figure N3). The intuition is that increasing $D$ increases the heterogeneity of user embeddings, which increases the likelihood of specialization as suggested by our theoretical results in Section 3. Figure N3 suggests that the platform may influence the level of specialization by tuning $D$.
- As the cost function parameter $q$ increases, the value of $\beta^u$ *increases* and specialization is less likely to occur (see Figure N3). This confirms our theoretical insights about the subtle role of the cost function in whether specialization occurs.

**Details of empirical setup**: We use the MovieLens 100K dataset which consists of 943 users, 1682 movies, and 100,000 ratings. To obtain $D$-dimensional user embeddings, we ran NMF (with $D$ factors) using the scikit-surprise library on the full dataset. For Figure N3, we directly calculate $\beta^u$ using Corollary 4. For Figures N1-N2, we numerically solve the optimization program in Corollary 5: we directly solve this using CVXPY for (C1), and we use projected gradient descent with step size 1.0 over 100 iterations with projection done with CVXPY for (C2).

We will use the extra page in the final version to include the details of this empirical analysis.

## Nature of contribution: Proposing and analyzing a stylized model

Some of the reviewers asked for comparison against baselines or asked questions about the stylized nature of our model. We would like to clarify the nature and style of our contribution.

**Proposing and analyzing a mathematical model:** Our contribution is to develop a mathematical model and analyze the model to provide economic insights about real-world marketplaces.  As a result, we do not design an algorithm or a method, but rather analyze *behavior* within the mathematical model that we propose (i.e., *specialization* within a model for content creator competition).

Proposing and analyzing behavior within a mathematical model is standard in economics and machine learning, machine learning theory, and the societal aspects of machine learning more broadly (e.g. [A], [B], [C]). These subfields are listed in the NeurIPS call for papers: Theory (algorithmic game theory) and Social and economic aspects of machine learning (strategic behavior).

[A] Lydia T. Liu, Sarah Dean, Esther Rolf, Max Simchowitz, and Moritz Hardt. “Delayed Impact of Fair Machine Learning”. ICML 2018 Best Paper.

[B] Simon Zhuang and Dylan Hadfield-Menell. “Consequences of Misaligned AI.” NeurIPS 2020.

[C] Kate Donahue and Jon Kleinberg. “Optimality and Stability in Federated Learning: A Game-Theoretic Approach.” NeurIPS 2021.

**Stylized nature of our model:** We don’t think that incorporating all of the complexities of real-world recommender systems into our model would have improved our analysis of specialization. If a model has too many changing components and degrees of freedom, it is hard to pinpoint exactly which component is responsible for a given phenomena. In general, complex models are often less robust, whereas results from simple models tend to at least qualitatively generalize beyond their assumptions. Our stylized model is one of the simplest that enables us to study specialization (see Appendix A.1).

---

### Decision · Program_Chairs · 2023-09-21

**Decision:**

Accept (poster)

**Comment:**

The paper studies the supply-side equilibria in personalized content recommender systems balancing producer incentives along the heterogeneous preferences of users. All the reviewers appreciate the timely topic studied in the paper, the rigorous theoretical analysis under the game-theoretic model framework, and the insights well explained. Multiple reviewers did ask for more discussions on the practicality of the proposed model, e.g., theories are built upon stylized models, and empirical validation. Please consider address them in your manuscript.